# Oxytocin induces the formation of distinctive cortical representations and cognitions biased toward familiar mice

David Wolf [1,2], Renée Hartig [1], Yi Zhuo[1], Max F. Scheller[1], Mirko Articus[1,2], Marcel Moor[1], Valery Grinevich [3], Christiane Linster[4], Eleonora Russo [1,5], Wolfgang Weber-Fahr[6], Jonathan R. Reinwald [1,2,6] & Wolfgang Kelsch [1,2] ✉

Social recognition is essential for the formation of social structures. Many times, recognition comes with lesser exploration of familiar animals. This lesser exploration has led to the assumption that recognition may be a habituation memory. The underlying memory mechanisms and the thereby acquired cortical representations of familiar mice have remained largely unknown, however. Here, we introduce an approach directly examining the recognition process from volatile body odors among male mice. We show that volatile body odors emitted by mice are sufficient to identify individuals and that more salience is assigned to familiar mice. Familiarity is encoded by reinforced population responses in two olfactory cortex hubs and communicated to other brain regions. The underlying oxytocin-induced plasticity promotes the separation of the cortical representations of familiar from other mice. In summary, neuronal encoding of familiar animals is distinct and utilizes the cortical representational space more broadly, promoting storage of complex social relationships.

Social recognition memory (SRM) relies on the sensory discrimination of individuals and the retrieval of the memory of being familiar with an individual[1]. Such recognition is relevant to parental care[2], pair bonding[3] and, more generally, the formation of social structures among unrelated individuals. Insights into the behavioral expression and molecular pathways of same-sex recognition have emerged[4–7]. Most studies however examine one behavioral consequence that follows the actual recognition[8–12]. Specifically, the behavioral consequence is the bias to spend less time with a familiar than with a novel animal. The actual recognition process preceding the approach decision, is however usually not studied in rodents. Thus, little is known about how neuronal representations of familiar animals are modified to recognize them as such[1,4,13,14].

Animals integrate information from multiple sensory modalities to recognize others. However, only olfactory cues may be available in many natural conditions. In rodents, sampling of odor information from an interaction partner has been considered mostly in close proximity, such as during anogenital sniffing[15–17]. Social recognition from volatile odors would be more efficient because they can be sampled at a distance, reducing the risk of aggression and serving as a guide for a wide repertoire of behaviors.

Social recognition is a memory that is enabled by state modulation through the neuropeptide oxytocin (OXT)[9–11,18–23]. Boosting OXT release when rodents first meet prolongs the duration of the behavioral SRM, while depletion of OXT receptors in the anterior olfactory nucleus (AON) prevents its behavioral expression[24]. These findings

[1]Department of Psychiatry and Psychotherapy, University Medical Center, Johannes Gutenberg University, 55131 Mainz, Germany. [2]Department of Psychiatry and Psychotherapy, Central Institute of Mental Health, Medical Faculty Mannheim, Heidelberg University, 68159 Mannheim, Germany. [3]Department of Neuropeptide Research in Psychiatry, Central Institute of Mental Health, Medical Faculty Mannheim, Heidelberg University, 68159 Mannheim, Germany. [4]Computational Physiology Laboratory, Department of Neurobiology and Behavior, Cornell University, Ithaca, New York, NY 14850, USA. [5]The BioRobotics Institute, Department of Excellence in Robotics and AI, Scuola Superiore Sant'Anna, 56127 Pisa, Italy. [6]Department of Neuroimaging, Translational Imaging, Central Institute of Mental Health, Medical Faculty Mannheim, Heidelberg University, 68159 Mannheim, Germany. ✉e-mail: wokelsch@uni-mainz.de

suggest that OXT can boost the induction of familiarity memory in the neuronal representation while a lack of OXT action prevents its formation. At the neuronal level, OXT acutely modulates the processing of sensory signals[21,24,25]. For instance, in the olfactory system of anesthetized rats, OXT receptor activation recruits AON, thereby modulating top-down inhibition on the main olfactory bulb (MOB) to increase the signal-to-noise ratio of its output neurons[24]. Yet, it is not known how and where olfactory familiarity memories are represented and whether the familiar animal is perceived more or less saliently.

There are competing hypotheses how SRM is encoded in olfactory regions. Repeated exposure to odors can result in adaptation both in behavior and in the neuronal responses of the MOB[26–28] and cortex[29]. This, together with the fewer spontaneous approaches towards familiar animals, has led to models of social recognition memory as a habituation process[29–31]. It may thus be a habituation memory in terms of an odor familiarization process that results in lower salience of familiar subjects. On the other hand, social interactions are innately rewarding[20]. A competing hypothesis thus states that social interactions reinforce the representations of these familiar animals through experience-dependent plasticity. In this case, a familiar conspecific should trigger a stronger neuronal response than a novel conspecific. We tested the competing hypotheses. We therefore establish a general approach to examine the cognition of features shared among conspecifics, applied to social recognition in mice. This experimental configuration allows for the precise presentation of volatile body odors from different individuals to test the perceived salience and single-unit population responses. We find that SRM is encoded by reinforced and more distinct population responses to the smell of the familiar than to the smell of a novel animal in a network comprising the AON and the posterior piriform cortex (pPC), but not the lateral entorhinal cortex (LEC). OXT enables the formation of such reinforced representations and this information is then transmitted top-down from AON to MOB.

## Results

### Cortical populations encode identity of conspecifics

We first established an approach to study the neuronal encoding of conspecifics based on their volatile body odors. To examine the responses to such sensory cues, an experimental configuration is needed that allows for the repeated presentation of fresh body odors of different individuals during the same session and under controlled conditions. Towards this aim, emitter mice are placed into sealed isobaric containers with continuous airflow coupled to an olfactometer. The olfactometer is set to present repeatedly social and nonsocial odors for 1 s in a pseudorandomized order every 10–12 s to the male, adult receiver mouse (C57BL/6J background, see methods for details on recording cohorts) in head-fixed configuration (Fig. 1a). We presented the odors of two male C57BL/6 mice (#1: age: P(ostnatal day) 35 to P50; #2: age: P84 to P105) and one male CD1 mouse (age: P84 to P105) and also of peanut butter and a flower for comparison. The emitter mice were unfamiliar to the receiver mice. We focused here on the neuronal population responses in the olfactory cortices that form the three main olfactory connectivity hubs along the rostro-caudal axis[32]. Specifically, we recorded single-units with custom-built chronic tetrode arrays in the AON, the pPC and the LEC ($n = 8$ mice for AON, $n = 11$ mice for simultaneous recordings of pPC and LEC; Supplementary Figs. 1, 2a). Upon an initial adaptation shared by all social odors, odors elicited stable neural responses throughout the recording session (Fig. 1b, Supplementary Fig. 2b).

The process of recognizing a conspecific entails two aspects. Firstly, individual animals need to be discriminated by their unique odor signatures. Secondly, familiarity with the animal's odor is recalled. The recalled familiarity is thus a response feature shared among these animals and should reflect in a common signature between them. Other types of features shared across individuals can be intrinsic to the

emitter sources like different mouse strains. We can study how such features impact the processing of the odor by observing what is common across emitter mice with and without that feature. The first experiment served to establish the social odor representation approach and describe the difference between stimulus identity and feature encoding (Fig. 1c).

Firstly, we probed if the identities of unfamiliar individuals can be discriminated from the activation pattern in the neuronal population response. Indeed, odor signatures emitted by individuals were encoded by diverging activation patterns in cortical neurons (Fig. 1d, Supplementary Figs. 2c, 3) as also previously observed for identity encoding of non-social odors in the piriform cortex[33,34]. Consistently, a linear classifier reliably predicted the identity of the different odors from the neuronal activation pattern (Fig. 1e; two-sided Fisher's exact test against classifier trained with shuffled labels; AON: $p = 3e-31$, pPC: $p = 5e-26$, LEC: $p = 4e-29$).

Secondly, the encoding of features like salience or familiarity differentiating two groups of otherwise very similar odor emitters should reflect in the amplitude of the population response. The amplitude of the population response can be quantified using the Euclidean distance from the baseline population vector (Fig. 1c). Consequently, unfamiliar conspecifics that share also other features like strain and sex should not differ in the amplitude of the population response. We probed this on the population vectors of the single-unit spike counts concatenated across emitters. Indeed, two unknown C57BL/6 odors produced responses with a similar Euclidean distance from baseline (Fig. 1f, g). This finding applied to all three olfactory cortices (Fig. 1f, g, Supplementary Fig. 2e, f) as well as to two subcortical regions, the MOB and the VTA (see Supplementary Fig. 4). However, the population response amplitude differed between some of the odors that originate from food, flower or mice and are expected to differ in their features (Fig. 1f, Supplementary Fig. 2d–h).

We also tested two physiological proxies for salience, namely the pupil responses and sniffing measured with a pressure sensor in the odor port mask (Fig. 1h, Supplementary Fig. 5a, e). While all odors led to an increase of the sniffing frequency upon stimulus onset, the increase was more sustained in response to social odors (Supplementary Fig. 5a–d). Consistently, the three social odors also elicited comparable pupil dilations (Supplementary Fig. 5e–g). In summary, repeated presentations of volatile body odors emitted from a distant source elicit reliable responses. The physiological and neuronal population responses differ between broader classes of odor objects. Importantly, the Euclidean distance from baseline of the examined regions and the physiological readouts pooled across individuals do not differentiate under our conditions among unfamiliar male mice of the same C57BL/6J genetic inbred strain independent of age variations ranging from adolescent to adult. The social odor assay is thus suited to study familiarity memory in the neuronal population code.

### Population response amplitude encodes familiarity

We next tested how SRM is encoded in the olfactory cortices in a social exploration-recognition paradigm. Freely moving mice first explored a same sex adolescent ('familiar') for five minutes (Fig. 2a). Approximately ten minutes after the exploration, mice were transferred to the head-fixed recording set-up. During the recognition phase, odors from the previous interaction partner and from a novel age-matched mouse were presented for 1 s in a pseudorandomized order (Fig. 2a). The combinations of emitter and receiver mice were permuted across recording sessions, so that each emitter-receiver combination occurred only once. Each emitter mouse contributed both as 'familiar' and 'novel' to different receiver mice in balanced numbers of session (for number of mice see Supplementary Table 2). Receiver mice participated in at maximum one session per week. We pooled data across emitters. In the recognition phase, the average sniff frequency

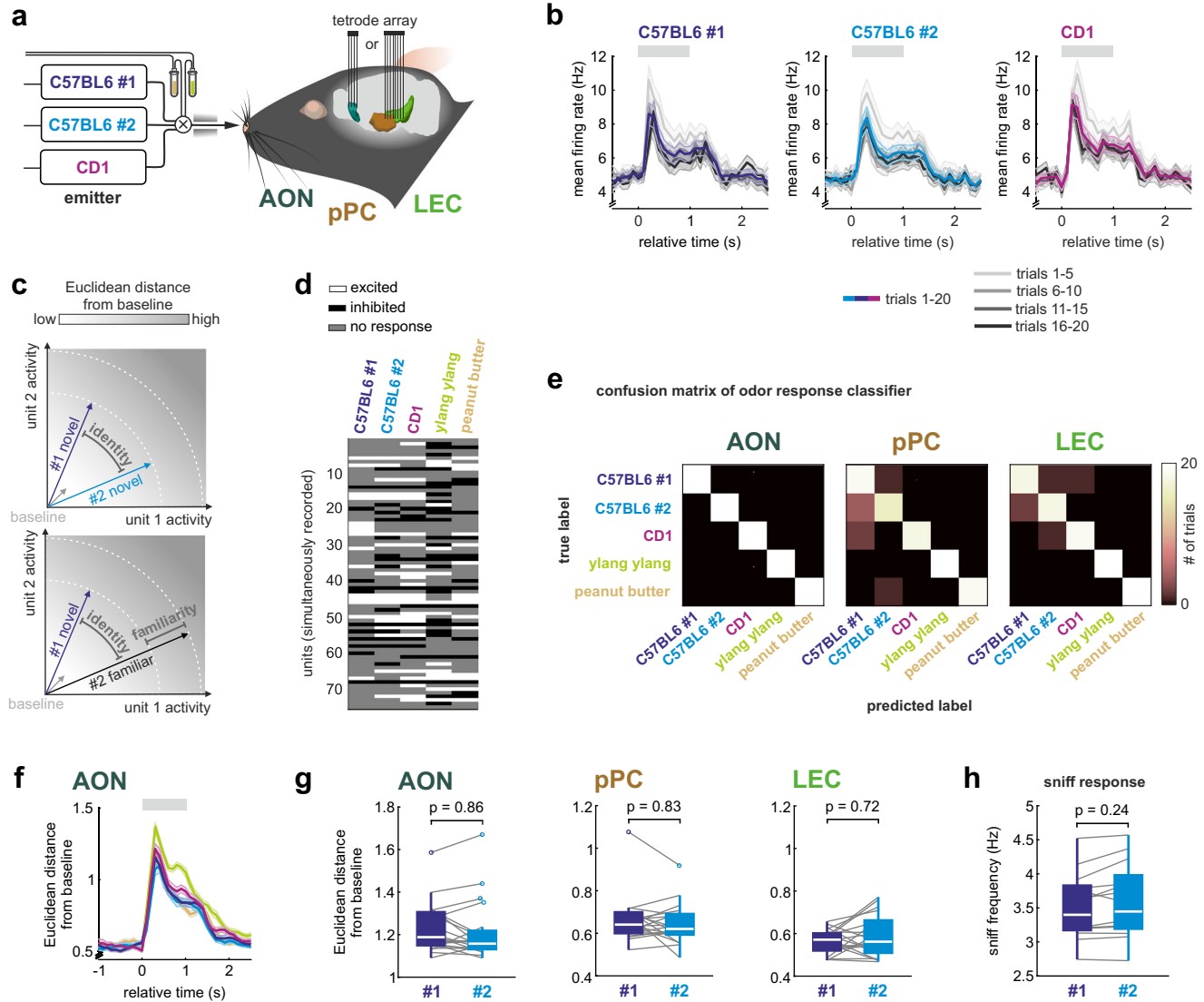

**Fig. 1 | Cortical populations encode identity of conspecifics. a** Odors from emitter mice or a natural floral odor (0.1% ylang ylang), and peanut butter were presented to head-fixed receiver mice in pseudorandomized trials. Each emitter mouse, namely male, unfamiliar C57BL/6#1 and C57BL/6#2 as well as a male CD1, was placed into a sealed isobaric container with continuous air flow regulated by the olfactometer. The combination of individual emitter mice was permuted to avoid repetition of unique combinations of emitter mice. Each session contained 20 trials per odor with a 1 s stimulus presentation and jittered trial durations of 10–12 s. **b** Stability of the firing rate response to social odors is shown for the AON in sequential blocks of 5 trials (grayscale, mean ± SEM, number of trials is indicated in the figure). All odors show initial adaptation in the first block. After the first five trials, responses were stable in amplitude and shape throughout the session (see Supplementary Fig. 2b for non-social odors). **c** The population vectors encode two components. Firstly, they encode the individual identity of an odor in their orientation, which stems from differential cortical activation patterns. Secondly, we hypothesize that they encode features like familiarity in their overall response amplitude, which can be quantified using the Euclidean distance from baseline. **d** Population responses in a representative experiment with 75 simultaneously recorded neurons from AON responding to the 5 different odorants. Single emitter individuals can be discriminated based on diverging responses in single-units. **e** The

confusion matrices of linear decoders, which were trained to predict the odor identity of a single trial from the neuronal population activity, shows high accuracy in AON, pPC and LEC. Prediction accuracy was determined on trials, that were not included in the training dataset. **f** The temporal evolution of the Euclidean distance from baseline of the population vector in the AON (mean ± SEM, $n = 20$ trials per odor). Gray bar represents odor duration. **g** The mean Euclidean distance from baseline was compared for the different odors (0 to +1 s relative to odor onset; repeated-measures one-way ANOVA with a post-hoc two-sided Tukey's test for multiple comparisons). None of the recorded cortices showed significant differences between the two unfamiliar mice from the same genetic background (see Supplementary Fig. 2d, g, h for all pairwise comparison results). **h** The sniff frequency response also did not differ between the two C57BL/6 mice (repeated-measures one-way ANOVA with post-hoc two-sided Tukey's test for multiple comparisons; $n = 13$ animals with 1 session each; see Supplementary Fig. 5c for all pairwise comparison results). In the figure, test results are indicated as exact p-values or as a heatmap (see also Supplementary Table 4 for details on test statistics). Boxplots with a horizontal line as median, the box edges indicating the 25th to 75th percentiles, a vertical line extending to the most extreme data points excluding outliers, and outliers plotted individually as circles. Source data are provided as a Source Data file.

response was higher for the smell of the familiar mice as compared to the novel ones (Fig. 2b, Supplementary Fig. 6a) and the familiar smell elicited a stronger pupil dilation (Supplementary Fig. 6b, c). Together, these first findings support the notion that social odors suffice to retrieve SRM.

Contrary to the hypothesis favoring habituation, the smell of the familiar animal elicits stronger salience responses than the novel one. We thus tested whether we would find a neural correlate of SRM. Indeed, we found a larger deflection from baseline in response to the odor of the familiar than a novel conspecific (Fig. 2c and see

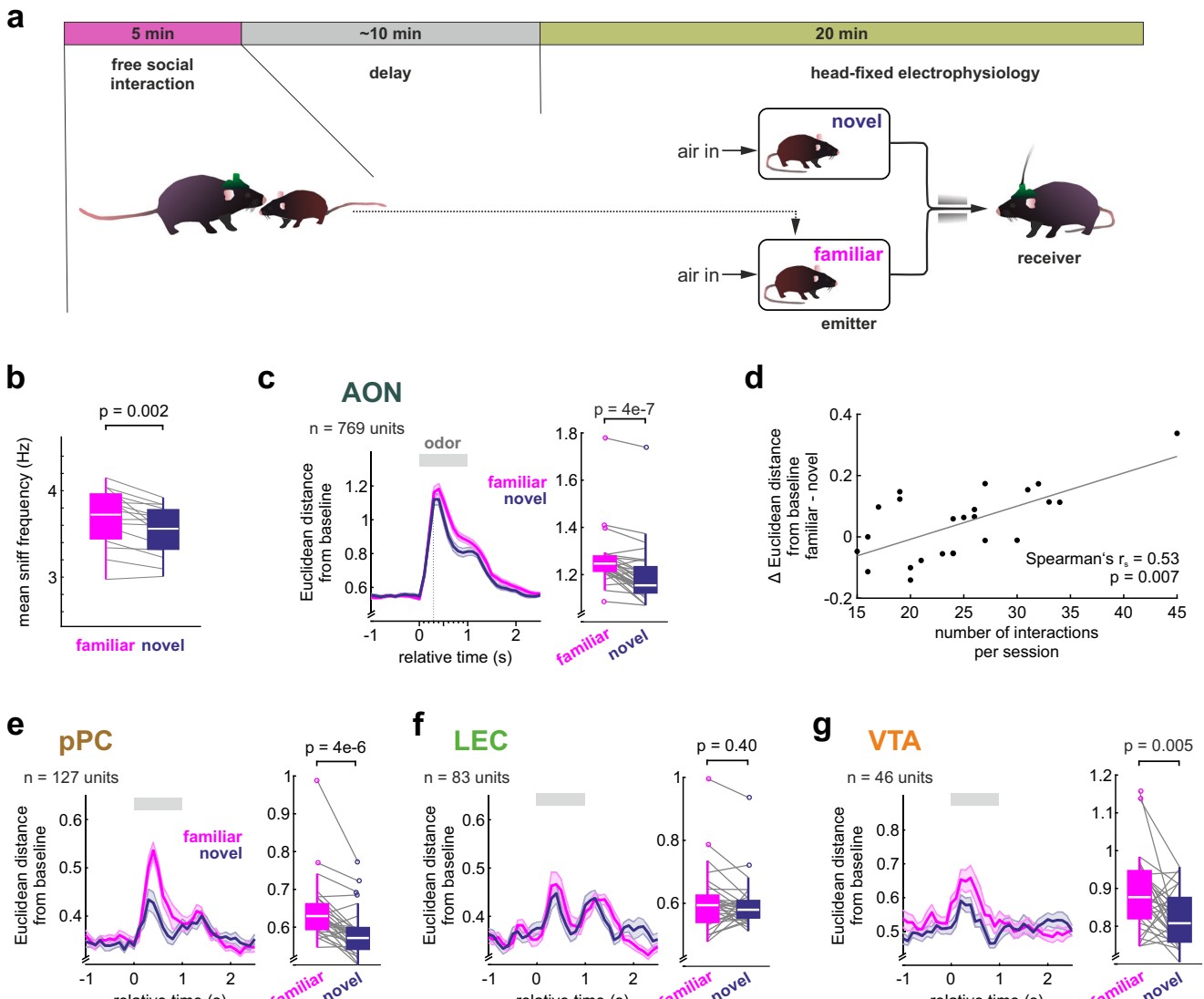

**Fig. 2 | Neural coding of familiarity. a** Mice were left to freely interact with a same-sex adolescent conspecific for 5 min. The head-fixed recording session started 10–15 min after the interaction, and the odors of the previous interaction partner (familiar) and that of a novel adolescent conspecific were presented for 1 s in pseudorandomized order (30 trials for each emitter). **b** The trial-averaged sniff frequency responses to the smell of the familiar and novel animal show higher sniff frequencies in response to the familiar one ($n = 15$ mice with 1 session each; two-sided paired t-test for session averages). **c** The (left) temporal evolution of the Euclidean distance from baseline of the population vectors from the AON (mean ± SEM, $n = 30$ trials per odor; number of units indicated in the figure) and (right) the mean Euclidean distance from baseline (0 to +1 s relative to odor onset)

compared for the responses to familiar and novel emitters (two-sided paired t-test) indicate stronger population responses to the familiar odor. **d** The correlation of the memory strength (difference in Euclidean distance from baseline between familiar and novel) and the number of interaction bouts during the freely-moving familiarization period shows a positive association ($n = 8$ animals, 3 sessions each; same data as **c**). Same as **c** for population responses in (**e**) pPC, (**f**) LEC and (**g**) VTA. In the figure, test results are indicated as exact *p*-values (see also Supplementary Table 4 for details on test statistics). Boxplots with a horizontal line as median, the box edges indicating the 25th to 75th percentiles, a vertical line extending to the most extreme data points excluding outliers, and outliers plotted individually as circles. Source data are provided as a Source Data file.

Supplementary Figs. 7, 8). The expressed neuronal memory (familiar – novel) positively correlated with the number of sampling events during the exploration phase (Fig. 2d), but not the sheer total contact duration (Supplementary Fig. 9a). This may hint on that social memory formation is promoted by repeated sampling. The stronger neuronal response in the AON to the familiar odor was however not explained by the sniff rate modulation in a trial-by-trial correlation (Supplementary Fig. 9b, c). We tested the prediction that the stronger the memory, the more it reflects in both session-wise average sniff and neuronal responses. Indeed, the difference in mean firing rate response or Euclidean distance from baseline correlated positively with simultaneously recorded sniff response differences between familiar and novel emitters (Supplementary Fig. 9d, e). AON units excited by both

conspecifics had stronger mean responses than selective units and also showed stronger firing rate responses to the familiar odor as compared to the smell of the novel mouse (Supplementary Fig. 8e). Among the other recorded cortical regions, the pPC, but not the LEC population response amplitude differed between the two odors (Fig. 2e, f). These results were independent of the order of cross-session trial-matching (as evidenced by a permutation test, when we repeated the analysis 300 times with random permutations of the trial order of the units composing the population vectors for each odor, $p < 0.05$ in 100% of permutations AON and pPC, $p < 0.05$ in 0% of permutations in LEC, see methods for details). In summary, the contribution of AON and pPC, but not LEC, to SRM encoding reveals a functional differentiation among the three cortices. VTA dopamine neurons reflect

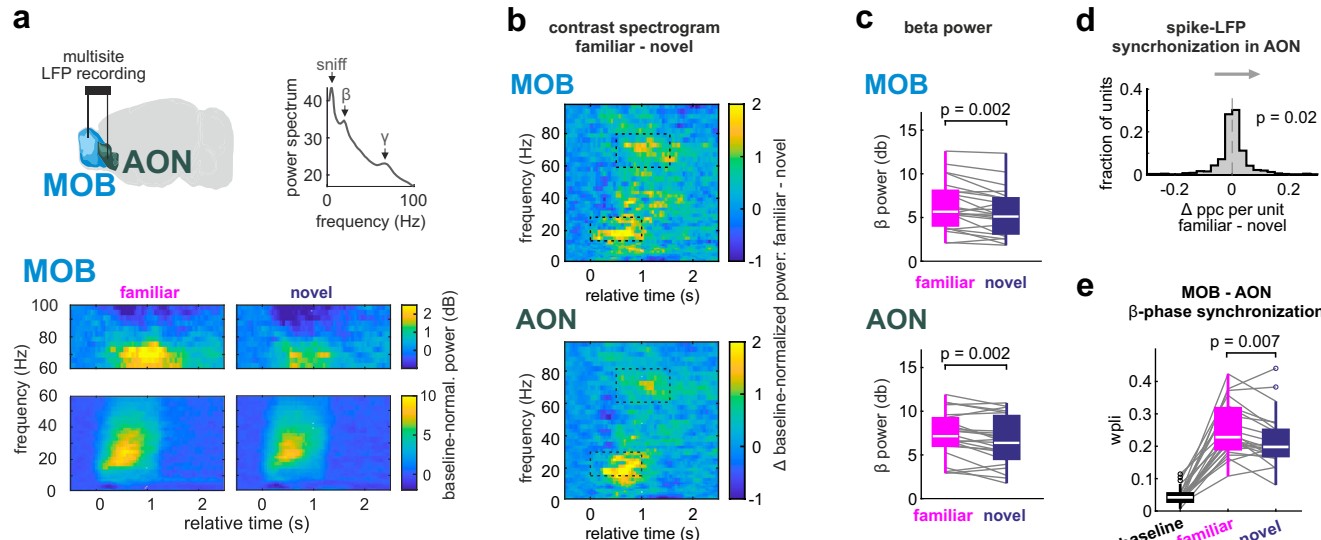

**Fig. 3 | Corticobulbar communication during retrieval of familiarity. a** The LFP was recorded simultaneously in the MOB and AON during head-fixed presentation of familiar and novel social odor stimuli (top, left). The spectrogram of the oscillation power during odor presentations showed peaks in the β and γ bands (15–30 Hz and 60–80 Hz, respectively) and sniff-locked frequencies (2–4 Hz) (top right). Odor-specific power spectrograms were normalized to baseline and averaged across sessions (*n* = 23 sessions from 8 mice) and show an increase in oscillatory power in the β and γ bands upon odor presentation from the familiar or novel mice (bottom). **b** Contrast spectrograms were computed by subtracting the session-averaged spectrogram of the response to the novel from the familiar mouse for MOB and AON, respectively. In both regions the familiar odor evokes stronger oscillations in the β and γ bands than the novel one. **c** Within-session comparison of time- and frequency-averaged β power increase (from baseline) confirms a stronger oscillatory response for the familiar than the novel smell in both regions (β band time-frequency window: 15–30 Hz, 0 to +1 s relative to odor onset; *n* = 23 sessions, two-sided paired t-test). **d** Single-units from the AON show more consistent phase-synchronization to the local β oscillations during presentation of the familiar animal (*n* = 750 units, two-sided Wilcoxon signed-rank test for difference between familiar and novel in averaged pairwise phase consistency (ppc)[74] in the β band). **e** Inter-regional phase synchronization of β oscillations between the MOB and AON increases for both odors as compared to baseline and is stronger for familiar than novel body smells (weighted phase lag index (wpli)[73] per session compared with two-sided Wilcoxon rank sum test, *n* = 23 sessions from 8 mice). In the figure, test results are indicated as exact p-values (see also Supplementary Table 4 for details on test statistics). Boxplots with a horizontal line as median, the box edges indicating the 25th to 75th percentiles, a vertical line extending to the most extreme data points excluding outliers, and outliers plotted individually as circles. Source data are provided as a Source Data file.

stimulus salience[35]. We therefore predicted to find reinforced responses also in the VTA. Indeed, putative dopamine neurons had stronger population responses to the familiar smell (Fig. 2g). Again, the familiar animal caused a stronger response in odor-excited units (Supplementary Fig. 8c). Taken together, these results highlight the reinforced and more distinct nature of the representation of familiar mice in the olfactory cortex and also in the VTA.

**Corticobulbar communication during retrieval of familiarity**
We then tested whether the SRM would be communicated from cortex to other regions. The AON sends rich top-down projections to the MOB and can modulate bulbar processing[24,36,37]. We examined how the AON interacts with the MOB to modulate its activity during odor-cue triggered memory retrieval. We recorded local field potentials (LFP) simultaneously in the MOB and AON of 8 mice (Fig. 3a). The social odor stimuli elicited increases in β and γ band oscillations (Fig. 3a and Supplementary Fig. 10a). In both regions, β and γ oscillations increased more to the familiar than to the novel smell (Fig. 3b, c, Supplementary Fig. 10b). The response to the familiar animal came also with enhanced phase-synchronization of single-unit spiking to the β oscillation in a separate cohort of 8 mice with single-unit recordings in the AON (Fig. 3d).

Beta oscillations are a network phenomenon involving the olfactory cortex and have been linked to olfactory learning processes[38]. Our findings could thus indicate a stronger top-down functional interaction with the MOB during recognition of the familiar conspecifics. Indeed, the phase-synchronization between MOB and AON in the β band increased with respect to baseline during the presentation of both odors, and again more to the familiar than to the novel smell (Fig. 3e).

In contrast, inter-regional phase-synchronization in the γ band did not significantly differentiate between familiar and novel animals (Supplementary Fig. 10c); consistent with the idea that olfactory γ oscillations are generated locally[38]. To directly test the hypothesis that the SRM is transmitted from the AON to the MOB, we performed fiber photometry recordings of top-down projections. GCaMP7f was injected into the right AON of 10 mice and a fiber optic was implanted into the granule cell layer of the ipsilateral MOB (Fig. 4a). GCaMP7f-expressing axons mostly terminated in the granule cell layer (Fig. 4b). The social odor stimuli increased top-down projection activity with stronger responses to the familiar smell (Fig. 4c, d).

The activity of putative mitral cells in the MOB recorded with chronic tetrode arrays (Fig. 4e, Supplementary Fig. 11a, b; *n* = 8 mice) was modulated by sniffing and its coupled LFP (Supplementary Fig. 11c, d). The excited and inhibited social odor responses produced balanced activity in the MOB (Supplementary Fig. 11e, f). However, the familiar odor elicited a stronger deflection from baseline than the novel smell (Fig. 4f, g, Supplementary Fig. 11g, permutation test: *p* < 0.05 in 100% of permutations). While the odor input initially flows from the MOB to AON, cortical back-projection activity during odor recognition may communicate SRM signals later in the odor response. Consistently, the peak power in the β band was reached only approximately half a second after odor onset, and later in the familiar than the novel (Supplementary Fig. 10d, e). Consistent with the late increase in top-down β activity, the AON units peaked in their firing response before the MOB (Supplementary Fig. 11h–j), providing correlational support of top-down modulation. In summary, these observations support that SRM information is transmitted top-down from AON to the MOB.

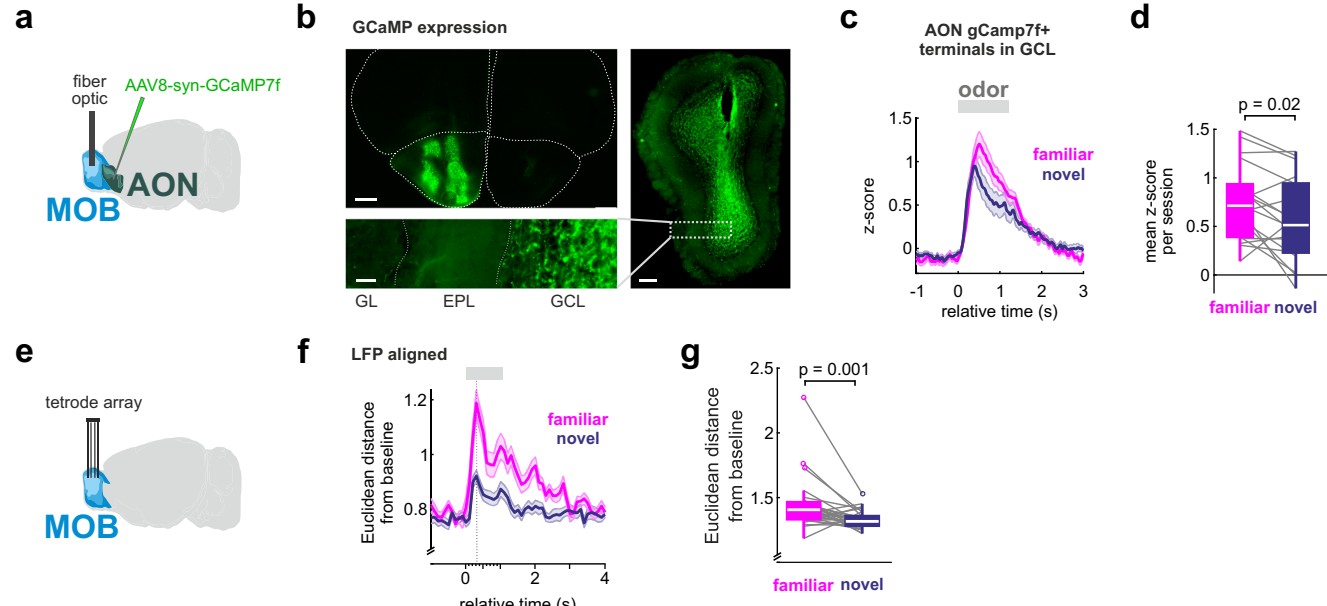

**Fig. 4 | Corticobulbar top-down projections transmit familiarity information.**
**a** Fiber photometry recording of axonal top-down projections from AON to MOB
were performed by injection of AAV8-syn-GCaMP7f into the right AON and fiber
placement in the ipsilateral granule cell layer (GCL) of the MOB. **b** Top left: Example
of GCaMP expression in AON. Right: Example of fiber position in the GCL. Bottom
left: Top-down projections preferentially target the GCL (scale bars: top left –
500 μm, right – 200 μm, bottom left – 50 μm) (n = 8 mice). **c** Average traces of
responses to familiar and novel odors in the MOB (mean ± SEM, n = 18 sessions).
**d** Trial-averaged responses show higher top-down projection activity for the
familiar animal (two-sided paired t-test, n = 18 sessions). **e** MOB single-units were

recorded using chronic tetrode arrays (n = 8 mice). **f** The temporal evolution of the
Euclidean distance from baseline (mean ± SEM, n = 30 trials per odor). **g** The mean
Euclidean distance from baseline (0 to +1 s relative to odor onset) was compared for
the responses to familiar and novel emitters; indicating a significantly stronger
population response to the familiar odor (two-sided paired t-test, n = 30 trials per
odor). In the figure, test results are indicated as exact p-values (see also Supple-
mentary Table 4 for details on test statistics). Boxplots with a horizontal line as
median, the box edges indicating the 25th to 75th percentiles, a vertical line
extending to the most extreme data points excluding outliers, and outliers plotted
individually as circles. Source data are provided as a Source Data file.

## Oxytocin neurons recruit limbic networks

The behavioral expression of SRM requires OXT during initial
exploration[24] but not during recognition[10]. One necessary prediction
is that the formation of cortical SRM traces is OXT-dependent. One of
the main sources of OXT release to the forebrain are axonal projec-
tions from the paraventricular nucleus (PVN)[39] of the hypothalamus.
As a first step, we aimed to identify brain networks that are func-
tionally activated by OXT neurons of the PVN. To explore this,
we performed fMRI in awake mice to simultaneously capture both
cortical and subcortical activations by optogenetically evoked
OXT release. We expressed the excitatory opsin ChR2 selectively in
OXT neurons of the PVN by injection of AAV5-FLEX-ChR2:mCherry in
OXT-Cre mice (ChR2^OXT/PVN mice; Fig. 5a, b and Supplementary
Fig. 12a).

A cohort of 23 ChR2^OXT/PVN mice for awake fMRI was habituated to
head-fixation and MR noise in mock scanners. We then examined a
network of candidate brain regions involved in social odor processing
(Supplementary Fig. 12b, c). During fMRI, intermittent optical burst
stimulation was applied to OXT neurons in the PVN (4 trials of blue
laser stimulation with 5 ms pulses at 30 Hz for 2 s with an inter-trial
interval of 5 minutes; Fig. 5c, Supplementary Fig. 12d, e). Optic burst
stimulation evoked a peak BOLD activation locally in the hypothalamus
(T = 6.75). OXT neuron stimulation recruited two clusters of candidate
regions in the network (Fig. 5c; Supplementary Table 3). The posterior
activated cluster comprised mainly parts of the hippocampal forma-
tion in addition to the aforementioned hypothalamus. The anterior
recruited limbic cluster contained the AON, the septal area, and medial
parts of the ventral striatum. Outside of the hypothalamus, the AON
showed the strongest BOLD activation upon OXT neuron excitation
(T = 5.18; Fig. 5c), which was also the only primary olfactory region
significantly activated. It should be noted that some regions that also

express OXT receptors like the piriform cortex[19] or the amygdala[39,40],
might not be captured for instance due to more complex BOLD acti-
vation patterns or preferential receptor expression in other cell-types.
Yet, the goal was to identify the regions with the most prominent
activation by OXT. We confirmed that the evoked OXT release acti-
vated the AON also with single-unit recordings in awake ChR2^OXT/PVN
mice (n = 8 mice, Fig. 6a, b, Supplementary Fig. 12f). Indeed, optoge-
netic burst-activation positively modulated the firing activity of AON
neurons in a transient manner in awake mice (Fig. 6a, b). Such mod-
ulation was not observed when a longer wavelength (593 nm) of light
was applied to PVN that does not activate ChR2, or when blue light
transmission into the brain was blocked between the patch cord and
the implanted fiber optic (Fig. 6b). OXT neurons also project to locus
coeruleus[41]. Consistently, optogenetic burst activation elicited tran-
siently pupil dilations in ChR2^OXT/PVN mice (Fig. 6c, d). This effect was
not observed in control conditions (longer wavelength stimulation or
blocked transmission condition; Fig. 6c, d). This attentional effect may
support the OXT-induced increase in signal-to-noise ratio and
enhanced olfactory sampling during initial exploration[24]. In summary,
OXT release recruits a distributed network with prominent activation
of the AON, the region outside of the hypothalamus where we
observed the strongest activation and the only primary olfactory
region with significant activation.

## Oxytocin enables the formation of cortical memory traces of familiarity

We thus tested how OXT modifies the encoding of SRM in the AON.
During the freely-moving exploration phase, additional OXT release
was evoked ('OXT' condition with 2 s burst stimulation of 5 ms pulses
at 30 Hz every 30 s in PVN) (Fig. 7a). When OXT release had been
boosted during initial exploration, we observed an increased

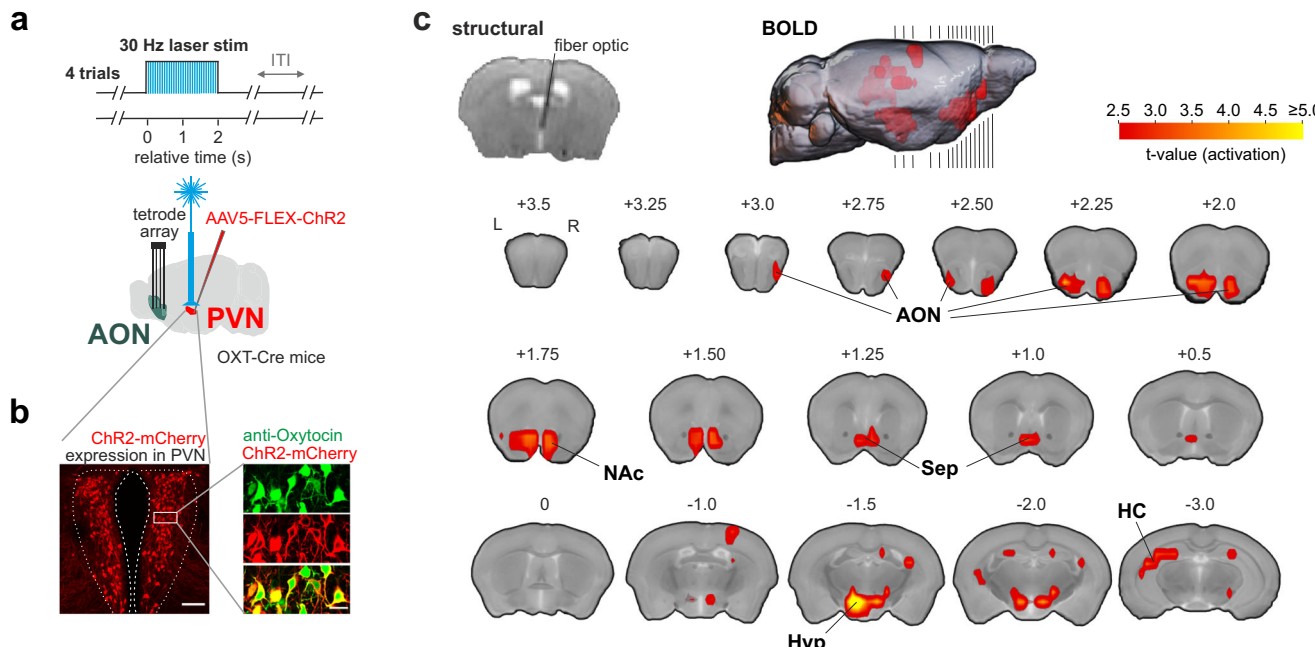

**Fig. 5 | Oxytocin modulates network activity in awake mice. a** Optogenetically triggered OXT release from the paraventricular nucleus (PVN) of the hypothalamus. Cre-dependent AAV expressing ChR2:mCherry was injected bilaterally in the PVN of OXT-Cre mice (ChR2$^{OXT/PVN}$ mice). OXT release was triggered four times by 30 Hz blue laser stimulation for 2 s (473 nm, 5 ms pulse duration, 5 min inter-trial interval). **b** Expression of ChR2:mCherry (red) is selective to the PVN with signal co-localization in neurons immuno-reactive to anti-oxytocin antibody (green) (coronal section, scale bar = 120 μm for overview and 20 μm for co-localization) (*n* = 6 mice).

**c** Functional MRI was performed in 23 mice with 1 session per mouse with opto-genetic OXT release. An exemplary T2-weighted structural image shows positioning of the fiber optic dorsal to the PVN (top left). Group-level t-statistic maps show prominent BOLD responses to OXT stimulation in the hypothalamus (Hyp), AON, NAc, septal area (Sep) and the hippocampus (HC) (two-sided t-test and Family wise error cluster-correction (FWE$_c$) with a cluster-defining threshold of pCDT <0.01 and $p_{FWEc}$ < 0.05, see also Supplementary Table 3 for details on test statistics). Black lines along the sagittal plane indicate the positions of coronals shown below.

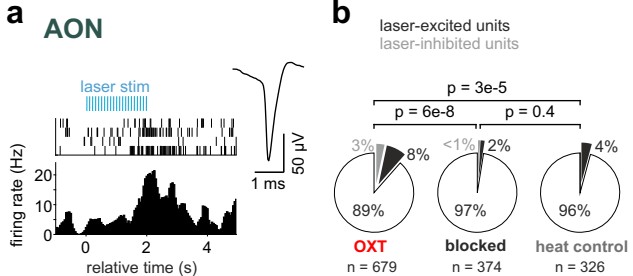

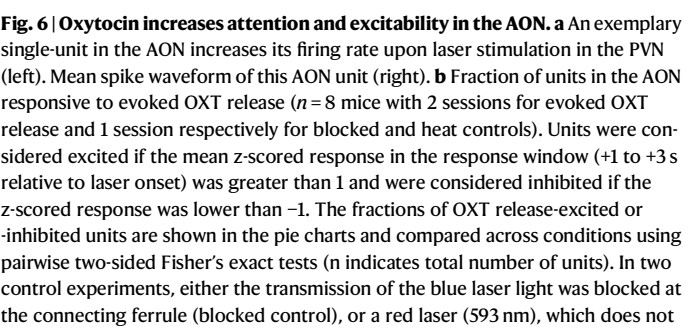

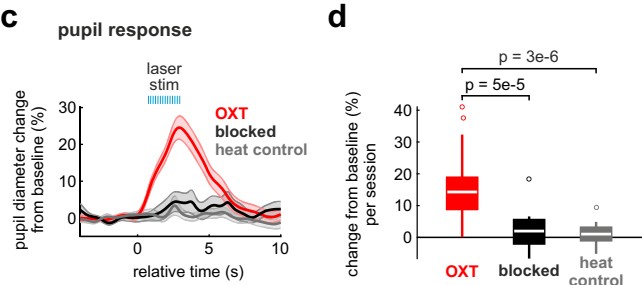

**Fig. 6 | Oxytocin increases attention and excitability in the AON. a** An exemplary single-unit in the AON increases its firing rate upon laser stimulation in the PVN (left). Mean spike waveform of this AON unit (right). **b** Fraction of units in the AON responsive to evoked OXT release (*n* = 8 mice with 2 sessions for evoked OXT release and 1 session respectively for blocked and heat controls). Units were considered excited if the mean z-scored response in the response window (+1 to +3 s relative to laser onset) was greater than 1 and were considered inhibited if the z-scored response was lower than −1. The fractions of OXT release-excited or -inhibited units are shown in the pie charts and compared across conditions using pairwise two-sided Fisher's exact tests (n indicates total number of units). In two control experiments, either the transmission of the blue laser light was blocked at the connecting ferrule (blocked control), or a red laser (593 nm), which does not

activate ChR2, was used at the same light power (heat control). **c** Temporal evolution of the pupil response to evoked OXT release and control conditions (average % change to baseline ±SEM, *n* = 28, 12, 14 sessions for OXT, blocked and heat conditions, respectively). **d** Average change in pupil diameter for the time window from 1 to 5 s after laser onset (*n* = 28, 12, 14 sessions for OXT, blocked and heat conditions, respectively; one-way ANOVA with two-sided Tukey's honest significance test for multiple comparisons). In the figure, test results are indicated as exact p-values (see also Supplementary Table 4 for details on test statistics). Boxplots with a horizontal line as median, the box edges indicating the 25th to 75th percentiles, a vertical line extending to the most extreme data points excluding outliers, and outliers plotted individually as circles. Source data are provided as a Source Data file.

difference in the deflection from baseline between familiar and novel animals during recognition, compared to the control condition without optogenetic boost (Fig. 7b, c, control group data from Fig. 2; permutation test for OXT condition: *p* < 0.05 in 100% of permutations; and see Supplementary Fig. 13a–d). Again, particularly the large fraction of units significantly excited by both conspecifics showed stronger firing rate responses to the familiar odor (Supplementary

Fig. 13e–h). Also, the sniff frequency was higher in response to the familiar odor than for the novel odor (Supplementary Fig. 13i).

If OXT indeed enables the formation of SRM, the differentiation of familiar and novel conspecific responses should be lost in mice with OXT receptor (OXTR) deletion in the AON. OXTR were deleted in the AON by injection of AAV-Cre into adult OXTR$_{fl/fl}$ mice (OXTR$^{ΔAON}$ mice; Fig. 7d, Supplementary Fig. 14a, b) which has been shown to reliably

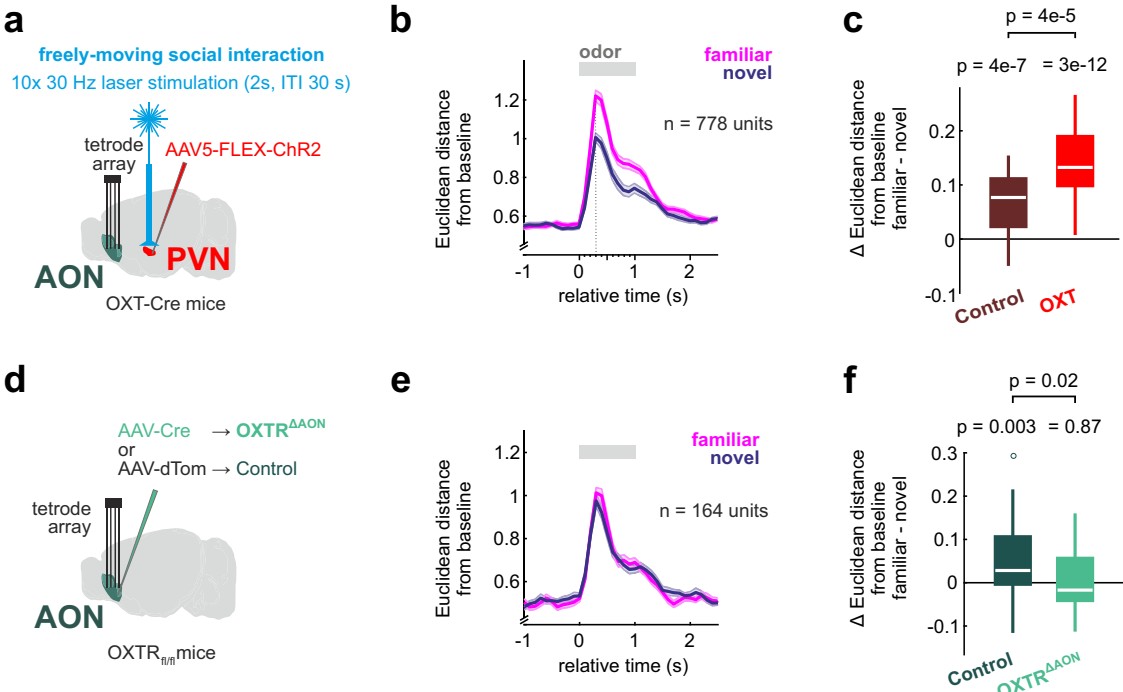

**Fig. 7 | Oxytocin enables the formation of cortical memory traces of familiarity.**
**a** Optogenetic OXT release was triggered 10 times (60 pulses at 30 Hz with 5 ms pulse length, every 30 s) during the interaction in the OXT condition. The control condition received no additional optogenetic OXT release (AON data from Fig. 2). **b** The temporal evolution of the Euclidean distance from baseline shows higher values in response to the familiar than the novel odor (mean ± SEM, $n = 30$ trials per odor). **c** The Euclidean distance from baseline was compared for the responses to familiar and novel emitters (0 to +1 s relative to odor onset); indicating a significantly stronger population response to the familiar odor (two-sided paired t-test) in each condition. Control and OXT conditions were compared using a two-sided two-sample t-test and showed a bigger difference between familiar and novel animals after boosted OXT release ($n = 30$ trials per odor). **d** Conditional OXT receptor knockout mice (OXTR$^{\Delta\Delta AON}$) were generated by injecting AAV-Cre into the AON of OXTR$_{fl/fl}$ mice (AAV-dTom was injected in OXTR$_{fl/fl}$ as control group) ($n = 6$

mice in each group). **e** The temporal evolution of the Euclidean distance from baseline shows no difference between responses to the familiar and novel social odors in OXTR$^{\Delta\Delta AON}$ mice (mean ± SEM, $n = 30$ trials per odor). **f** The Euclidean distance from baseline was compared for the responses to familiar and novel emitters (0 to +1 s relative to odor onset), indicating a significantly stronger population response to the familiar odor (two-sided paired t-test) in the control group but not the OXTR$^{\Delta\Delta AON}$ group. Control and OXTR$^{\Delta\Delta AON}$ groups were compared using a two-sided two-sample t-test and show a bigger difference between familiar and novel in the control group ($n = 30$ trials per odor). In the figure, test results are indicated as exact *p*-values (see also Supplementary Table 4). Boxplots with a horizontal line as median, the box edges indicating the 25th to 75th percentiles, a vertical line extending to the most extreme data points excluding outliers, and outliers plotted individually as circles. Source data are provided as a Source Data file.

delete the OXTR and prevent the behavioral expression of social recognition without impairing non-social odor discrimination[24]. Consistently, the familiar or novel odor did not differentiate in their sniff response in OXTR$^{\Delta\Delta AON}$ mice (Supplementary Fig. 14c). The baseline sniff or firing rates in the AON were similar in OXTR$^{\Delta\Delta AON}$ and matched control mice (Supplementary Fig. 14d–f). Importantly, in the AON of OXTR$^{\Delta\Delta AON}$ mice, there was no difference in the deflection from baseline between the responses to the familiar and novel odors (Fig. 7e, f; permutation test: $p < 0.05$ in 0% of permutations and see Supplementary Fig. 15). Note, that the identity of the odor could still be reliably predicted from the neuronal activation using a linear decoder, supporting the notion that odor discrimination was not affected (Supplementary Fig. 15h, Fisher's exact test against classifier trained with shuffled labels: $p < 0.0001$)[24]. To directly compare the manipulations of the OXT system and to account for differences in the number of recorded single-units, we used subsampling and analyzed the results with mixed-effects models (see methods). The joint analysis confirmed the bidirectionality of the OXT dependence on the induction of the SRM (Fig. 8a).

Finally, we tested whether the OXT-enabled memory also better separated the cortical representation of the familiar mice from other animals. Indeed, boosted OXT release during exploration increased the cross-odor distance between familiar and novel animals during recognition (Fig. 8b and see Supplementary Fig. 13j). These results

were independent of the order of cross-session trial-matching as evidenced by a permutation test ($p < 0.05$ in 72% of permutations for the test on cross-odor distance). Consistently, the Pearson cross-odor-correlation between the population vectors of the familiar and novel smell was smaller in the boosted OXT than in the control condition (Supplementary Fig. 13k). In contrast, the cross-odor distance between the familiar and novel smell during the recognition phase was smaller for the OXTR$^{\Delta\Delta AON}$ group compared to matched controls (Fig. 8c, Supplementary Fig. 15i; permutation test: $p < 0.05$ in 100% of permutations). Consistently, the Pearson cross-odor-correlation of responses to familiar and novel animals was more correlated in the OXTR$^{\Delta\Delta AON}$ than in the control group (Supplementary Fig. 15j). In summary, SRM is encoded in the olfactory cortices and associated regions by reinforced and more distinct representations (Fig. 8d). The AON propagates the SRM information top-down to the MOB. The formation of the SRM traces in the AON depends on OXT.

## Discussion

Different sensory stimuli like odors sampled in close proximity are used for social exploration[16,17]. It has however been debated whether also distant olfactory cues are sufficient for recognition[42,43]. The present data support that mice can use volatile body odors alone to decode the identity and SRM following brief encounters. Olfactory cues are thus in a position to serve broadly in social recognition.

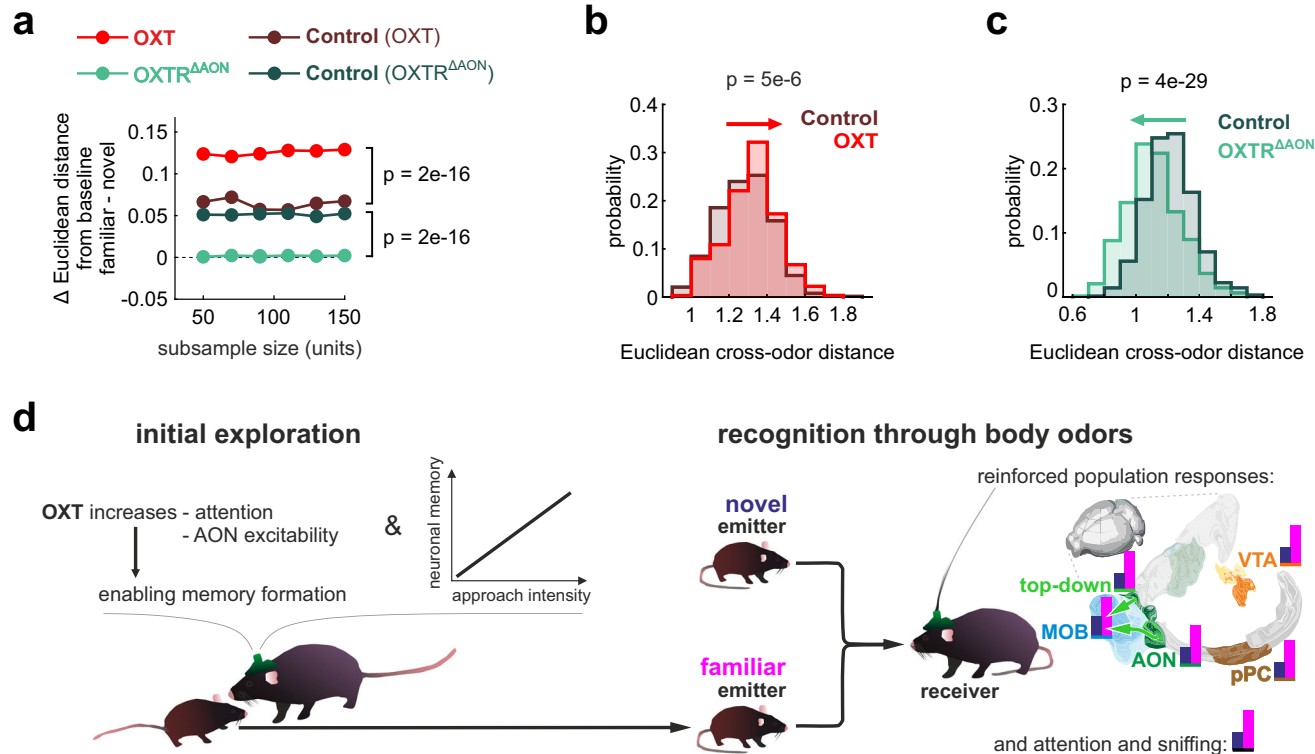

**Fig. 8 | Oxytocin enables reinforced and more distinct memory representations. a** To directly compare the effect of the OXT interventions and to account for differences in number of recorded units, we performed a subsampling analysis. Increasing numbers of units were repeatedly ($n = 500$ iterations) and randomly drawn from the pool of recorded units. The mean difference in Euclidean distance from baseline across draws was plotted as a function of the sample size. A linear mixed-effects model (see methods for details) confirms the bidirectional effect of the OXT manipulations (p-values from pairwise comparisons of the groups using a two-sided Tukey's test with Bonferroni correction for multiple comparisons indicated on the right). **b** The Euclidean cross-odor distance (familiar vs. novel) was computed for all possible combinations of trials (first 5 trials removed because of habituation, see Fig. 1b), and the resulting distributions in the control and OXT condition were compared ($n = 625$ trial combinations). The boosted OXT condition led to more differentiated population responses (two-sided Wilcoxon rank-sum test; see also Supplementary Fig. 13j). **c** OXTR$^{\Delta AON}$ mice had a smaller Euclidean cross-odor distance, indicating less differentiated population responses ($n = 625$ trial combinations, two-sided Wilcoxon rank-sum test; see also Supplementary Fig. 15i). **d** Summary schematic highlighting how OXT increases attention and AON excitability to enable recognition of body odors through reinforced population responses and sniffing. In the figure, test results are indicated as exact *p*-values (see also Supplementary Table 4 for details on test statistics). Boxplots with a horizontal line as median, the box edges indicating the 25th to 75th percentiles, a vertical line extending to the most extreme data points excluding outliers, and outliers plotted individually as circles. Source data are provided as a Source Data file.

Answering this question necessitated the development of an experimental configuration to present fresh body odors from different emitter animals under controlled conditions. The configuration produces robust, repeatable responses to volatile body odors at the level of neuronal coding and also with regards to the sniff and pupil responses, the latter physiological responses function as surrogate for salience[35,44–48]. Thus, this general approach can be applied for a comprehensive study on social cognition in mice to unravel the neural mechanisms of how an individual perceives others based on prior experience and as a function of the state of the receiver mouse. We focused here on SRM.

Social recognition requires the identification of an individual and the retrieval of the recognition memory. Information about the identity of individuals is provided by their olfactory signatures, which are encoded by divergent responses in individual neurons. The pooling of individuals isolates shared features across animals like familiarity. Consistently, cognitive surrogates and neuronal population responses pooled across subjects did not discriminate between two unknown mice of the same strain, and only diverged when one of the emitters was familiar. In line, such discrimination was lost in the cortical encoding of OXTR$^{\Delta AON}$ mice. Thus, SRM can be separated from the coding of individual identity. And, volatile body odors are sufficient for social recognition.

We had tested two scenarios for how SRM is implemented in the neuronal responses. From a behavioral perspective, familiar animals are approached less, and this might be reflected at the neural level by adaptation, i.e., attenuation of the response to the familiar one[29,30]. Alternatively, repeated encounters induce learning that produces more distinct representations of the animal. Our results show that the latter scenario applies in the olfactory cortices. Scaling of responses has also been observed in primates for salience coding of visual objects[49]. At first glance, it may seem counterintuitive that reinforced representations are associated with decreased exploration of familiar animals. However, it is important to keep in mind that the recognition process is completed before it triggers behavioral responses such as decisions to approach one animal more than the other. Adaptation of neural responses would fit the lower probability to explore the familiar animal in dyadic interactions, but would not be optimal for building a map of social interaction partners because familiar mice should obtain more distinctive cortical representations to better differentiate individuals. Thus, the preferential approach may reflect the need of sampling the new animal to build a representation of it. The corticobulbar system is involved in social perception, and social recognition memory by reinforced AON population responses to the familiar animal's smell.

The top-down projections from AON to MOB also show stronger responses to familiar animals. One consequence of reinforced AON output is to drive top-down MOB interneurons that then modulate mitral cells[24,36,37,50,51]. Olfactory sensory neurons first excite mitral cells in the MOB that then output to cortices like the AON. Interestingly, the

peak of the responses triggered by the social odors was reached only a few hundred milliseconds after the onset of the odor, and later in the MOB than in the AON. This could hint at top-down modulation affecting the development of the bulbar response. Also, the increase in β oscillations peaked several hundreds of milliseconds after odor onset. Learning is predicted to enhance cortical β that is conveyed top-down to the MOB[38,52]. Consistently, top-down projection activity and β-phase-synchronization between AON and MOB was stronger when smelling the familiar mouse. The top-down systems of the anterior olfactory cortices operate in awake animals with a state-dependent, relatively high baseline activity that transiently changes in response to odor stimuli (this study; refs. [24,53–55]). Within the range of normal activity states of awake mice, the cortical plasticity described here, as well as acute top-down modulation by oxytocin[24], rather strengthen both MOB peak responses and background inhibition. Taken together, these data support that corticobulbar information flow supports more discrete representations of familiar animals.

Thus, the corticobulbar system is involved in the entire sequence of social odor learning and its memory retrieval. During initial exploration, the release of OXT increases social attention, as indicated by acute pupillary dilation; this is most likely mediated by OXT projections to the noradrenergic locus coeruleus[41]. This attentional effect, together with the OXT-induced increased excitability of the AON-centered network and the top-down enhanced signal-to-noise ratio in the MOB put the brain in a state for processing social stimuli. Boosting OXT release during initial exploration further enhances plasticity, allowing the representation of the familiar odor to be better separated from that of other animals, which promotes the formation of a unique representation of the familiar emitter mouse. Conversely, such SRM traces were not expressed in the AON of animals with region-specific knockout of OXT receptors. This positions the AON as a key region among circuits necessary for social perception, learning, and subsequent recognition[9,11,13].

The smell of the familiar animal elicited stronger sniff and pupil responses. In general, for learned responses, increased sniffing and pupil dilation reflect also the assigned salience of that stimulus[35,56–58]. If this contributes to the salience response to the familiar individual, familiarity should also be reflected in the cortical and subcortical regions that code for value. In fact, two regions that are involved in the value encoding of odor stimuli responded stronger to the familiar animal, namely the pPC[59] and putative dopamine neurons in the VTA[35,60]. This complements previous work showing that OXT receptor activation in the piriform cortex entrains association learning of non-social odors to social cues[19] and highlights the various contributions of piriform cortex also to social memories. Future studies may dismantle the respective contributions of different olfactory cortices to retrieved social recognition memories and how the cortices interact with each other in these processes. Here, the connectivity of each cortex with other brain regions and their local computations of the same stimulus will have to be clarified. However, the enhanced familiarity representations were not ubiquitously expressed in the olfactory cortices; the LEC recorded simultaneously with the pPC, showed no such enhanced responses. This is consistent with the involvement of the LEC in environment associations, which are not relevant to animals in this paradigm. Thus, reinforced social representations are expressed in two of the three recently identified olfactory anatomical connectivity hubs[32], the AON and the pPC.

Taken together, social recognition memory is encoded in a distributed olfactory network of selected anterior and posterior olfactory regions. OXT induces learning of reinforced and distinct neural representations of distant smells of familiar individuals. This parallels the recognition processes of familiar faces in humans[61], which is impaired in autism spectrum disorders[62]. The ability to study these cognitions in mice may help fill a translational gap and fully understand the underlying neural circuit mechanisms. In general, the

reinforced cortical responses of familiar mice render them more distinct from others, which becomes relevant when a large number of animals need to be stored. Dyadic recognition memory thus provides the atomic structure of more complex social systems. Such SRM, enabled by reinforcing network plasticity, can be viewed as a building block for establishing social relationships between familiar animals, predicting the behavior of others, and the planning of interactions.

## Methods

### Animals and husbandry
In the experiments, the following transgenic mice were used (at least 12 weeks old at the beginning of the experiment): 29 male heterozygous OXT-Cre mice (B6;129S-$Oxt^{tm1.1(cre)Dolsn}$/J, RRID:IMSR_JAX:024234, Jackson Laboratory[63]) divided into 6 animals for histological confirmation and quantification of viral expression and 23 animals for neuroimaging, of which 16 proceeded to the in-vivo electrophysiology cohort, with 8 animals for recordings from the AON and 8 animals for recordings of multisite LFP and single-units from the MOB. Twelve male homozygous OXTR$_{fl/fl}$ mice (B6.129(SJL)-$Oxtr^{tm1.1Wsy}$/J, RRID: IMSR_JAX:008471, obtained from W.S. Young, NIMH[64]) were randomly assigned to the Cre injection or control group at a ratio of 1:1. All transgenic mice were bred in-house and maintained in a C57BL/6J (Charles River Laboratories) background (>F10). 3 male mice for in-vivo recording from VTA. 21 male C57BL/6J mice were obtained from Charles River Laboratories for in-vivo recording from LEC and pPC ($n = 11$) and for fiber photometry of top-down projections from the AON to the MOB ($n = 10$).

We used a pool of emitter mice (Supplementary Table 1 and 2). Emitter mice were either male C57BL/6J mice or male CD1 mice from Charles River Laboratories. Note that different recording condition groups were run with partially overlapping emitter animals. As an additional constraint, any one emitter mouse served only once per receiver mouse. The number of receiver and emitter mice in each experiment is given in Supplementary Tables 1 and 2.

All emitter mice were single housed in fresh cages for at least 24 h before the experiment to prevent cross-contamination of odors from other cage mates. Animals were single housed following surgical procedures, supplied with ad-libitum access to food and water for the complete duration of the experiments and kept on a 12-h light-dark-cycle (room temperature 24 °C, air humidity 55%).

All procedures were approved by the local animal welfare authority (Regierungspräsidium Karlsruhe) and in accordance with the EU Directive 2010/63.

### Virus preparation and stereotactic surgery
For optogenetic activation of OXT neurons in the PVN, $rAAV_5$-DIO-hChR2(H134R)-mCherry (20297-AAV5, addgene) was used. For conditional knockout of the OXT receptor in the AON, recombinant (r)AAV vectors were produced with AAV$_{1/2}$ coat proteins and purified with heparin columns to a final virus concentration of ~$10^{16}$ genome copies/ml: $rAAV_{1/2}$-CBA-Cre and $rAAV_{1/2}$-CBA-dTomato for Cre depletion and control groups, respectively[65]. For stereotactic injections, pre- and post-surgery analgesia (meloxicam, Metacam Boehringer Ingelheim) was administered. Mice were anesthetized with isoflurane and kept on a heating pad throughout surgery (Stoelting Rodent). The head was fixed and leveled using a stereotaxic instrument (Kopf Instruments). Lidocaine was administered topically. Using a glass micropipette connected to a nanoinjector (MO-10, Narishige), virus particles were injected (0.5 µl per site, 2 sites per hemisphere). Injection coordinates for the PVN in relation to bregma were (in mm): 0.1/0.3 posterior, 1.0 lateral, 4.8 ventral, with an angle of 10° relative to the vertical axis. AON injection coordinates for AAV-Cre or AAV-syn-GCaMP7f (104488-AAV8, addgene) injections in relation to bregma were (in mm): 3.0 anterior, 0.7/1.2 lateral, 3.5 ventral.

## Immunohistochemical staining

To quantify the expression of ChR2:mCherry in the PVN, 6 animals were examined for immunohistology 3 weeks after virus injection. Animals were anesthetized with ketamine/xylazine (300 mg/kg BW ketamine and 60 mg/kg BW xylazine diluted in 0,9% saline) and perfused transcardially with phosphate buffered saline 0.9% (Dulbecco's PBS), followed by 4% paraformaldehyde (PFA, ROTI®Histofix 4%). The extracted brain was post-fixed in 4% PFA overnight at 4 °C. Serial coronal 50 μm sections were prepared with a vibratome (Microm HM 650 V, Thermo Scientific). Free-floating sections were stained with anti-OXT (1:1000, mouse; kindly provided by Harold Gainer)[66] at 4 °C overnight. The signals were visualized with anti-mouse ALEXA 488 secondary antibody (goat IgG, 1:1000, ThermoFisher Invitrogen, cat. n. A-11001). Images were acquired with a Leica TCS SP5 confocal laser scanning microscope (20× or 63× oil immersion objectives). Cre-expression for generation of OXTR[ΔΔAON] animals was validated by incubating free-floating sections with anti-Cre (1:1000, rabbit; Novagen, cat. n. 69050-3) at 4 °C overnight. GCaMP7f-expression was validated by incubating free-floating sections with anti-GFP (1:1000, chicken, Abcam, cat. n. 13970) at 4 °C overnight. The signals were visualized with anti-rabbit or anti-chicken ALEXA 488 secondary antibody (goat IgG, 1:1000, ThermoFisher Invitrogen, cat. n. A-11008 or A-11039, respectively). Images were acquired with an Olympus VS200 slide scanner.

## Odor delivery

For odorant delivery, a custom-built air-dilution olfactometer was used[35,67]. Non-social odorants were kept in liquid phase in dark vials and mixed into a nitrogen stream, before being further diluted 1:10 into a constant air stream leading to a final valve which controls odor flow to the odor port where the experimental subject was placed for recording.

For the delivery of body odors, animals that served as odor emitters were placed in small hermetic plastic containers (dimensions: 14 (length) × 8 (width) × 7.5 (height) cm). The in- and out-flow of air were kept at a steady state. The airflow outlet of the containers was switched via olfactometer-controlled valves into the air flow of the olfactometer. The output resistances of each container were separately calibrated to a small overpressure (0.3–0.5 bar) to match the pressure difference between the container and the odor port. This prevented pressure changes in the social odor box upon switching the valves and allowed for alternating presentations of social and non-social odors within one session. The emitter animals inside the containers were monitored through a transparent top-cover throughout the session for signs of distress or discomfort. Also, the pressure of each container was constantly monitored. To avoid accumulation of the receiver animals' own or the presented odors in the recordings chamber, an active exhaust matching the flowrate of the continuous inflow from the olfactometer was applied. Valves (NResearch) were controlled by Arduino microcontrollers (Arduino Mega 2560) and the trial structure was programmed using MATLAB scripts connected to the microcontroller.

## Experimental paradigms: social odor processing

To investigate the responses to social odors, we conducted an experiment where the receiver male subject was placed in a head-fixed setup for recording of neural activity and physiological responses. We used 5 different odor sources: the body odor of an adolescent (#1; P35-50) and a young adult C57BL/6 mouse (#2; P84-105), and of a CD1 mouse (P84-105) as well as the natural flower odor ylang ylang (1% dilution in mineral oil, W311936, Sigma Aldrich) and mineral oil-dissolved peanut butter (Griff GmbH). The combination of individual emitter mice was randomly permuted to avoid repetition of unique combinations. A specific emitter was presented to each receiver mouse only once across sessions. The number of emitter animals in each

cohort is given in Supplementary Table 1. Each odor was presented for 20 trials in pseudo-randomized order with no more than three consecutive trials of the same odor. Trial numbers were kept sufficiently high per odor to allow for statistical testing considering trial-by-trial noise in cortical responses. The trials in a session were limited to a number at which the animals did not become drowsy. In each trial, one odor was presented to the animal for a duration of 1 s, followed by an inter-stimulus interval jittered evenly between 10 and 12 s.

## Experimental paradigms: social interaction and recognition

To probe the neural coding of familiarity, we modified a social recognition paradigm in mice[4] for head-fixed recordings. In this study, we either compared the response to two unfamiliar body odors as a control condition or, in a separate experiment, the response to a familiar and an unfamiliar mouse. This design was chosen to maintain the order of the social recognition test. We used a same-sex (male) adolescent interaction partner to avoid sexual and aggressive behaviors during the interaction. To balance their representation, one emitter mouse served in an experiment both as a 'novel' and as a 'familiar' for different receiver mice. The number of emitter animals in each cohort is given in Supplementary Table 2. For the interaction phase, the experimental subject was transferred to a fresh home-cage (70 lux ambient light). An unfamiliar adolescent C57BL/6 was introduced into the cage. The two mice were left to interact freely for 5 min. The interaction was recorded on video using a Sony FDR-X1000V camera with a frame rate of 30 fps. Exploration behavior was later annotated manually using the BORIS software[68]. Frames where the experimental subject's nose touched the conspecific were labeled as interaction. Two subsequent interaction bouts were counted as individual events if they were separated by more than one second.

For the recognition phase, the interaction partner was removed from the cage and transferred to a social odor container (familiar conspecific). A second, novel adolescent was placed in another social odor container. Then, the experimental subject was placed in the head-fixed recording setup. The time between interaction and start of the recording session was approximately 10 min. The body odors of the familiar and novel conspecifics were each presented 30 times for 1 s in a pseudorandomized order, with an inter-stimulus interval jittered between 10 and 12 s.

In the 'OXT' condition, optogenetic OXT release was evoked every 30 s (60 pulses with 5 ms duration at 30 Hz) selectively during the free interaction phase. In the matched control condition, laser transmission to the brain was blocked at the connection site between the ferrule and the patch cord.

## Experimental paradigms: optogenetic OXT release

To investigate the effects of optogenetic activation of OXT neurons, we performed a laser stimulation experiment without odor presentation. Animals expressing ChR2 under the control of the OXT promotor were placed in the recording setup and a 200 μm diameter multimode fiber optic patch cable (ThorLabs) was connected to a fiber optic placed directly dorsal of PVN (see 'Optical fiber and head bar implantation' for fabrication and surgical procedure for fiber implantation). Each session had 4 trials of optogenetic stimulation with an inter-trial interval of 5 min. The long inter-trial interval was used to promote the recovery of OXT release and postsynaptic receptors after burst stimulation. In each trial, the burst stimulation consisted of a train of 60 blue laser pulses at 30 Hz with a pulse duration of 5 ms (473 nm, Shanghai Laser & Optics). Laser power output at the tip of the optic fiber was calibrated to 5 mW measured on the detector of a PM100USB power meter (ThorLabs). For the experiments in Figs. 5, 6 and Supplementary Fig. 12, animals received 1% saline solution as drinking water for the 12–16 h before the beginning of the experiment to optimize OXT release[69].

To mask potential attentional responses to the blue light flashes above the head of the animal, ambient blue light had the same wavelength and the mating connector of the patch cable with the optic ferrule was shielded with black rubber tubing. We also performed control experiments (Figs. 5, 6, Supplementary Fig. 12) in which the laser light transmission was blocked at the mating connector, so that there was no blue light delivery into the brain, only the blue light pulse above the animal's head was still visible. Further, to control for possible light-induced artifacts during recordings, an additional control experiment using an orange laser (593 nm, Shanghai Laser & Optics) was performed with the same stimulation protocol.

## Electrophysiological recording arrays

Recording arrays were designed in-house and custom-built to allow for flexible targeting of one or multiple regions of interest[35]. Briefly, we designed for each experiment anatomically adapted printed circuit boards (Multi Leiterplatten GmbH). Circuit boards contained guiding holes for the tetrodes in the desired geometrical configuration. Tetrodes were spun from 12.5 μm Teflon-coated tungsten wire (California Fine Wire). Wires were then added through the guiding holes. The desired tetrode length was determined by a mounting scaffold. Then tetrodes were fixed with liquid acrylic adhesive. The single wires of the tetrodes were soldered to gold-coated through-holes that served as a break-out board to a soldered-on connector (model n. 5024307030-7030, Molex SlimStack) for connection to the Intan amplifier headstage via a custom-built adapter. Wires were coated in two-component epoxy for protection. Before implantation, the impedance of each tetrode channel was lowered to a target of 300 kΩ by plating with gold solution (HT1004, Sigma Aldrich) using a NanoZ-device (Multi Channel Systems).

For AON recordings, both hemispheres were targeted with 8 tetrodes each, arranged in two rows of 4 tetrodes. Inter-tetrode distance was 300 μm. The MOB was targeted with 8 movable tetrodes. The LFP from the AON was recorded simultaneously to the MOB with two single-wire 50 μm Teflon-coated tungsten wire (California Fine Wire) electrodes per hemisphere. For LEC and pPC, 14 tetrodes were arranged in two rows and angled at 20° tilt laterally to unilaterally target the right hemisphere. For VTA recordings, each hemisphere was targeted with 8 tetrodes; with 4 tetrodes glued to an optical fiber and the remaining 4 fibers were placed at a distance of approximately 200 μm from the fiber. Target coordinates (all in mm relative to bregma, depth relative to dorsal brain surface) are given for the center of the respective tetrode array: MOB: 4.0 anterior, 1.0 lateral, 0.4 ventral; AON: 3.0 anterior, 0.95 lateral, 3.5 ventral.; LEC/pPC: 2.2 posterior, 4.3 lateral, 4.0 ventral; VTA: 2.7 posterior 0.5 lateral, 4.5 ventral.

## Implantation of recording array

For implantations, mice were anesthetized with isoflurane and pre- and post-surgery analgesia (meloxicam, Metacam Boehringer Ingelheim) was administered. After anesthesia induction, mice were placed in a stereotactic instrument with non-traumatic ear bars (Kopf Instruments) and on a heat pad (Stoelting Rodent Warmer). Lidocaine was administered topically. The fur covering the dorsal skull was removed and the skin disinfected. The skin covering the dorsal cranium was resected, the resection margins fixed by tissue adhesive (3 M Vetbond), and the skull was prepared for craniotomy. Craniotomies were performed at the indicated coordinates. An additional hole was drilled in the posterior skull and the ground pin (Gold pin, NeuraLynx) placed into this hole. Then, the skull was coated with a layer of dental multi-component bond (C&B Superbond, Sun Medical), sparing the craniotomy holes. The implant was prepared, attached to a motorized 3-axis micromanipulator (Luigs & Neumann) and slowly guided into position. Tetrode lengths had been chosen so that the bottom side of the PCB would be placed parallel to and just above the skull surface when the tips reached the desired depth. Dental Cement (Paladur,

Kulzer) was then applied to fill the gap and form a mechanical connection between the implant and the Superbond-covered skull. Mice were monitored after surgery and recovered in their home cage for at least seven days before proceeding to habituation for experiments.

## Histological confirmation of recording site

For histological confirmation of recording sites and viral expression after experiment completion, animals were anesthetized with ketamine/xylazine (300 mg/kg BW ketamine and 60 mg/kg BW xylazine diluted in 0,9% saline), followed by transcardial perfusion with 0.9% PBS and then 4% paraformaldehyde for fixation. Further immersion fixation in 4% paraformaldehyde of the whole skull for two weeks allowed visualization of the tetrode tracts (Supplementary Figs. 1, 14a) and bilateral ChR2:mCherry expression in the PVN. Serial coronal sections were prepared using a vibratome with 200 μm slices for visualization of tetrode tracts and 50 μm slices for fluorescence images of the PVN sections. Images were acquired on a Axio Imager 2 (Zeiss). Tetrode position was mapped to an atlas template. Units from LEC/pPC recordings were assigned to their respective region based on tetrode position on the histological images (Supplementary Fig. 1b).

## Head-fixed recordings

Mice were initially habituated to the head-fixed setup for five days. Head-fixed electrophysiological recordings were performed inside a Faraday cage with blue ambient light (LED, 465 nm, approximately 70 lux). All recordings were performed with Intan 64 channel RHD 2164 miniature amplifier boards connected to a RHD2000 interface board (Intan Technologies). Trial times were recorded from TTL-outputs of the olfactometer to the interface board. Data were sampled at 30 kHz.

## Data processing and spike sorting of single-units

Raw data output from Intan recordings was converted to binary data files suitable for KiloSort[70] (https://github.com/MouseLand/Kilosort) using software provided by Intan Technologies (https://intantech.com/files/RHD_MATLAB_functions.zip) and custom-written scripts. Spike sorting was then performed using KiloSort2 and code from the mlib toolbox by Maik Stüttgen (https://de.mathworks.com/matlabcentral/fileexchange/37339-mlib-toolbox-for-analyzing-spike-data) and the spikes toolbox from the Cortical Processing Laboratory at UCL (https://github.com/cortex-lab/spikes). Subsequent visual curation of KiloSort output was done in Phy2 (https://github.com/cortex-lab/phy). Selection criteria were based on the putative unit's waveform and refractory period violations (<2% of all spikes within refractory period of 2 ms). Spike clusters were merged if the waveforms and PCA features of two putative units were highly overlapping and the cross-correlograms showed no coincident spiking; spike clusters were split if they had multiple waveforms and overlapping clusters in feature-space. We further analyzed AON units with a mean firing rate between 0.1 and 20 Hz and MOB, LEC and pPC units with a mean firing rate above 0.1 Hz. In the VTA we analyzed single-units with firing features of dopamine neurons (mean firing rate between 0.1 and 12 Hz)[60].

## Single-unit analysis

Single-unit responses to odors were first assessed based on their changes in spiking activity. Trials were aligned to the odor onset and peri-stimulus time histograms (PSTH) were computed with a bin-width of 100 ms. Responsivity to the different stimuli was tested by comparing the baseline firing rate (-4 to 0 s relative to odor onset) to the firing rate during odor presentation (0 to +1 s relative to odor onset) using a two-sided Wilcoxon signed rank test with Bonferroni correction for multiple comparisons (all units tested for each odor). Single-units with a significant increase in firing rate to at least one odor were defined as 'odor-excited' and units showing a significant decrease to at

least one odor were defined as 'odor-inhibited'. Units that showed both 'odor-excited' and 'odor-inhibited' responses to multiple odors were defined as 'mixed response'.

To normalize for the spontaneous firing activity of the units before averaging across odor-excited or odor-inhibited populations, we computed the z-scored response for every unit. First, responses were averaged across trials of one odor-type, FR, and then the z-score was computed for every unit at time-bin $t$ by: $z(t) = \frac{FR(t) - FR_{baseline}}{\sigma_{baseline}}$, where $FR_{baseline}$ and $\sigma_{baseline}$ are the mean firing rate and standard deviation during the baseline window.

Single-unit responses to the optogenetically evoked OXT release were also analyzed based on their change in z-scored response (baseline −10 to 0 s before first laser pulse of a trial). The response window was set to be from +1 to +3 s after laser onset. Units showing a mean z-scored response of more than +1 in the response window were defined as 'laser-excited' and units with a mean z-scored response below −1 were defined as 'laser-inhibited'. The fractions of laser-responsive units (either excited or inhibited) between the conditions (laser, blocked-laser control, and heat control) were tested using pairwise Fisher's exact tests.

## Population analysis: population vectors

To capture the evolution of the activity of the whole recorded population of single-units, we applied a series of analysis techniques based on population vectors. For $n$ simultaneously recorded units from session $s$, the population vector at time-bin $t$ is defined as $\mathbf{v}_t^s = [FR_t^{s,u_1}, \cdots, FR_t^{s,u_n}]$, where $FR_t^{s,u_1}$ contains the spike count of unit 1 at time-bin $t$. Population vectors were built using a bin-width of 100 ms. We were interested in revealing the information related to strain, age, and familiarity memory. This has to be differentiated from odor identity coding. To this aim, we pooled units across sessions and concatenated single-session vectors into a global population vector $\mathbf{V}_t = [\mathbf{v}_t^1, \cdots, \mathbf{v}_t^S]$. Trials were matched by odor type (for example to strain, age, or familiarity). By this procedure, any change observed between population vectors of different types can be ascribed to a coding difference of a specific odor feature (strain, age, or familiarity) and not to a difference in mouse identity.

## Population analysis: odor response classifier

We evaluated odor discriminability by measuring the prediction accuracy of a linear classifier. For each brain region separately, we trained multiclass models using support vector machines on the odor presentation period (0 to +1 s relative to odor onset) of the global spike count population vectors (concatenated across sessions and z-scored within unit). Prediction accuracy was determined using leave-one-out cross-validation, and thus by comparing to the true label the predicted label of a population response vector which was not included in the training dataset. Confusion matrices were plotted for visualization of the classifier performance. Odor discriminability was tested statistically by comparing against the classification accuracy of a classifier which was trained with shuffled labels using Fisher's exact test.

## Population analysis: Euclidean distance from baseline

To investigate and compare the temporal evolution of odor responses in the different recorded populations, we computed the deviation of the population vector from its baseline. At each time step $t$, the deviation from baseline was computed as Euclidean distance between the population vector $\mathbf{V}_t$ and the baseline vector $\mathbf{B}$, divided by the square root of the number of units in a population. $\mathbf{B}$ was obtained in the window −1 to 0 s before odor onset and calculated by averaging across all trials. Population vectors were built using a bin-width of 100 ms and, for visualization (but not for statistical testing), smoothened with a sliding window of length 3. Differences in the response to social and non-social odors were tested using a repeated-measures one-way ANOVA with post-hoc two-sided Tukey's test for multiple

comparisons on the mean distance during odor presentation. Differences between the familiar and novel odor were tested with a two-sided paired t-test. An increase or decrease in the familiarity effect due to interventions in the OXT system (that is, optogenetic OXT release versus control condition or OXTR$^{\Delta AON}$ versus control group) was tested with a two-sided two-sample t-test on the difference between familiar and novel. The plots displaying the temporal evolution of the deflection from baseline show distances computed with population vectors built by matching trials of the same type across sessions according to their order of occurrence. To exclude the possibility that the obtained results depend on the specific trial-matching applied when testing, we also performed a 'permutation control' by testing the difference in the Euclidean distance from baseline with population vectors that were constructed using random permutations of the trial order matching (preserving the odor-type label)[35]. The fraction of those 300 performed tests with a $p$-value < 0.05 is reported together with the respective main result.

## Correlation analysis: neural response vs. sniff response

To investigate whether the population response amplitude was mediated by the sniff frequency, we performed a trial-by-trial correlation of sniff frequencies versus the Euclidean distance from baseline. We included all AON sessions with simultaneous sniff recordings, as the per-session unit count was highest in AON among regions and sufficient to build population vectors for single sessions. For every session, we then computed the Euclidean distances from baseline from the single-session population response vectors and correlated the mean distance during odor presentation (0 to +1 s relative to odor onset) to the average sniff frequency of the trial (0 to +1 s relative to odor onset) using Spearman's correlation coefficient.

To investigate if the memory strength was associated with the averaged sniff response during the recognition, we computed Spearman's correlation coefficient for the single-session average difference in Euclidean distance from baseline between familiar and novel responses or the mean difference in firing rate response (0 to +1 s relative to odor onset) versus the difference in sniff frequency between familiar and novel responses. We included all AON sessions with simultaneous sniff recording.

## Correlation analysis: neural response vs. exploration behavior

To investigate if the memory strength was associated with the intensity of exploratory behavior during the familiarization period, we performed a correlation of the number of interaction bouts or the investigation time versus the difference in Euclidean distance from baseline between the familiar and novel odor. We included all AON sessions as the per unit session count was sufficient here to build single-session population vectors. For every session, we computed the mean difference in Euclidean distance from baseline between familiar and novel odor responses and correlated that to the number of interaction bouts or to the total interaction time using Spearman's correlation coefficient.

## Population analysis: Euclidean cross-odor distance

To assess the effect of interventions in the OXT system on social odor familiarity, we computed the Euclidean cross-odor distance of the mean population vectors during odor presentation (from 0 to +1 s after odor onset) for all combinations of trial pairs. Cross-odor distances were normalized to baseline by dividing the distance value for the mean response by the matching baseline distance value (computed for the population vectors from the time window from −1 to 0 s from odor onset). This allowed us to test if the distance between the odor responses became significantly larger or smaller upon manipulating the OXT system as compared to the trial-by-trial variability in the control (Supplementary Figs. 13j, 15i). Since we noticed a transient habituation period in the neuronal responses during the first trials of

the session (see Fig. 1b), we tested for the robustness of the result and repeated the analysis excluding the first five trials of the session, thus comparing only stable responses (Fig. 8b, c). Two-sided Wilcoxon rank-sum tests were conducted to compare the distributions of distances. Again, we performed a permutation test on the trial order as described for the Euclidean distance from baseline.

## Population analysis: cross-odor-correlation

To assess to what degree the different odors recruit overlapping sets of units, we computed the Pearson cross-odor-correlation between population vectors of trials from different odors throughout the trial progression. For every combination of trials and for each time step $t$, Pearson's correlation coefficients were computed between population vectors of different odors. Finally, we averaged over the resulting distribution of correlation coefficients. Higher values of cross-correlation indicate a greater shared subpopulation of units that is commonly recruited by the two odors. Two-sided Wilcoxon rank-sum tests were conducted to compare the distributions of mean correlation coefficients (0 to +1 s after odor onset) for boosted OXT versus control condition and OXTR$^{\Delta\Delta AON}$ versus matched control group.

## Population analysis: subsampling and mixed-effects model

To conduct a comprehensive comparison of various manipulations within the OXT system while accounting for variations in the number of recorded units, we performed a subsampling analysis. Multiple ($n = 500$ iterations) random unit subsets were drawn from the complete population of recorded units. For each iteration, the trial-averaged difference of the Euclidean distance from baseline between familiar and novel odor responses was computed. The subpopulation size was systematically increased, ranging from 50 to 150 units in increments of 20. Note that the maximum subsample size is still lower than the smallest recorded population (Control (OXT): $n = 769$, OXT: $n = 778$, Control (KO): $n = 275$, OXTR$^{\Delta\Delta AON}$: $n = 164$). We employed a linear mixed-effects model to predict the difference in Euclidean distance between familiar and novel responses for the four OXT conditions (Control (OXT), OXT, Control (KO), OXTR$^{\Delta\Delta AON}$). To account for the hierarchical structure of the data arising from repeated measurements (iterations) within each subsample size, we incorporated random effects in the models. Specifically, we considered random intercepts and slopes for the grouping factor representing the different number of units in the subpopulations. To determine the significance of the random effect in the full model, we compared it to a reduced model that excluded the number of subsampled units as a random effect. We employed the likelihood ratio test to assess the improvement in model fit due to the inclusion of random effects. There was no significant dependence of the difference in response amplitude on the number of sampled units (Likelihood Ratio Test: $p = 0.79$). Additionally, a two-sided Tukey's post-hoc test with Bonferroni correction for multiple comparisons was conducted to compare the different OXT conditions. Models and statistical tests were conducted in R using the 'lme4', 'lmerTest' and 'multcomp' packages.

## MOB spike-time realignment

M/T cells in the MOB show a sniff-dependent firing modulation[67,71]. Since we had recorded the sniff simultaneously only for 13 out of 49 M/T cells, we used the LFP from the MOB instead. The coherence between the sniff and averaged LFP oscillations showed a peak between 2 and 4 Hz (Supplementary Fig. 11c). The LFP was bandpass filtered between 2 and 4 Hz using a second-order Butterworth filter and the difference between the first zero crossing with a negative first derivate and the odor onset was used to shift the spike times of a given trial. This reproduced modulated firing rate patterns seen after sniff-alignment (Supplementary Fig. 11d). We again computed the trajectory and Euclidean distance from baseline for putative mitral (including middle

tufted) cells pre- and post-alignment and received similar results to the odor-aligned population (Supplementary Fig. 11e–g).

## Response latencies

The latency to peak response was computed for the Euclidean distance from baseline and for the z-scored responses of odor-excited units. For the latency to peak of z-scored responses, familiar and novel odors were analyzed separately and only units that showed a significant excitatory response for the respective odor were included. The distributions from AON and MOB were then compared using a two-sided Wilcoxon rank sum test.

The latency to firing onset was determined for odor-excited cell-odor pairs from MOB and AON using the z-scored response with a binning of 10 ms to capture the onset more precisely. The latency to response onset was defined as the time when units surpassed a z-score of 1.96 (which corresponds to a firing rate response outside the approximate 95% confidence interval of firing activity without stimulus presentation). The distribution of onset latencies across excited cell-odor pairs was visualized in a histogram with a binning of 50 ms.

## Data processing of local field potentials (LFPs)

LFP traces were extracted from the raw data and down sampled to 1 kHz. Data quality was assessed by visual curation of the recorded channels. When extracting the LFP from tetrode recordings, the median of the four channels of a tetrode was used. Subsequent LFP analysis was then performed using fieldtrip[72]. After importing the data in the fieldtrip format an integrated artifact rejection was performed to exclude data-segments with electrical artifacts (for example because of motion). The raw data was z-transformed and a cutoff z-value of 20 or higher qualified as artifact and the identified section was excluded from further analysis. On average $2.8 \pm 0.6$ (mean ± SEM) from a total of 60 trials were excluded per session.

## LFP spectral power

We analyzed oscillatory activity based on different physiological frequency bands (β, γ). Power spectra (the power of a frequency component of the signal during a discrete time-interval) of LFP channels were calculated for single trials by using a time-frequency transformation with hanning tapers based on multiplication in the frequency domain (multi-taper-method convolution). A sliding window with the size of 0.5 s was shifted along the time series at steps of 50 ms, resulting in a temporal resolution of 50 ms and a frequency resolution of 1 Hz in a frequency range between 1 and 100 Hz. Power spectra were then averaged across repeated trials of one condition (familiar or novel odor). For every session, channels were averaged to receive a mean power spectrum for every region and condition. Oscillatory power was normalized to the baseline ($-1$ to 0 s relative to odor onset) by computing the relative change, expressed in decibels: relative power(db) = $10 \cdot \log_{10}(\frac{P_{t,f}}{P_0})$, with $P_0$ denoting the average power of the frequency during baseline and $P_{t,f}$ denoting the spectral power for a given time-frequency bin. For visualization, time-frequency plots were generated by averaging across sessions. Contrast spectrograms (familiar vs novel) were computed by subtracting the power of the novel from the familiar odor response and then averaging across sessions.

Statistical tests were conducted based on average spectral power in the β or γ band per session. The β band was defined as ranging from 15 to 30 Hz and the time-window of interest was set to the duration of the odor presentation (0 to +1 s relative to odor onset). The γ band was defined as ranging from 60 to 80 Hz[38] and the time-window of interest was set to +0.5 to +1.5 s relative to odor onset.

## LFP inter-regional synchronization

We used the weighted phase lag index (wpli) to assess synchronization of the LFP between regions. This metric has been shown to be less sensitive to over-estimation of true synchronization due to volume

conductance[73]. The cross-spectra for all pairs of LFP channels from different regions were computed using the same settings as described above for power-spectra. The wpli was then computed for the two odors. For every session, wpli values were averaged in the β and γ time-frequency windows as defined above and we conducted a two-sided Wilcoxon signed rank test to compare the wpli between familiar and novel odor stimuli.

### Spike-field analysis

To investigate if the spike timing of single-units was differentially coupled to the LFP for the two odors, we computed the pairwise-phase consistency (ppc) for all single-units[74]. First, the spike-triggered LFP spectrum for every spike was estimated based on the averaged LFP from tetrodes in the same hemisphere (only tetrodes that also had units were considered) and the ppc was computed. We averaged ppc values over the duration of the odor presentation (0 to 1 s relative to odor onset) and tested the difference between familiar and novel in the β band using a two-sided Wilcoxon signed-rank test. For data from the AON, 19 out of the 769 units were excluded from this analysis, since no reliable ppc could be estimated because of their low firing rate (mean firing rate: 0.53 Hz).

### Fiber photometry of top-down projections

To test if the familiarity information in the AON is transmitted top-down to the MOB, we performed fiber photometry of axonal projections to the MOB. The surgery procedure for virus injection and fiber implantation was performed as described in 'Virus preparation and stereotactic surgery'. A total of 400 nl AAV8-syn-jGCaMP7f-WPRE (104488-AAV8; addgene) was injected into the right AON. Optic fibers (MFC_200/250-0.66_3mm_MF2.5_FLT; doric lenses) were implanted into the ipsilateral MOB (4.0 anterior, 0.9 lateral, 1.5 ventral). A layer of dental glue (C&B Superbond, Sun Medical) and an additional layer of dental cement (Paladur, Kulzer) were applied. Animals recovered at least 3 weeks before the start of imaging experiments.

Signals were recorded during head-fixed presentation of familiar and novel social stimuli as described in 'Social interaction and recognition' using the fiber photometry console (doric lenses) with a fluorescence mini-cube with integrated photodetectors (iFMC5; doric lenses) in lock-in amplifier mode. LED power was calibrated to 35 μW output power at fiber tip, measured on the detector of a PM100USB power meter (ThorLabs). Doric Neuroscience Studio V5 was used for data collection (doric lenses).

Raw signals were loaded into MATLAB, filtered using a low-pass filter (first-order Butterworth filter) with a cutoff frequency of 10 Hz, detrended (Matlab integrated function, 9$^{th}$-degree polynomial trend) and z-scored across the whole session. Signal traces were realigned to stimulus onset and averaged across trials of the same type and time (0 to +1 relative to odor onset). Session-averages of familiar and novel odor response amplitudes were compared with a two-sided paired t-test.

### Sniff recording

The sniff was recorded in a subset of electrophysiology sessions in Figs. 1–4, 7 and 8 using a custom-built snout mask that also served to deliver the odors (originally designed by D. Rinberg, NYU). The mask was gently pressed onto the nose of the mouse, generating a cavity in the mask in which pressure fluctuations resulting from the breathing cycle were continuously measured through a HDI pressure sensor (HDIM020GBY8H3; First Sensor Inc.). The pressure signal was recorded on the RHD2000 Interface Board also used for the tetrode recordings at a sampling rate of 30 kHz. Before every recording session the inflow and outflow of the olfactometer was calibrated to zero changes in flowrate during valve switching. The sniff signal and pressure levels in the olfactometer were monitored visually by the experimenter throughout the recording.

### Sniff analysis

The sniff signal was down-sampled to a sampling rate of 100 Hz and band-pass filtered (first-order Butterworth filter) between 2 and 30 Hz. The sniff frequency was then estimated from the power spectra of the signal using the FieldTrip toolbox[72]. Power spectra were calculated for each trial using a time-frequency transformation with Hanning tapers based on multiplication in the frequency domain (multi-taper-method convolution). A sliding window with the size of 0.5 s was moved along the time series with a step size of 50 ms, thus obtaining a temporal resolution of 50 ms and a frequency resolution of 0.1 Hz in a frequency range between 2 and 8 Hz. For every session, we averaged across trials of the same odor and determined the sniff frequency over time from the maxima of the power spectrum. We then compared the mean sniff frequency during odor presentation (0 to +1 s relative to odor onset) to the mean baseline sniff frequency (−1 to 0 s relative to odor onset) for each odor and session. The change from baseline was tested using a two-sided paired t-test. For data in Fig. 1h, the change from baseline was also compared across different odors using a repeated-measures one-way ANOVA with a two-sided post-hoc Tukey's test for multiple comparisons. Results from pairwise comparisons are illustrated in a p-value heatmap, with white squares indicating non-significant p-values. For data in Supplementary Fig. 14, the change from baseline was compared between groups (OXTR$^{ΔAON}$ vs. control) using a two-sided two-sample t-test.

### Pupil recording

The pupil was imaged during head-fixed recording sessions with a digital infrared camera (AD4113T-I2V Dino-Lite Pro2 digital microscope) at a sampling rate of 10 frames per second with a resolution of 1280 × 960 pixels using the DinoCapture 2.0 software. Ambient light intensity was calibrated so that the pupil was moderately dilated at the beginning of the session to obtain a large dynamic response range of pupil diameter changes. To align the videos to the trial information data (TTL triggers from laser or olfactometer), a red LED (650 nm) was activated for 1 s at the start of the session.

### Pupil analysis

A DeepLabCut network[75] was trained to predict the positions of eight pupil landmarks that were spaced evenly around the pupil. The pupil diameter was then calculated by taking the median of the Euclidean distances between opposing markers. Time-points where the DeepLabCut network returned no markers with a likelihood of 99% or higher (for example, when the animal was blinking and the diameter therefore could not be calculated), were interpolated using a spline-interpolation. A second-order Butterworth low-pass filter with a cutoff frequency of 0.75 Hz was applied to the pupil data. The mean diameter changes in relation to the baseline window (−1 to 0 s before odor onset in the case of odor trials, −4 to 0 s before first laser pulse for OXT release experiment in Fig. 6) were calculated for each experimental condition. Considering the temporal lag between pupil responses and stimulus onset, the response window was set to start at 1 s after odor onset. Statistical comparison of the pupil diameter change was performed on the mean over all trials in a session for the response window of 2 s for odors and 4 s for optogenetic OXT release.

### Optical fiber and head bar implantation for functional MRI

The surgery procedure for head bar implantation and stereotactic injection of the PVN was performed as described in 'Implantation of recording array' and 'Virus preparation and stereotactic surgery'. The optic fiber (FT-200-EMT with ceramic ferrule CFLC230-10, Thorlabs) for optogenetic activation of PVN-OXT neurons was implanted according to the following coordinates relative to bregma (in mm): 0.2 posterior, 0.8 lateral, 4.3 ventral, with an angle of 10° to the vertical axis. A layer of dental glue (C&B Superbond, Sun Medical)

 

was applied to the skull and around the optical fiber. Then, the head bar used to fix the mouse in the fMRI cradle was attached using another layer of dental glue, followed by an additional evenly dispersed layer of dental cement (Paladur, Kulzer). Placement of the fiber tip dorsal to the third ventricle was later confirmed by structural MRI. All mice recovered at least 7 days before proceeding to fMRI habituation.

## Habituation to functional MRI

The cohort of mice designated for neuroimaging ($n = 23$) underwent habituation training to reduce motion and elevation of stress levels during MRI that may arise from head-fixation and auditory noise from the scanner[57]. All subjects underwent a habituation protocol which we previously established for an odor-guided reward learning task where mice showed sufficient habituation as assessed by comparable behavioral performance during fMRI and training outside of the scanner[57]. Mice were first habituated to the head-fixation setup with gradually increasing session durations. Next, mice were placed in a mock-scanner setup and habituated to increasing volumes of recorded MRI pulse sequence noise. After about 7–10 sessions of habituation over the span of 1–2 weeks, mice were habituated.

## Functional MRI acquisition

Functional MRI data were acquired using a small-animal 9.4 Tesla Magnetic Resonance (MR) scanner (94/20 Bruker Biospec) with Avance III hardware, BGA12S gradient system with maximum strength of 705 mT/m and running ParaVision 6 software. Functional scans were acquired with a gradient-echo echo planar imaging (GE-EPI) sequence, with the following parameters: voxel dimensions: 0.3 ×0.3 ×0.6 mm; 1300 volume acquisitions; flip angle: 60°; TR/TE: 1200/18 ms; slice number: 20; matrix size: 64 ×64; field-of-view (FOV): 19.2 ×19.2 mm. GE-EPI was conducted during optogenetic OXT release. Image slice volumes were acquired in contiguous sections without interslice gap. The EPI session was followed by a high-resolution T2-weighted Rapid Imaging with Refocused Echoes (RARE) scan to image the native structural space (TR/TE: 1200/6.3 ms; matrix size: 96 ×113 ×48; voxel size: 0.2 ×0.2 ×0.3125 mm; FOV: 19.2 ×22.6 mm; RARE factor 16). A fieldmap was acquired before the EPI to correct for geometric distortion (TE1/TE2: 1.725/5.725 ms; TR: 20 ms; matrix size: 64 ×64 ×64; FOV: 20 ×20 ×20 mm).

## Image processing

Acquired data were converted from Bruker file format to NIfTI file format, resized by a factor of 10 for better visualization, and reoriented using pvconv.pl (http://pvconv.sourceforge.net/) and a custom in-house MATLAB (Version 2020, MathWorks) routine. The first five volumes of the functional time series were removed to exclude T1 effects, leaving the remaining image volumes for further preprocessing. These data were unwarped using the presubtracted phase and magnitude field map images and realigned with a 7th degree B-spline interpolation using SPM12 (Statistical Parametric Mapping; Wellcome Department of Imaging Neuroscience; https://www.fil.ion.ucl.ac.uk/spm/) to obtain 6 rigid-body transformation parameters. fMRI time-series data were slice-time corrected (to the mean slice), and then linearly aligned to the subject's native anatomical scan. Quality checks were performed manually by visually inspecting preprocessing outputs. The 3D-anatomical data were segmented into tissue classes and a group-template in the space of the stereotactic anatomical atlas[76] was created using Diffeomorphic Anatomical Registration Through Exponentiated Lie Algebra (DARTEL)[77]. The resulting nonlinear flow-fields were then applied to the functional images to transform them to atlas space. Spatially normalized results were visually examined for each case. The processed EPI data outputs were smoothed using a Gaussian kernel (with 0.6 mm full-width at half-maximum, fwhm).

## Functional MRI analysis

The acquired BOLD time-series data were modeled at the single-subject level by a univariate General Linear Model (GLM) using SPM12. Events were modeled at the onset of each laser stimulation convolved with a mouse-specific hemodynamic response function[78]. Convolved event responses and their temporal derivative were included in the GLM as well as the 6 realignment parameters obtained by rigid-body head motion correction. The data were high-pass filtered with a cutoff of 128 s, a first-degree auto-correlative model was included to correct for aliasing, and a masking threshold (as proportion of globals) of 0.2 was applied.

At the second-level, a parametric one sample t-test was conducted to test for brain wide BOLD correlates of laser-triggered OXT release at the multi-subject level. The resulting event-modeled BOLD activation and deactivation map to laser stimulation was then examined after Family Wise Error (FWE) cluster correction for multiple comparison (cluster-defining threshold (CDT), $p_{CDT} < 0.01$; $p_{FWEc} < 0.05$). For visualization purpose, the group T contrast maps were masked with a social odor network mask containing the selected regions of interest (see Supplementary Fig. 12c). This mask was created in the stereotactic space on basis of the Allen-Mouse brain atlas[76] with an in-house software[79]. All resulting BOLD significance maps were overlaid onto a high-resolution template brain and illustrated with MRIcroGL (https://github.com/rordenlab/MRIcroGL)[79].

To rule out that residual motion affected BOLD activation maps, we tested whether the pattern of BOLD responses to OXT neuron stimulation was preserved in a sub-sample of animals with low motion. For this purpose, we stratified the animals by their framewise displacement (FD), defined by the mean FD value of the frames during laser stimulation, and re-ran the second-level analysis only using the 11 animals with FD lower than the groups' median. The statistical threshold was set to $p_{CDT} < 0.05$ due to lower power and only clusters > 50 voxels were shown (Supplementary Fig. 12d).

## Data analysis and statistics

Electrophysiological and behavioral data were analyzed with MATLAB (Mathworks, R2021a/2023a). Repeated-measures one-way ANOVAs were computed in Prism 9 (GraphPad Software). Mixed-effects models were computed in R (https://www.r-project.org/, 4.0.3). Statistical tests: test statistics, sample size, and multiple comparison corrections were indicated for each performed test and reported either in the related method sections and in Supplementary Table 4. For functional MRI data, in SPM12, a general linear model (fixed effects) was used, where stimulus onset times were modeled as events (stick functions) convolved with a mouse-specific HRF[78]. Individual voxel-wise T-contrast maps were tested at the multi-subject level by one-sample t-test. The resulting statistical maps for BOLD activation/deactivation were each thresholded at a cluster-defining threshold of $p_{CDT} < 0.01$ and then Family Wise Error (FWE) cluster-corrected ($p_{FWEc} < 0.05$).

Graphical visualization: IoSR Matlab toolbox (https://github.com/IoSR-Surrey/MatlabToolbox) and mseb (https://www.mathworks.com/matlabcentral/fileexchange/47950-mseb-x-y-errbar-lineprops-transparent) were used for visualization. Whenever reporting averaged or collective results, the number of units, animals, and sessions used were reported directly next to the graph or in the respective figure legends; boxplots were centered on the median, solid polygons indicated the 25th and 75th percentiles and whiskers indicated the most extreme data points not including outliers. Outliers were indicated with circles; average values were always reported with SEM. In all figures, exact $p$-values were reported, and the significance threshold was set to $\alpha = 0.05$.

## Reporting summary

Further information on research design is available in the Nature Portfolio Reporting Summary linked to this article.

## Data availability

The electrophysiology and fMRI data generated in this study are under active use by the reporting laboratory; all data presented in this manuscript are available upon request from the Lead Contact. Processed AON single-unit data are available for download at https://doi.org/10.6084/m9.figshare.24637986.v1. Source data are provided with this paper.

## Code availability

Code for generating the figures from the deposited data is available on https://github.com/KelschLAB/OXT-Wolf[80].

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

## Acknowledgements

We thank Cathrin Löb, and Felix Hörner for excellent technical assistance, Marina Antonini and Paul Brandes for contributions to the recordings, Luise Staatsmann and Claire Julliot de la Morandière for help with the histology, Alan Kania and Walter Cañedo Riedel for comments on the manuscript, and Laurens Winkelmeier for support with fMRI and discussions. This work was supported by the BMBF-NSF CRCNS grant 'Oxystate' BMBF 01GQ1708 to W.K. and NSF 1724221 to C.L., Boehringer Ingelheim Foundation grant 'Complex Systems' to W.K., Leibniz Society SAW grant 'Learning Resilience' to R.H. and W.K., and DFG grants GR 3619/13-1 and 3619/19-1 to V.G. R.H. received support from the Focus Program Translational Neurosciences at the University Medical Center

Mainz. E.R. was supported by #NEXTGENERATIONEU (NGEU) and funded by the Ministry of University and Research (MUR), National Recovery and Resilience Plan (NRRP), project MNESYS (PE0000006) – A Multiscale integrated approach to the study of the nervous system in health and disease (DN. 1553 11.10.2022).

## Author contributions

Conceptualization, D.W., R.H. and W.K.; Methodology, D.W., E.R., W.W.F. and W.K.; Investigation, D.W., R.H., Y.Z., M.F.S.; Resources, M.A., W.K., W.W.F., J.R.R. and V.G.; Formal Analysis and Visualization, D.W., R.H., M.M., W.K. and J.R.R.; Writing – Original Draft, D.W., R.H. and W.K.; Funding Acquisition, C.L. and W.K.; Supervision, J.R.R., W.W.F and W.K.

## Funding

## Competing interests

The authors declare no competing interests.
