## [Peer Review File · Nature Communications]

Oxytocin induces the formation of distinctive cortical representations and cognitions biased toward familiar miceREVIEWER COMMENTS

Reviewer #1 (Remarks to the Author):

This is an exceptionally thorough and compelling manuscript which provides advances in our understanding whereby oxytocin influences social behavior. The authors use an outstandingly combination of methods, all rigorously described in their use, to test the regulation of social odor encoding by oxytocin. For decades we have known that social recognition memory (habituation in responding to familiar conspecifics) is influenced by oxytocin and the binding of oxytocin upon oxytocin receptors which are distributed throughout the brain -- including densely in the anterior olfactory nucleus the focus of the current manuscript. Given the rich expression of oxytocin in the AON and the monosynaptic input of odor information in the AON the authors investigate how oxytocin shapes the representation of odors among AON ensembles and also how this influences activity in downstream structures. The results that familiarity drives a more pronounced representation of social odors of familiar conspecifics vs simply a habituation of their response is interesting and changes common notions regarding the actions of this neuropeptide and how it regulates social behavior.

This manuscript was a pleasure to read -- easy to follow with methods carefully described and conclusions well-articulated and supported by the results. From my perspective this would be a great fit in Nature Communications and should be appealing to a variety of readers coming with interests in neuromodulation, sensory processing, and social behavior. I have only a few points to address in revision:

I would be interested in hearing the authors speculation on this idea which is certainly an experiment for a far later paper. And that is...With oxytocin the cortex doesn't distinguish, but what about the behavior? In other words, does a change in behavior directly track changes in cortical representations? This is what the manuscript is suggesting ... that the representation would directly inform and even predict behavior in real-time.

How does the experimental design account for contamination of the smell of the experimental animals own body odor to mix with the familiar (or unfamiliar) odors of the test subjects? This may be a limitation in the otherwise experimental clarity whereby the authors delivered familiar/unfamiliar odors and monitored responses. I see no way around this inherent issue in social odor experiments. Perhaps a short mention in discussion?

Super minor:

Not everyone agrees beta is generated in olfactory cortex as eluded to at one place, some suggest it is generated in bulb and is refined/modulated by cortical feedback. Could this be clarified or softened.

The first few subtitles in Results would benefit from revision to more say a result versus just the method/approach as well done in later Results sections.

Since animals don't "acquire encoding" please adjust: "In summary, familiar animal NEURAL NETWORKS [or something other wording] acquire a distinctive encoding that utilizes the cortical representational space more broadly, promoting storage of complex social relationships."

Check spelling throughout, including minor misc words like ketaminE, xylazinE, etc.

Reviewer #2 (Remarks to the Author):

The manuscript by Wolf et al. addresses a very important question, i.e., how social recognition memory (SRM) is encoded in the olfactory cortices. In social recognition assays, a test mouse typically spends more time investigating a novel mouse than a familiar one. Since repeated exposures to odors lead to adaptation in behaviors and neural representations in the olfactory system, less investigation of the familiar mouse is often interpreted as a habituation process. The authors devised an apparatus which allowed direct measurement of neural representations of the volatile body odors from different mice in a controlled manner. They found that body odor-induced neural representations in two olfactory cortical regions (anterior olfactory nucleus (AON) and posterior piriform cortex (pPC)) were sufficiently distinct to decode the identity of familiar versus novel mice. Contrary to what the habituation model would predict, familiar mice induced stronger sniff and pupil responses than novel mice, suggesting higher salience is assigned to familiar mice. Oxytocin released from hypothalamus is an essential mediator for social recognition and AON has a high expression level of oxytocin receptor. Using fMRI, the authors identified AON as one of the most activated brain regions upon optogenetic oxytocin release. Furthermore, oxytocin release led to diverging population responses to the familiar and novel social odors while knockout of oxytocin receptor in the AON had the opposite effect. These findings support that oxytocin-induced plasticity in the AON promotes the separation of the cortical representations of familiar social odors from others. The results are interesting, and the manuscript is well written. The manuscript could be further improved by addressing the following points.

Major

1. The authors examined social odor representations in several olfactory cortical regions including the AON, pPC and LEC. Curiously, the anterior piriform cortex, a major olfactory cortical hub, is not included. In addition, the manuscript focuses on the role of AON. However, the difference of the AON neuron responses to familiar versus novel social odors was quite small (Fig. 2c, 2d; Supplementary Fig. 6), apparently smaller than that in pPC. The sample size in AON is much larger than the other regions, which would reduce the p value. Since the piriform cortex also expresses oxytocin receptor (e.g., Choe et al., 2015, Neuron), its role in social recognition could be better addressed experimentally or by more in-depth discussion.

2. In Supplemental Figure 5, it is surprising that the sniff frequency did not increase upon peanut butter odor presentation. Mice show innate attraction to this odor, which triggers strong sniffing and investigation in freely behaving mice. Is the lack of sniffing due to head-fixation? Since odor representation in the olfactory system is significantly shaped by sniffing (odor sampling), the minimal changes in breathing rates in response to social or non-social odors raises the possibility that the findings may not apply to freely behaving mice. It would be great if the authors can address this point experimentally or by discussion.

3. Figure 2 and Supplementary Fig. 6 showed that the test mouse showed a stronger response to the familiar than the novel mouse odor in AON and pPC. Since the novel mouse odor was not associated with any object of importance, the difference in their neural representations could arise from several possibilities: a) the slightly increased sniff rate in response to the familiar mouse odor; b) simple exposure to the familiar mouse odor; c) social memory formed during social interaction. More experiments would be required to differentiate these possibilities. For example, an additional control with 5 min exposure to the familiar mouse odor but without actual social interaction would distinguish b) and c). Comparing the responses in trials with similar sniff rates may help to rule out a).

4. Related to the above point #3. How does pre-exposure to a non-social odor (e.g. peanut butter) for 5 min influence the neural representation later on? This will reveal whether the phenomenon observed in Figure 2 is specific to social odors.

Minor

5. On Page 5 and Figure 3, the authors concluded that the AON propagates the social recognition memory information top-down to the MOB, which was supported by evidence including “the AON units peaked in their firing response before the MOB” (Line 205-207). The firing peak is partially determined by the intrinsic neuronal properties and may be reached after behavioral decisions.

Have the authors looked at other measurements, for example, the latency to firing upon odor onset? In addition, the MOB shows more diverging responses to familiar and novel mouse odors than the AON (c.f. Fig. 3k,l to Fig. 2c,d). To fully support the top-down flow of SRM information, the authors may have to examine the effects of disrupting the AON to MOB projection or consider toning down this claim.

6. The current study focuses on AON as the only olfactory cortical region that is highly activated by optogenetic oxytocin release. How do the authors reconcile this finding with previous reports that oxytocin mediates social learning and social behavior via its action in other brain regions (e.g., the piriform cortex and amygdala)?

7. Could the authors clarify how the emitter mice were housed? In Figure 1, were the C57BL6#1 and #2 mice housed together or separately? Were the familiar and novel juvenile mice from different cages?

Reviewer #3 (Remarks to the Author):

In this manuscript, the authors investigate neural phenomenon related to social recognition memory. In many animals, including mice, individuals that an animal has previously encountered in a social setting is recognized later as familiar, when presented with other familiar or unfamiliar individuals. This study focuses specifically on the recognition part and the idea that when an animal smells a familiar individual, somewhere in the brain the evoked neural activity will be altered by its previous experience – effectively triggering social recognition memory (SRM). Another key focus is the role of oxytocin, which is released during social interactions and affects neural circuits, leading to changes in behavior.

There is a lot of excellent data in the manuscript. Some of the key observations are that neural activity in some early olfactory sensory areas (and the mesolimbic dopaminergic region) reflect the knowledge of the familiarity of the individual animal that is being sensed. The investigators have developed an excellent assay for studying this phenomenon, which allows them to take volatile body odors from so-called emitter mice to be presented in repeatable and systematic manner to so-called receiver mice. They have also done a huge number of recordings from multiple different areas – tetrode recordings from the anterior olfactory nucleus, piriform cortex, lateral entorhinal cortex, ventral tegmental area as well as the main olfactory bulb. They also have done fiber photometry of cortical feedback as well as field potential recordings. Moreover, they have used oxytocin manipulations since a key hypothesis being tested is that oxytocin plays a key role in the familiarization step when an individual encounters a conspecific. At that stage, oxytocin is released, which alters sensory processing (seemingly by changing the signal-to-noise ratio), and triggers plasticity mechanisms that allows the signals from individuals encountered during that social encounter to be now coded differently in the early olfactory areas.

Overall, this manuscript offers a novel preparation to study social signaling and adds enormous amount of interesting new data. There are some potential concerns that need to be clarified. First, the authors make a puzzling choice of analysis for differentiating neural representations of different stimuli. A rather standard way, by now, of discriminating different stimuli is to do linear (or nonlinear) decoding analysis, which compares population responses for different stimuli. One needs to compare across stimuli – here the authors seem to be comparing distances with respect to the baseline for each of the stimuli (effectively comparing some sort of an amplitude metric). This could be problematic, since the trajectory of the familiar mouse could be very different from the neural population trajectory of other mice, but essentially trace a path that is similar in the sort of Euclidean measure the authors use (for example, think of an ellipse that is exactly mirror image in the 3-dimensional PC space they show – the trajectories are very different, but the Euclidean measure might look the same). I confess that it's unclear whether the authors do what I think they are doing, so if I am mistaken, it might be worth clarifying the methods in detail in the manuscript.

There's also some concern about the use of a very small number of distinct stimuli. This may be inescapable for the kind of experiments done here, however using just two individuals from the same strain and another from the strain along with just two other general odors, seems rather marginal. It would be useful to see some discussion (and reassurance) from the authors whether the results can be biased by such small number of stimuli.

Specific Comments:

Line 126: It's not clear how to match the trajectories in Fig 1ef, with the results in fig 1g. For example, the differences between blue and magenta trajectories are minimal, but yet the p values in Fig 1g are very low (black squares).

Line 132: would be good to see the original sniff rates – I think this is in Supp Fig 5, but perhaps the authors can make it clear?

Line 138: The number of distinct stimuli is still low (5), and it's hard to be sure about chance effects - that is, by chance the 2 same-strain individuals happen to be similar, but if you test many other pairs there might be differences. While I understand the difficulty of getting many animals, I worry about selection bias and the authors could perhaps assure the reader that this is not an issue?

Line 161 and elsewhere: Could the higher firing rate to familiar mouse be simply due to higher sniff freq, as shown in Fig 2b?

Line 186: This term coupling seems to always imply (in a lot of the literature) some sort of physical connection - but really what the data show is that there is greater coherence in some frequency band. Why not use a neutral, more justified terminology like "coherence between x and y is increased"? But this is not a demand by any means – but a point of view that may be shared by a lot of the readership.

Line 245: This observation about AON being the main olfactory region to get activated by OXT stimulation seems very interesting, and I urge the authors to make this more prominent!

Figure 1g: To reiterate an earlier point made above: It's not clear that mean Euclidean distance is the best metric. For example, two different trajectories that are widely divergent can have similar Euclidean distances from baseline. In fact, the difference between the green and blue trajectories in panel e is very large, but if we look at panel f, they look much closer.

Reviewer #4 (Remarks to the Author):

In their manuscript the authors examined the neural basis for social recognition memory testing the hypothesis whether familiar odors from other mice are encoded by habituation memory or by gaining more distinct representations that are reinforced by social interactions through experience dependent plasticity. In the latter case the conspecific should trigger "stronger and more differentiated neuronal responses".

To this purpose they recorded multiple single unit activities in different nuclei of male C57BL/6J OXT-Cre transgenic mice: the anterior olfactory nucleus (AON, 8 animals), the posterior piriform cortex (pPC, n=11) and the lateral entorhinal cortex (LEC, n=11). In addition they recorded for some experiments in the main olfactory bulb (MOB, 8 animals) and the ventral tegmental area (VTA, 3 animals). Single unit identification was achieved by using tetrodes arranged in multiple arrays (2x 4 or 7), depending on the nucleus.

The mice were head-fixed for recording spiking activities before being repeatedly exposed for 1 second to five different odors: peanut butter, ylang ylang, and odors from same sex juvenile mice: one mice from CD1 (a different strain), and two C57BL/6J mice (#1 and @2).

In a second series of experiments the mice were first exposed for five minute interaction with one of two C57BL/6J mice so as to familiarize the experimental mice with this odor and test whether the encoding for this odor would have changed in one of these nuclei.

In a third series of experiments the authors looked at the effects of oxytocin receptor activation or inhibition on the representations of the encodings.

This is an interesting series of experiments of how encoding of familiar and unfamiliar mice may occur by neuronal activity in different nuclei and the role of oxytocin signaling in this encoding.

However, there are a number of questions that arise on this work.

Major points

1. Representation of social odors.

Line 138: "Importantly, the examined regions and physiological readouts pooled across individuals do not differentiate among unfamiliar C57BL/6 males under our conditions. The social odor assay is thus suited to study if and how familiarity memory is implemented in the neuronal population code."

I am not sure if this is fully correct. The authors tested the lengths of the Euclidian distance of each of the vector to baseline, but not the orientation of the vectors. Although the PCA analyses show similar traces for the AON, in the pPC and LEC, the representations of the two C57BL6 mice #1 and #2 reveal rather distinct traces. . The authors should test for the two unfamiliar mice of similar strain across all nuclei the Euclidian distance between the endpoints of the vectors encoding for different non-familiar animals and compare the orientations of the different vectors.

In all, it seems in fact most surprising that in the nuclei they tested, they found differential representations of ylang ylang and peanut butter, but no differential representations of different individuals. It is important to fully nail this down, because this point has major implication for the changes that occur upon familiarization of one of the two animals.

2. Neuronal encoding of familiarity

It is a bit surprising that the authors make a cross-comparison of the representation of two different mice, one with which the experimental animal has become familiar, and one with which it has not become familiar. They then again compare vector length by Euclidian distance to baseline.

If indeed the training of a familiar mouse leads to a stronger representation (potentiation of responses) by the originally encoding neurons, it would be more straight forward to longitudinally test in the experimental mouse what changes occur in the representation of the demonstrator mouse before and after familiarization in a longitudinal comparison. Indeed, it seems rather surprising that the length of the vector (the euclidian distance between baseline and stimulus exposure) for the familiar stimulus would be longer in length than the unfamiliar stimulus as both are likely to have completely differently oriented representations in this multidimensional vector space. On the other hand, if indeed the social recognition memory is represented by (as the authors hypothesize in line 83) "stronger neuronal responses", the representation in the multidimensional space should stay in similar direction, but with a longer Euclidian length of the population vector from the baseline.

Thus, the crucial experiment to address their hypothesis would be to make longitudinal comparisons between neuronal representations before and after familiarization with the same mouse.

Other points:

1. To obtain a better appreciation of "ground-truth" differences in representations of single unit responses to the odor in the different nuclei the supplementary figures 3,4,7, 9b, 11b,13e&h are most useful. In particular, to gain an understanding concerning the differences in encodings of different C57BL6 mice (both unfamiliar vs familiar), it is important to sort the activity of these units according to one of the C57BL6 mice. In Supplementary Figure 3, for example, unit activity seems to have been sorted according to different single unit responses to the odor ylang ylang. To gain an understanding of the differences between C57BL6#1 and #2, it would be important to add a similar figure, but with all units sorted according to the responses of the units to the odor of C57BL6#1. This would allow a better appreciation of the level to which the responses to C57BL6#2 would be different (or similar) and give us a more intuitive understanding of why there is so much overlap in the PCA representation by these 313 units of C57BL6#1 and #2 in figure 1e.

Idem for supplementary figures 4a&b.

A similar questions arises for the representation presented in Supplementary figure 7. Here, the authors would like to claim that the representation of the familiar mouse (versus the novel mouse) occurs through stronger single units reponses (and thus a longer Euclidian distance of the population vector to the baseline). At first glance this may indeed seem to be the case (there are more, and more intense, and longer red traces of units in the familiar versus the novel mouse).

However, this impression may also be caused by the sorting of the units according to the familiar mouse. The authors seem to be aware of this, and have thus also included a "resorted" column on the far right, that seems to show the sorting of the units according to the "novel mouse". However, one crucial column is lacking in this representation, namely the column of the "familiar mouse" when its units have also been sorted according to the "novel mouse". To appreciate to what extent this visual impression of a stronger representation of the familiar mouse persists, it is crucial to also show this fourth column of the familiar mouse when the units have been resorted according to the novel mouse. (idem for supplementary figures 11b and 13 e and h).

Fig. 2c. and Supplement. The absolute differences in spiking activity of units during exposure to familiar versus novel mice seems very small and not very convincing. The significant differences that the statistical tests come out with may possibly be caused by which test is chosen and how it is applied. For that matter, one wonders to what extent these differences in unit activity per individual animal show a correlation with individual differences in behavioral measurements that have been recorded. Indeed, supplementary figure 6a shows quite some variability in sniffing frequencies across trials for the (8?) mice that have been tested. These differences in behavior should correlate with differences in spiking activities of neurons in the AON, pPC and LEC that the authors measure across these different 8 mice. Did these two parameters (sniffing frequency per animal versus differences in single unit activity per animal) show a correlation? Can the authors provide such a plot (and a table) showing the differences in behavior per animal and the number of units per animal that responded significantly differently between exposure to novel versus familiar rats? Can the authors correlate the differences in behavioral responses per mouse (e.g. sniffing frequencies) with the differences in single unit activities per mouse (Euclidian lengths of the population vectors). Such a correlation could strengthen the relationship between encoding of familiarity responses by single unit activity.

(or, for example, normalize the differences in unit responses on the differences in sniffing behavior

3. Idem for the modulation of the oxytocin responses, be it by potentiation of oxytocin or inhibition with antagonist: The authors seem to show only single unit responses but none of the behavioral responses of the individual animals.

Minor points

The structure of the introduction seems to lack some logic. In particular, the authors conclude in line 75 that these findings generate the prediction that the plasticity in the neuronal representation of conspecifics should be modulated bi-directionally". Do they mean both potentiation and depotentiation? If so, could they clearly state this and also on which arguments they conclude that

this should be in both directions? (from the preceding it only seems that OT increases behavioral SRM.

2. Line 78: "There are thus two competing hypotheses how SRM is encoded". From the previous text it is not immediately clear why there would be TWO competing hypotheses. Can the authors explain this argumentation more clearly? Or maybe slightly rephrase, since the premiss does not automatically lead to this conclusion.

3. Line 82: "In this case, sampling a familiar conspecific should trigger a stronger and more differentiated neuronal responses." This sentence is unclear: "stronger and more differentiated" "Stronger" -where?, "differentiated" - From what? I would say this should depend in which area of the brain one measures, one could imagine: E.g. an upstream area where the responses are more strongly differentiated between familiar and non-familiar conspecific, and a downstream area where there is less response to familiar conspecifics and thus less drive for action.....

4. Line 87: "more distinct population responses" distinct from what?

5. "The cortical SRM is communicated top-down to earlier sensory circuits" What do the authors mean with "earlier sensory circuits"? What do they mean with "cortical SRM"? A (presumed) representation of the SRM in the AON?, pPC ? MOB?

Results:

Line 102: "We presented the odors of two C57BL/6 and one CD1 male (mice, we assume?). To what kind of receiver mouse? (which line, which sex?) And what ages? It would be useful for the reader to receive this information at this point without having to search in the methods section.

Line 104: "The contributions of different olfactory cortex regions in the coding of SRM are not known." (not clear what is the function of this sentence at this point in the text. Seems to more belong to the introduction.

Line 111: "SRM entails two aspects" The two aspects that follow have nothing to do with the "memory of SR", rather with the "social recognition itself "

Line 121 "The population responses to the odors from unknown C57BL/6 displayed a shared trajectory across the cortices (Fig. 1e, Supplementary Fig. 2e)". In fact, the trajectories shown in suppl Fig. 2e (for the pPC and LEC) are very different.

Line 123: In the AON, the difference between the "CD1" and "ylang ylang" in the first half second is (according to Supplem. Fig. 1d) significant but not anymore in the second half second. However, the Euclidian distance between these two odors is very small in the first half second and larger in the second half second (Fig. 1f). Could the authors verify this significance?

Supplem. Fig. 3: The most relevant information (namely on the hard core data of single unit responses) can be found in Supplem. Fig. 3. From that figure it seems that the authors had very few units in the pPC and LEC responding to the different odors. In fact, how did the authors calculate the z-score (compared to baseline?).

Line 160: "These units recruited by the familiar mouse also had stronger firing rate responses in AON and pPC as compared to the smell of the novel mouse (Supplementary Fig. 6f)."

This statement is unclear and confusing. If a unit is recruited by the familiar mouse (and thus was not considered recruited to the unfamiliar mouse), then it should of course be showing stronger firing rates as compared to the smell of the novel mouse.

What is important here is to define the criteria by which a unit is considered to be recruited or not (responsive or not) to the familiar versus novel mouse.

The question here is: HOW did the authors average their responses in supplementary figure 6f? Did they take averages only of those units that were recruited and compared these with their activities to a mouse that was novel? (e.g. the 9% that is shown in Suppl. Fig. 6e). What about the units that responded to BOTH familiar and novel mice (grey pie chart). Were there any differences in responses of these units to familiar and non-familiar mice ?

Point-by-point response to the reviewers' comments to Wolf et al.

Reply to Reviewer #1 (Remarks to the Author):

This is an exceptionally thorough and compelling manuscript which provides advances in our understanding whereby oxytocin influences social behavior. The authors use an outstandingly combination of methods, all rigorously described in their use, to test the regulation of social odor encoding by oxytocin. For decades we have known that social recognition memory (habituation in responding to familiar conspecifics) is influenced by oxytocin and the binding of oxytocin upon oxytocin receptors which are distributed throughout the brain -- including densely in the anterior olfactory nucleus the focus of the current manuscript. Given the rich expression of oxytocin in the AON and the monosynaptic input of odor information in the AON the authors investigate how oxytocin shapes the representation of odors among AON ensembles and also how this influences activity in downstream structures. The results that familiarity drives a more pronounced representation of social odors of familiar conspecifics vs simply a habituation of their response is interesting and changes common notions regarding the actions of this neuropeptide and how it regulates social behavior.

This manuscript was a pleasure to read -- easy to follow with methods carefully described and conclusions well-articulated and supported by the results. From my perspective this would be a great fit in Nature Communications and should be appealing to a variety of readers coming with interests in neuromodulation, sensory processing, and social behavior. I have only a few points to address in revision:

We thank the Reviewer for the feedback and helpful comments. We addressed them accordingly. The line numbers indicated in this reply refer to the numbers of the attached manuscript .pdf with highlighted changes in red.

I would be interested in hearing the authors speculation on this idea which is certainly an experiment for a far later paper. And that is...With oxytocin the cortex doesn't distinguish, but what about the behavior? In other words, does a change in behavior directly track changes in cortical representations? This is what the manuscript is suggesting ... that the representation would directly inform and even predict behavior in real-time.

We thank the reviewer for raising this implication. If we understand the question correctly, there would be the following prediction that could be tested in the future: The population response amplitude upon recognition via volatile odor cues predicts the subsequent exploration behavior (for instance approach or no approach). When envisaging a study that directly addresses this point, it comes with a number of challenges in power to detect this in a self-paced condition as it is not clear when exactly and in which level of proximity animals perceive each other, given the possibly turbulent propagation of odor plumes. Adopting an "inside-out" perspective, where neural dynamics are used to predict observable behaviors, could be an effective approach for investigating the temporal and spatial dynamics of interacting animals.

How does the experimental design account for contamination of the smell of the experimental animals own body odor to mix with the familiar (or unfamiliar) odors of the test subjects? This may be a limitation in the otherwise experimental clarity whereby the authors delivered familiar/unfamiliar odors and monitored responses. I see no way around this inherit issue in social odor experiments. Perhaps a short mention in discussion?

In our head-fix testing condition, we minimized the possible contamination of own smells. In the recording chamber, we had an active exhaust to reduce the accumulation of own and presented odors. Further, there is a continuous air flow (clean air switching with odorized air) onto the nose of the animal. The in- and outgoing continuous airflows are configured to minimize the persistence of odors in the recording chamber. We added this point now to the method section (l. 519): **“To avoid accumulation of the receiver animals’ own or the presented odors in the recordings chamber, an active exhaust matching the flowrate of the continuous inflow from the olfactometer was applied.”**

Super minor:

Not everyone agrees beta is generated in olfactory cortex as eluded to at one place, some suggest it is generated in bulb and is refined/modulated by cortical feedback. Could this be clarified or softened.

We agree that there are competing theories on the generation of olfactory beta-oscillations. We have adjusted the main text in l. 238 as suggested: **“Beta oscillations are a network phenomenon involving the olfactory cortex—originate in the olfactory cortex** and have been linked to olfactory learning processes (Kay et al., 2009)”.

The first few subtitles in Results would benefit from revision to more say a result versus just the method/approach as well done in later Results sections.

We thank the reviewer for this feedback and have adjusted the first two subtitles in the Results accordingly. We replaced „Representations of social odors.” (1st subheading) with **“Cortical populations encode identity of conspecifics.”** We replaced „Neuronal coding of familiarity.” (2nd subheading) with **“Population response amplitude encodes familiarity.”**

Since animals don’t “acquire encoding” please adjust: “In summary, familiar animal NEURAL NETWORKS [or something other wording] acquire a distinctive encoding that utilizes the cortical representational space more broadly, promoting storage of complex social relationships.”

We adjusted the abstract as suggested: **“In summary, neuronal encoding of familiar animals is distinct and utilizes the cortical representational space more broadly, promoting storage of complex social relationships.”**

Check spelling throughout, including minor misc words like ketaminE, xylazinE, etc.

We apologize for the spelling mistakes and have corrected them.

Reply to Reviewer #2 (Remarks to the Author):

The manuscript by Wolf et al. addresses a very important question, i.e., how social recognition memory (SRM) is encoded in the olfactory cortices. In social recognition assays, a test mouse typically spends more time investigating a novel mouse than a familiar one. Since repeated exposures to odors lead to adaptation in behaviors and neural representations in the olfactory system, less investigation of the familiar mouse is often interpreted as a habituation process. The authors devised an apparatus which allowed direct measurement of neural representations of the volatile body odors from different mice in a controlled manner. They found that body odor-induced neural representations in two olfactory cortical regions (anterior olfactory nucleus (AON) and posterior piriform cortex (pPC)) were sufficiently distinct to decode the identity of familiar versus novel mice. Contrary to what the habituation model would predict, familiar mice induced stronger sniff and pupil responses than novel mice, suggesting higher salience is assigned to familiar mice. Oxytocin released from hypothalamus is an essential mediator for social recognition and AON has a high expression level of oxytocin receptor. Using fMRI, the authors identified AON as one of the most activated brain regions upon optogenetic oxytocin release. Furthermore, oxytocin release led to diverging population responses to the familiar and novel social odors while knockout of oxytocin receptor in the AON had the opposite effect. These findings support that oxytocin-induced plasticity in the AON promotes the separation of the cortical representations of familiar social odors from others. The results are interesting, and the manuscript is well written. The manuscript could be further improved by addressing the following points.

We thank the Reviewer for the feedback and helpful comments. We addressed them accordingly. The line numbers indicated in this reply refer to the numbers of the attached manuscript .pdf with changes highlighted in red.

Major

1. The authors examined social odor representations in several olfactory cortical regions including the AON, pPC and LEC. Curiously, the anterior piriform cortex, a major olfactory cortical hub, is not included. In addition, the manuscript focuses on the role of AON. However, the difference of the AON neuron responses to familiar versus novel social odors was quite small (Fig. 2c, 2d; Supplementary Fig. 6), apparently smaller than that in pPC. The sample size in AON is much larger than the other regions, which would reduce the p value. Since the piriform cortex also expresses oxytocin receptor (e.g., Choe et al., 2015, Neuron), its role in social recognition could be better addressed experimentally or by more in-depth discussion.

The anterior piriform cortex may also be involved in recognition memory as the here identified AON and pPC. We however observed in another study directly comparing AON and aPC on social and non-social odor encoding that AON and aPC differ substantially in how they encode odors in awake mice (Wolf, Oetl, Linster, Kelsch, unpublished observation). Specifically, while AON is providing net excited output to odors, the net output of anterior piriform cortex is mostly an inhibition relative to baseline activity. Thus, learning plasticity in anterior PC may involve other mechanisms and deserves a separate study. We

therefore decided to focus the present manuscript on the AON and pPC and other relevant attached regions.

We agree with the reviewer that the difference between familiar and novel mice appears somewhat larger in the pPC than in the AON, at least relative to the overall odor response amplitude. Previous work showed that oxytocin (OXT) in the AON is necessary for behavioral expression of social recognition memory in mice (Oettl et al., 2016) and we identified the AON as a hub with prominent BOLD activation upon optogenetic OXT release using fMRI. Given these findings, we focused on the AON in the current manuscript and use interventions in the OXT system as a causal test for memory induction. This certainly doesn't intend to exclude that also OXT receptors in the piriform cortex contribute to social recognition memory and we now highlight this point as suggested through an in-depth discussion starting l. 426:

“This complements previous work showing that OXT receptor activation in the piriform cortex entrains association learning of non-social odors to social cues (Choe et al., 2015) and highlights the various contributions of piriform cortex also to social memories. Future studies may dismantle the respective contributions of different olfactory cortices to retrieved social recognition memories and how the cortices interact with each other in these processes. Here, the connectivity of each cortex with other brain regions and their local computations of the same stimulus will have to be clarified.”

2. In Supplemental Figure 5, it is surprising that the sniff frequency did not increase upon peanut butter odor presentation. Mice show innate attraction to this odor, which triggers strong sniffing and investigation in freely behaving mice. Is the lack of sniffing due to head-fixation? Since odor representation in the olfactory system is significantly shaped by sniffing (odor sampling), the minimal changes in breathing rates in response to social or non-social odors raises the possibility that the findings may not apply to freely behaving mice. It would be great if the authors can address this point experimentally or by discussion.

We thank the reviewer for raising this clarifying point. We added a new analysis of the sniff frequency for the first part of the time window of the odor presentation to Supplementary Fig. 5b-d. We find that within the first 330 ms of odor presentation (corresponding approximately to the duration of the first sniff cycle at a sniff frequency of ~3-4 Hz, cf. Supplementary Fig. 5a), the sniff frequency increased uniformly to all five odors (Supplementary Fig. 5b-c) with little adaptation throughout the session (Supplementary Fig. 5d). Together with the analyses of the full odor presentation window of 1 s, this supports the notion that the three odors of the conspecifics lead to more sustained increases in sniff frequency than peanut butter or the flower odor. It is certainly possible that sniff frequency increases would be larger in freely moving animals. The size of the relative rate increases are consistent with a recent study where plasticity was induced optogenetically to an odor in the same awake head-fix preparation (Oettl et al., 2020).

We modified the Results section at l. 168: *“While all odors led to an increase of the sniffing frequency upon stimulus onset, the increase was more sustained in response to social odors (Supplementary Fig. 5a-d).”*

And provide the respective figures:

Supplementary Fig. 5, related to Fig. 1. Physiological responses to social and non-social odors.

b Boxplot of the session-average change in sniff frequency from pre-stimulus baseline (top) in the first part (0 to +0.33 s) or (bottom) during the full window (0 to +1 s) of the odor presentation (paired t-test). **c** P-values of pairwise comparisons of sniff frequency responses between odors during the (top) first part (0 to +0.33 s) or (bottom) during the full time (0 to +1 s) of odor presentation (repeated-measures one-way ANOVA with post-hoc Tukey's test for multiple comparisons; $n = 13$ animals with 1 session each). **d** Stability of sniff responses to each of the sampled odors is shown using the same metrics as in Fig. 1h for blocks of 5 consecutive trials as mean \pm SEM during (top) the first part (0 to +0.33 s) or (bottom) the full time (0 to +1 s) of odor presentation.

3. Figure 2 and Supplementary Fig. 6 showed that the test mouse showed a stronger response to the familiar than the novel mouse odor in AON and pPC. Since the novel mouse odor was not associated with any object of importance, the difference in their neural representations could arise from several possibilities: a) the slightly increased sniff rate in response to the familiar mouse odor; b) simple exposure to the familiar mouse odor; c) social memory formed during social interaction. More experiments would be required to differentiate these possibilities. For example, an additional control with 5 min exposure to the familiar mouse odor but without actual social interaction would distinguish b) and c). Comparing the responses in trials with similar sniff rates may help to rule out a).

We thank the reviewer for this comment that we addressed with adding new data and analyses. We have investigated possibility **a)** by adding a trial-by-trial correlation of sniff rate vs population response amplitude (Euclidean distance from baseline) to Supplementary Fig. 9b-c. We included all AON sessions with sniff recordings. We focused on AON as the per-session unit count was highest among regions and

sufficient to build population vectors for single sessions. We then determined the correlation using Spearman's correlation coefficient. We find that single-trial sniff rate and response amplitude are not correlated, supporting that the stronger neuronal response was not caused by the higher sniff rate.

We added to the Results section, l. 205: “The stronger neuronal response in the AON to the familiar odor was however not explained by the sniff rate modulation in a trial-by-trial correlation (Supplementary Fig. 9b-c).”

In the method section l. 733: “**Correlation analysis: neural response vs. sniff response.** To investigate whether the population response amplitude was mediated by the sniff frequency, we performed a trial-by-trial correlation of sniff frequencies versus the Euclidean distance from baseline. We included all AON sessions with simultaneous sniff recordings, as the per-session unit count was highest in AON among regions and sufficient to build population vectors for single sessions. For every session, we then computed the Euclidean distances from baseline from the single-session population response vectors and correlated the mean distance during odor presentation (0 to +1 s relative to odor onset) to the average sniff frequency of the trial (0 to +1 s relative to odor onset) using Spearman's correlation coefficient.”

And provide the new Supplementary Fig. 9b-c:

Supplementary Fig. 9, related to Fig. 2. Intra-individual correlations of behavioral readouts and neuronal activity. b-c For all AON sessions with simultaneous sniff recordings, we plotted the sniff frequency during (b) familiar and (c) novel odor presentations versus the mean Euclidean distance from baseline of that trial (0 to +1s relative to odor onset). We computed the trial-by-trial correlation for every session individually using Spearman's correlation coefficient and tested the significance of that correlation coefficient. We find no significant association of trial-by-trial sniff frequencies versus population response amplitude (p-values for every session indicated in the figure). Note that the reference lines for the correlation within the sessions was plotted for illustration without considering outliers, while the correlation coefficient was tested for significance considering all trials.

To differentiate between **b) and c)** we have added additional data and analyses to Fig. 2d and Supplementary Fig. 9a. Towards this aim, we considered several approaches to disentangle these scenarios. Fencing to prevent social interaction may be generally an option; yet, this may be ambiguous when it comes to drawing conclusions, as it will entirely change the exploration behavior. Alternatively, the intensity (be it frequency or total duration of interactions) of direct social contacts (i.e., interaction bouts) during spontaneous exploration can be tested for its correlation to the strength of the later retrieved neuronal memory. We therefore performed behavior annotation of the video recordings from

the initial freely-moving exploration period (preceding the head-fixed recording) where the mouse familiarizes itself with the conspecific. We find that the number of interaction bouts shows a positive correlation with the memory strength during later retrieval (Fig. 2d), but not the total interaction time (Supplementary Fig. 9a). This provides additional evidence supporting that the social interaction mediates the memory formation and that more “induction” events lead to stronger memory expression.

We added to the Results section, l. 202: “The expressed neuronal memory (familiar – novel) positively correlated with the number of sampling events during the exploration phase (Fig. 2d), but not the sheer total contact duration (Supplementary Fig. 9a). This may hint on that social memory formation is promoted by repeated sampling.”

and in the method section l. 548: “The interaction was recorded on video using a Sony FDR-X1000V camera with a frame rate of 30 fps. Exploration behavior was later annotated manually using the BORIS software (Friard and Gamba, 2016). Frames where the experimental subject’s nose touched the conspecific were labelled as interaction. Two subsequent interaction bouts were counted as individual events if they were separated by more than one second.”

and l. 745: “**Correlation analysis: neural response vs. exploration behavior.** To investigate if the memory strength was associated with the intensity of exploratory behavior during the familiarization period, we performed a correlation of the number of interaction bouts or the investigation time versus the difference in Euclidean distance from baseline between the familiar and novel odor. We included all AON sessions as the per unit session count was sufficient here to build single-session population vectors. For every session, we computed the mean difference in Euclidean distance from baseline between familiar and novel odor responses and correlated that to the number of interaction bouts or to the total interaction time using Spearman’s correlation coefficient.”

And provide the new Figure 2d and Supplementary Figure 9a panels:

Fig. 2. Neural coding of familiarity.

d The correlation of the memory strength (difference in Euclidean distance from baseline between familiar and novel) and the number of interaction bouts during the freely-moving familiarization period shows a positive association ($n = 8$ animals, 3 sessions each; same data as c).

Supplementary Fig. 9, related to Fig. 2. Correlations of behavioral readouts and neuronal activity.

a The correlation of the memory strength (difference in Euclidean distance from baseline between familiar and novel) and the total interaction time during the freely-moving familiarization period shows no association ($n = 8$ animals, 3 sessions each). A linear fit was plotted for illustration of the slope. Spearman's rank correlation coefficient is indicated in the figure and was tested for significance.

4. Related to the above point #3. How does pre-exposure to a non-social odor (e.g. peanut butter) for 5 min influence the neural representation later on? This will reveal whether the phenomenon observed in Figure 2 is specific to social odors.

We thank the reviewer for raising this point. It is an interesting question whether reinforced cortical representations are also a coding mechanism for non-social odor objects, albeit not topic of this study. This question may be addressed in future studies focusing on foraging behavior. However, one difficulty with this question in experimental design is that we can control for individual identity of body odors from conspecifics with the same genetic background, but this approach cannot be applied to different food odors. Before becoming familiar with a conspecific, body odors of different individuals elicit the same length of the Euclidian distance vector (cf. Fig. 1). Therefore, a novel mouse can be used as a control stimulus for the body odor of a familiar animal. Different food smells however result in different lengths of the Euclidian distance vectors already before any memory or reinforcement (Supplementary Figs. 2 and 4 and unpublished observations), thus making it difficult to interpret the results. Thus, we would assume that similar plasticity mechanisms also apply to non-social odors, but with the current approach, we find this question difficult to answers. More importantly, this question is complicated by the point that social interactions are in itself rewarding and thus reinforcing (consistent with the new data provided related to the previous question). This is not the case for repeated exploration of a non-social odor. So, it could be considered a CS with zero reward. Here, others and we have seen that the response components in piriform cortex diminish with repeated exposure (conditioning) (e.g. Winkelmeier et al., 2022) . In case we were to scent a food with a neutral odor or consider the natural smell of a repeatedly palated food, we would have a classical CS-US conditioning known to produce reinforcement in many brain regions.

We made this point now more clear and modified the statement in the introduction, l. 86: **“On the other hand, social interactions are innately rewarding (Dölen et al., 2013a). A competing hypothesis thus states that social interactions reinforce the representations of these familiar animals through experience-dependent plasticity.”**

Minor

5. On Page 5 and Figure 3, the authors concluded that the AON propagates the social recognition memory information top-down to the MOB, which was supported by evidence including “the AON units peaked in their firing response before the MOB” (Line 205-207). The firing peak is partially determined by the intrinsic neuronal properties and may be reached after behavioral decisions. Have the authors looked at other measurements, for example, the latency to firing upon odor onset? In addition, the MOB shows more diverging responses to familiar and novel mouse odors than the AON (c.f. Fig. 3k,l to Fig. 2c,d). To fully support the top-down flow of SRM information, the authors may have to examine the effects of disrupting the AON to MOB projection or consider toning down this claim.

We thank the reviewer for this helpful comment. We have looked at the latency to firing upon odor onset. Consistent with the bottom-up flow of information, the first peak of the latency-to-onset distribution was earlier in MOB than AON. We now added this analysis to Supplementary Fig. 11h.

Supplementary Fig. 11, related to Fig. 3. MOB responses during recognition of familiarity.

h The latency to firing onset of excited cell-odor pairs from MOB and AON. The peak of the latency-to-onset distribution was earlier in MOB than AON, which is consistent with bottom-up flow of information.

There are however limitations, as glutamatergic monosynaptic bottom-up excitation of MOB to AON will have short latencies, and AON and MOB were recorded in the same apparatus and same conditions (olfactometer latencies), but not in the same animals.

In preliminary work, we had tried silencing top-down projections. We found however, that when silencing is effective, it massively disrupts odor representation in the cortex and olfactory bulb. It is therefore then difficult to disentangle the effect of disrupted odor responses in the cortical projections from the disrupted transmission of familiarity information in those projections.

Therefore, as suggested, we toned down the claim in that it is compatible with top-down modulation, but correlational evidence. The results from fiber photometry provide direct evidence that the SRM is transmitted from AON to MOB. We modified in the Results section l. 262: “Consistent with the late increase in top-down β activity, the AON units peaked in their firing response before the MOB (Supplementary Fig. 11h-j), providing correlational support of top-down modulation. In summary, these observations support that SRM ~~from the cortex~~ information is transmitted top-down from AON to the MOB ~~modulates top-down bulbar activity.~~”

And in the discussion, l. 394: “The top-down projections from AON to MOB also show stronger responses to familiar animals. One consequence of reinforced AON output is to drive top-down MOB interneurons

that then modulate mitral cells (Haberly and Price, 1978; Boyd et al., 2012; Markopoulos et al., 2012; Otazu et al., 2015; Oettl et al., 2016). Olfactory sensory neurons first excite mitral cells in the MOB that then output to cortices like the AON. Interestingly, the peak of the responses triggered by the social odors was reached only a few hundred milliseconds after the onset of the odor, and later in the MOB than in the AON. This ~~is consistent with a~~ could hint at top-down modulation affecting the development of the bulbar response.”

And in the Methods section, l. 816: “The latency to firing onset was determined for odor-excited cell-odor pairs from MOB and AON using the z-scored response with a binning of 10 ms to capture the onset more precisely. The latency to response onset was defined as the time when units surpassed a z-score of 1.96 (which corresponds to a firing rate response outside the approximate 95% confidence interval of firing activity without stimulus presentation). The distribution of onset latencies across excited cell-odor pairs was visualized in a histogram with a binning of 50 ms.”

6. The current study focuses on AON as the only olfactory cortical region that is highly activated by optogenetic oxytocin release. How do the authors reconcile this finding with previous reports that oxytocin mediates social learning and social behavior via its action in other brain regions (e.g., the piriform cortex and amygdala)?

We thank the reviewer for this clarifying point. The BOLD response is an indirect readout of brain activity as it relies on the change in blood oxygenation. Multiple cell-types are involved in generating a BOLD response and more complex activation patterns may not lead to significant BOLD activations. The cellular composition of the piriform cortex and amygdala and the cell-types expressing OXT receptors differs from the AON. For example, the amygdala has a large fraction of interneurons and oxytocin recruits prominently these amygdala interneurons and astrocytes (Knobloch et al., 2012; Wahis et al., 2021). This is different for the AON where glutamatergic projection neurons to the MOB express oxytocin receptors and oxytocin lowers the rheobase of putative excitatory neurons in the AON (Fig. 3F in Oettl et al., 2016). This could be one factor contributing to different BOLD response patterns upon OXT receptor activation. aPC has been shown to be a region with a comparatively high fraction of inhibitory responses (e.g. Winkelmeier et al., 2022) compared to AON. We are working currently on multi-region comparisons to understand potential sources of these differences. In the context of the present study, it was the goal to use the fMRI as a search machine to identify regions that are most prominently activated by oxytocin release as activation should facilitate reinforcement.

We have emphasized more in the Results section accordingly, l. 287: “It should be noted that some regions that also express OXT receptors like the piriform cortex (Choe et al., 2015) or the amygdala (Knobloch et al., 2012; Wahis et al., 2021), might not be captured for instance due to more complex BOLD activation patterns or preferential receptor expression in other cell-types.”

7. Could the authors clarify how the emitter mice were housed? In Figure 1, were the C57BL6#1 and #2 mice housed together or separately? Were the familiar and novel juvenile mice from different cages?

We added the information to the Methods section l. 470: "All emitter mice were single housed in fresh cages for at least 24 hours before the experiment to prevent cross-contamination of odors from other cage mates."

Reviewer #3 (Remarks to the Author):

In this manuscript, the authors investigate neural phenomenon related to social recognition memory. In many animals, including mice, individuals that an animal has previously encountered in a social setting is recognized later as familiar, when presented with other familiar or unfamiliar individuals. This study focuses specifically on the recognition part and the idea that when an animal smells a familiar individual, somewhere in the brain the evoked neural activity will be altered by its previous experience – effectively triggering social recognition memory (SRM). Another key focus is the role of oxytocin, which is released during social interactions and affects neural circuits, leading to changes in behavior.

There is a lot of excellent data in the manuscript. Some of the key observations are that neural activity in some early olfactory sensory areas (and the mesolimbic dopaminergic region) reflect the knowledge of the familiarity of the individual animal that is being sensed. The investigators have developed an excellent assay for studying this phenomenon, which allows them to take volatile body orders from so-called emitter mice to be presented in repeatable and systematic manner to so-called receiver mice. They have also done a huge number of recordings from multiple different areas – tetrode recordings from the anterior olfactory nucleus, piriform cortex, lateral entorhinal cortex, ventral tegmental area as well as the main olfactory bulb. They also have done fiber photometry of cortical feedback as well as field potential recordings. Moreover, they have used oxytocin manipulations since a key hypothesis being tested is that oxytocin plays a key role in the familiarization step when an individual encounters a conspecific. At that stage, oxytocin is released, which alters sensory processing (seemingly by changing the signal-to-noise ratio), and triggers plasticity mechanisms that allows the signals from individuals encountered doing that social encounter to be now coded differently in the early olfactory areas.

Overall, this manuscript offers a novel preparation to study social signaling and adds enormous amount of interesting new data. There are some potential concerns that need to be clarified.

We thank the Reviewer for the feedback and helpful comments. We addressed them accordingly. The line numbers indicated in this reply refer to the numbers of the attached manuscript .pdf with highlighted changes in red.

First, the authors make a puzzling choice of analysis for differentiating neural representations of different stimuli. A rather standard way, by now, of discriminating different stimuli is to do linear (or nonlinear) decoding analysis, which compares population responses for different stimuli. One needs to compare across stimuli – here the authors seem to be comparing distances with respect to the baseline for each of the stimuli (effectively comparing some sort of an amplitude metric). This could be problematic, since the trajectory of the familiar mouse could be very different from the neural population trajectory of other mice, but essentially trace a path that is similar in the sort of Euclidean measure the authors use (for example, think of an ellipse that is exactly mirror image in the 3-dimensional PC space they show – the trajectories are very different, but the Euclidean measure might

look the same). I confess that it's unclear whether the authors do what I think they are doing, so if I am mistaken, it might be worth clarifying the methods in detail in the manuscript.

We thank the reviewer for this clarifying feedback. We now changed the description to make the approach clearer and consequently use separate analyses for two components of interest, namely individual identity and (acquired) shared features.

Firstly, as suggested by the reviewer, we added a linear decoding analysis to show the individual identity component (presented in a receiver subject by diverging unit responses in line with distributed cortical odor identity coding described e.g. in (Stettler and Axel, 2009; Bolding and Franks, 2017) and this manuscript (Fig. 1d)). A linear classifier is trained on the cortical population vectors, which contain the activation pattern in response to an odor. It reliably predicted the identity of a trial which was left out from the training dataset (Fig. 1e).

We added to the results section l. 135:

“Firstly, we probed if the identities of unfamiliar individuals can be discriminated from the activation pattern in the neuronal population response. Indeed, odor signatures emitted by individuals were encoded by diverging activation patterns in cortical neurons (Fig. 1d, Supplementary Figs. 2c, 3) as also previously observed for identity encoding of non-social odors in the piriform cortex (Stettler and Axel, 2009; Bolding and Franks, 2017). Consistently, a linear classifier reliably predicted the identity of the different odors from the neuronal activation pattern (Fig. 1e; Fisher’s exact test against classifier trained with shuffled labels; AON: $p < 0.0001$, pPC: $p < 0.0001$, LEC: $p < 0.0001$).”

And provide the new figure panel:

Fig. 1. Cortical populations encode identity of conspecifics. Representation of social odors. e The confusion matrices of linear decoders, which were trained to predict the odor identity of a single trial from the neuronal population activity, shows high accuracy in AON, pPC and LEC. Prediction accuracy was determined on trials that were not included in the training dataset.

And we added to the method section l. 702:

“**Population analysis: odor response classifier.** We evaluated odor discriminability by measuring the prediction accuracy of a linear classifier. For each brain region separately, we trained multiclass models using support vector machines on the odor presentation period (0 to +1 s relative to odor

onset) of the global spike count population vectors (concatenated across sessions and z-scored within unit). Prediction accuracy was determined using leave-one-out cross-validation, and thus by comparing to the true label the predicted label of a population response vector which was not included in the training dataset. Confusion matrices were plotted for visualization of the classifier performance. Odor discriminability was tested statistically by comparing against the classification accuracy of a classifier which was trained with shuffled labels using Fisher's exact test."

Secondly, we also added a schematic in Fig. 1c. While an individual's identity is reflected in different orientations of the population vectors (i.e., the cortical activation pattern), we hypothesized that the familiarity component is encoded in the amplitude of the population response, which can be quantified by the Euclidean distance from baseline activation (Fig. 1c). This relates to a working model that has also been used in visual cortex (added to the Discussion l. 382: "Scaling of responses has also been observed in primates for salience coding of visual objects (Jaegle et al., 2019)."). Thus, two unfamiliar individuals might have different orientations of the vectors because their unique body odor compositions activate different cortical neurons, but two unfamiliar individuals yield the same Euclidean distance from baseline. When an animal becomes familiar, the Euclidean distance from baseline becomes larger for the familiar animal as the overall amplitude of the population response increases.

We rephrased the explanation of the approach accordingly in the results section starting l. 142:

"Secondly, the encoding of features like salience or familiarity differentiating two groups of otherwise very similar odor emitters should reflect in the amplitude of the population response. The amplitude of the population response can be quantified using the Euclidean distance from the baseline population vector (Fig. 1c). Consequently, unfamiliar conspecifics that share also other features like strain and sex should not differ in the amplitude of the population response. We probed this on the population vectors of the single-unit spike counts concatenated across emitters."

And provide the new schematic in Fig. 1c:

Fig. 1. Cortical populations encode identity of conspecifics. Representation of social odors. c The population vectors encode two components. Firstly, they encode the individual identity of an odor in their orientation, which stems from differential cortical activation patterns. Secondly, we hypothesize that they encode features like familiarity in their overall response amplitude, which can be quantified using the Euclidean distance from baseline.

Thirdly, we decided to **remove the plots of the PCA trajectories** from the manuscript. In the 3D reduced space of neuronal trajectories, changes in intensity of response are indistinguishable from changes in identity of responsive units. In the manuscript, we try to distinguish between these two components; for this purpose, the population trajectories may not be useful and rather confuse the reader.

We removed PCA trajectories from Figs. 1, 2, 3, 5, and Supplementary Figs. 2, 4.

In summary, the revised Fig. 1 shall now help explain better the approach and show that the individual identity of the body odors is reflected in the activation patterns of the population vectors, while the amplitude of the pooled population response (quantified by the Euclidean distance from baseline) can discriminate features in the odor source (mouse, food, flower). Indeed, the amplitude of the pooled population response does not differ for two unknown mice of the same genetic background (C57BL/6) and sex.

There's also some concern about the use of a very small number of distinct stimuli. This may be inescapable for the kind of experiments done here, however using just two individuals from the same strain and another from the strain along with just two other general odors, seems rather marginal. It would be useful to see some discussion (and reassurance) from the authors whether the results can be biased by such small number of stimuli.

We thank the reviewer for raising this clarifying point. Maybe in the initial version of the manuscript, our specific goal was not sufficiently expressed. In Fig. 1, we do not intend to identify object class features. Clustering of odors into classes has been examined for instance by the Datta and other labs. In Fig. 1, we aim to establish the social odor boxes and the behavior of the population response vectors for identity and feature coding.

Further, the total number of trials per session is limited during which animals remain comparably alert and therefore the system is in a comparable state (keeping attention-related variance under control). At the same time, we aimed for sufficient power given trial-by-trial variance in cortical odor responses. Given these restraints, the goal of this first experiment was to establish the approach of presenting fresh volatile body odors by showing: Firstly, that social body odors elicit stable neural and physiological responses (Fig. 1b, Supplementary Figs. 2b, 5d, g). Secondly, that individuals can be discriminated based on the neuronal responses to their body odors (Fig. 1e). And lastly, the principle that the pooled amplitude of the population responses can vary for odors that vastly differ like food, flower or mice, however, that the pooled amplitude does not differ between two unfamiliar odors of the male mice of the same genetic strain (Fig. 1f, g). Importantly, the population vectors were concatenated across sessions with different emitter individuals to prevent selection bias in the examined responses.

To make the rationale of Fig. 1 clear, we added to the result section I. 132: **“The first experiment**

served to establish the social odor representation approach and describe the difference between stimulus identity and feature encoding (Fig. 1c).”

And in the methods section l. 535: “Trial numbers were kept sufficiently high per odor to allow for statistical testing considering trial-by-trial noise in cortical responses. The trials in a session were limited to a number at which the animals did not become drowsy.”

Specific Comments:

Line 126: It’s not clear how to match the trajectories in Fig 1ef, with the results in fig 1g. For example, the differences between blue and magenta trajectories are minimal, but yet the p values in Fig 1g are very low (black squares).

As discussed above, in the 3D reduced space of neuronal trajectories, changes in intensity of response are indistinguishable from changes in identity coding patterns of units responses, and we therefore replaced the trajectories with the suggested linear decoding analysis throughout the manuscript. The differences between the C57BL/6 and CD1 response evolutions in Fig. 1f look small, however since we employ a paired test to compare across stimuli, the consistent small difference in population response amplitude results in a small p-value.

To better convey the main idea of the analysis in Fig. 1, we now focus first on the identity coding (Fig. 1 c-e) and then establish feature coding (Fig. 1f-g). We then focus on the main point in Fig. 1(g), that is that the post-hoc comparison between the two unfamiliar C57BL/6 odor responses is not significant, now illustrated by box plots with lines connecting the paired values. Consequently, we transferred the p-value heatmaps to Supplementary Fig. 2 as they are additional validation steps when demonstrating the principles of the approach and analysis.

Line 132: would be good to see the original sniff rates – I think this is in Supp Fig 5, but perhaps the authors can make it clear?

We have added to the original sniff traces a box plot of the baseline sniff rates (without stimulus presentation) to Supplementary Fig. 5a.

Line 138: The number of distinct stimuli is still low (5), and it's hard to be sure about chance effects - that is, by chance the 2 same-strain individuals happen to be similar, but if you test many other pairs there might be differences. While I understand the difficulty of getting many animals, I worry about selection bias and the authors could perhaps assure the reader that this is not an issue?

We thank the reviewer for raising this question. Using only one or very few emitter animals could indeed confound the result because observed differences (or the lack thereof) could be due to differences in the odor signature of that single individual. For this reason, we had used a pool of emitter animals from

which different combinations were drawn for every session, avoiding the repetition of unique combinations (the new Supplementary Tables 1 and 2 show the number of emitters used for each experiment). Also, we had a criterion that within one experiment, the emitter that served as familiar in a session with one receiver, served also as a novel mouse in a session with another receiver. By then pooling the data across sessions, we can study the encoding of attributes that are shared by all emitter animals from a pool.

We have now highlighted this information better in the Methods section I. 462: “

We used a pool of emitter mice (Supplementary Table 1 and 2). Emitter mice were either male C57BL/6J mice or male CD1 mice from Charles River Laboratories. Note that different recording condition groups were run with partially overlapping emitter animals. As an additional constraint, any one emitter mouse served only once per receiver mouse. The number of receiver and emitter mice in each experiment is given in Supplementary Tables 1 and 2.”

And I. 530: “The combination of individual emitter mice was randomly permuted to avoid repetition of unique combinations. A specific emitter was presented to each receiver mouse only once across sessions. The number of emitter animals in each cohort is given in Supplementary Table 1.”

And I. 543: “To balance their representation, one emitter mouse served in an experiment both as a ‘novel’ and as a ‘familiar’ for different receiver mice. The number of emitter animals in each cohort is given in Supplementary Table 2.”

And I. 189 of the Results section: “Each emitter mouse contributed both as ‘familiar’ and ‘novel’ to different receiver mice in balanced numbers of session (for number of mice see Supplementary Table 2). Receiver mice participated in at maximum one session per week.”

And provide the new Supplementary Tables 1 and 2:

Supplementary Table 1, related to Fig. 1. Number of emitter mice used in the different cohorts.

This table contains the number of unique emitter mice used in the different experiments. Note that different recording condition groups were run with partially overlapping emitter animals. This pool of animals was used to draw different combinations for every session, avoiding the repetition of unique combinations.

	number of receiver animals	size of C57BL/6 #1 emitter pool	size of C57BL/6 #2 emitter pool	size of CD1 emitter pool
AON	8	4	3	2
pPC/LEC	11	5	4	4
MOB	8	3	3	2
VTA	3	3	3	3

Supplementary Table 2, related to Figs. 2-5. Number of emitter mice used in the different cohorts.

This table contains the number of unique emitter mice used in the different experiments. Note that different recording condition groups were run with partially overlapping emitter animals. This pool of animals was used to draw different combinations for every session, avoiding the repetition of unique combinations. The same emitter served as familiar animal in a session with one receiver and as novel animal in another session with a different receiver.

	number of receiver animals	size of C57BL/6 emitter pool
AON ChR2 ^{OXT/PVN}	8	15
pPC/LEC	11	5
VTA	3	5
MOB	8	15
photometry	9	10
OXTR ^{AON}	12	10

Line 161 and elsewhere: Could the higher firing rate to familiar mouse be simply due to higher sniff freq, as shown in Fig 2b?

We thank the reviewer for this important comment. We have addressed this possibility by adding a trial-by-trial correlation of sniff rate vs population response amplitude to Supplementary Fig. 9b-c. We used all 8 sessions from animals with simultaneous sniff and AON recordings. For every session, we plotted the sniff frequency during a single trial versus the population response amplitude (Euclidean distance from baseline) of the same trial and calculated Spearman's correlation coefficient. The number of significant correlation coefficients among the sessions is indicated in the figure. We find that single-trial sniff rate and response amplitude are not correlated, supporting that the stronger neuronal response was not caused by the higher sniff rate.

This is now described in the results l. 205: **“The stronger neuronal response in the AON to the familiar odor was however not explained by the sniff rate modulation in a trial-by-trial correlation (Supplementary Fig. 9b-c).”**

and method section l. 733: **“Correlation analysis: neural response vs. sniff response. To investigate whether the population response amplitude was mediated by the sniff frequency, we performed a trial-by-trial correlation of sniff frequencies versus the Euclidean distance from baseline. We included all AON sessions with simultaneous sniff recordings, as the per-session unit count was highest in AON among regions and sufficient to build population vectors for single sessions. For every session, we then computed the Euclidean distances from baseline from the single-session population response vectors and correlated the mean distance during odor presentation (0 to +1 s relative to odor onset) to the average sniff frequency of the trial (0 to +1 s relative to odor onset) using Spearman's correlation coefficient.”**

Supplementary Fig. 9, related to Fig. 2. Intra-individual correlations of behavioral readouts and neuronal activity. b-c For all AON sessions with simultaneous sniff recordings, we plotted the sniff frequency during (b) familiar and (c) novel odor presentations versus the mean Euclidean distance from baseline of that trial (0 to +1s relative to odor onset). We computed the trial-by-trial correlation for every session individually using Spearman's correlation coefficient and tested the significance of that correlation coefficient. We find no significant association of trial-by-trial sniff frequencies versus population response amplitude (p-values for every session indicated in the figure). Note that the reference lines for the correlation within the sessions was plotted for illustration without considering outliers, while the correlation coefficient was tested for significance considering all trials.

Line 186: This term coupling seems to always imply (in a lot of the literature) some sort of physical connection - but really what the data show is that there is greater coherence in some frequency band. Why not use a neutral, more justified terminology like "coherence between x and y is increased"? But this is not a demand by any means – but a point of view that may be shared by a lot of the readership.

We agree and replaced the term “coupling” by the more accurate “phase-synchronization” (l. 236, 241, 243 and 402) and figure legends and titles of Fig. 3, Supplementary Figs. 10, 11.

Line 245: This observation about AON being the main olfactory region to get activated by OXT stimulation seems very interesting, and I urge the authors to make this more prominent!

We thank the reviewer for this suggestion and have emphasized this finding in l. 285: “Outside of the hypothalamus, the AON showed the strongest BOLD activation upon OXT neuron excitation (T = 5.18; Fig. 4c), which was also the only primary olfactory region significantly activated.”

Figure 1g: To reiterate an earlier point made above: It's not clear that mean Euclidean distance is the best metric. For example, two different trajectories that are widely divergent can have similar Euclidean distances from baseline. In fact, the difference between the green and blue trajectories in panel e is very large, but if we look at panel f, they look much closer.

As the reviewer made clear to us in an earlier point, the trajectory of time-embedded responses may not be the most helpful representation because it is a mixed representation of the direction and extent of the

population responses. This repeats here. We have therefore replaced all trajectories by the linear decoding analysis and separately presented the measure of the Euclidian distance from baseline.

We used this decoder approach now also in the new Suppl Fig. 15h to show that OXTR KO mice, even though not learning familiarity, still can decode identity normally. We have added to the results, l. 329: “Note, that the identity of the odor could still be reliably predicted from the neuronal activation using a linear decoder, supporting the notion that odor discrimination was not affected (Supplementary Fig. 15h; Fisher’s exact test against classifier trained with shuffled labels: $p < 0.0001$) (Oettl et al., 2016).”

Reviewer #4 (Remarks to the Author):

In their manuscript the authors examined the neural basis for social recognition memory testing the hypothesis whether familiar odors from other mice are encoded by habituation memory or by gaining more distinct representations that are reinforced by social interactions through experience dependent plasticity. In the latter case the conspecific should trigger "stronger and more differentiated neuronal responses".

To this purpose they recorded multiple single unit activities in different nuclei of male C57BL/6J OXT-Cre transgenic mice: the anterior olfactory nucleus (AON, 8 animals), the posterior piriform cortex (pPC, n=11) and the lateral entorhinal cortex (LEC, n=11). In addition they recorded for some experiments in the main olfactory bulb (MOB, 8 animals) and the ventral tegmental area (VTA, 3 animals). Single unit identification was achieved by using tetrodes arranged in multiple arrays (2x 4 or 7), depending on the nucleus.

The mice were head-fixed for recording spiking activities before being repeatedly exposed for 1 second to five different odors: peanut butter, ylang ylang, and odors from same sex juvenile mice: one mice from CD1 (a different strain), and two C57BL/6J mice (#1 and #2).

In a second series of experiments the mice were first exposed for five minute interaction with one of two C57BL/6J mice so as to familiarize the experimental mice with this odor and test whether the encoding for this odor would have changed in one of these nuclei.

In a third series of experiments the authors looked at the effects of oxytocin receptor activation or inhibition on the representations of the encodings.

This is an interesting series of experiments of how encoding of familiar and unfamiliar mice may occur by neuronal activity in different nuclei and the role of oxytocin signaling in this encoding.

However, there are a number of questions that arise on this work.

We thank the Reviewer for his feedback and addressed it point-by-point. The line numbers indicated in this reply refer to the numbers of the attached manuscript .pdf with highlighted changes in red.

Major points

1. Representation of social odors.

Line 138: "Importantly, the examined regions and physiological readouts pooled across individuals do not differentiate among unfamiliar C57BL/6 males under our conditions. The social odor assay is thus suited to study if and how familiarity memory is implemented in the neuronal population code."

I am not sure if this is fully correct. The authors tested the lengths of the Euclidian distance of each of the vector to baseline, but not the orientation of the vectors. Although the PCA analyses show similar traces for the AON, in the pPC and LEC, the representations of the two C57BL6 mice #1 and #2 reveal rather distinct traces. The authors should test for the two unfamiliar mice of similar strain across all nuclei the Euclidian distance between the endpoints of the vectors encoding for different non-familiar

animals and compare the orientations of the different vectors.

In all, it seems in fact most surprising that in the nuclei they tested, they found differential representations of ylang ylang and peanut butter, but no differential representations of different individuals. It is important to fully nail this down, because this point has major implication for the changes that occur upon familiarization of one of the two animals.

We thank the reviewer for raising this point. We agree that the wording in our sentence was misleading. We rephrased in the Results section, starting line 175: "Importantly, **the Euclidean distance from baseline of the examined regions and the physiological readouts pooled across individuals do not differentiate under our conditions among unfamiliar male mice of the same C57BL/6J genetic inbred strain independent of their age ranging from adolescent to adult.** The social odor assay is thus suited to study **if and how** familiarity memory **is implemented** in the neuronal population code."

This means that the amplitude of the population response does not differ between unknown C57BL/6J individuals in any of the regions investigated. We do, however, expect that a differentiation between individuals is still possible by looking at the specific activation patterns elicited by the odorants. In fact, our hypothesis is that while an individual's identity is reflected in the specific pattern of activated units, and therefore in the orientation of the population vector, familiarity is encoded in the amplitude of the population response, which can be quantified by the Euclidean distance from baseline (Fig. 1c). Conceptually, this reflects the principles of memory engrams. This relates also to a working model that has been also used in visual cortex (added to the Discussion l. 382: "**Scaling of responses has also been observed in primates for salience coding of visual objects (Jaegle et al., 2019).** "). We have adapted the schematic in Fig. 1c to describe the rationale for using the Euclidean distance from baseline as a metric for population response amplitude.

To better describe this point, we now **added Fig. 1e** with a classifier trained to distinguish odor identity in the three recorded regions. As shown in the new Fig. 1e, AON, pPC, and LEC can differentiate all odorants with high accuracy. As expected, this was also the case between the odors of unfamiliar individuals. (More details about the new classifier analysis can be found in the first point of the reply to reviewer #3).

Finally, we decided to **remove the plots of the PCA trajectories** from the manuscript. In the 3D reduced space of neuronal trajectories, changes in intensity of response are indistinguishable from changes in identity of responsive units. In the manuscript, we try to distinguish between these two components; for this purpose, the population trajectories may not be useful and rather confuse the reader.

In summary, the revised Fig. 1 now shows that the individual identity of the body odors is reflected in the composition of the response pattern (as captured by the classifier analysis, Fig. 1e), but not in the response intensity (as quantified by the Euclidean distance from baseline, Fig. 1g).

Together, we made the following changes:

- We removed the PCA trajectories throughout the manuscript.
- We added a schematic in Fig. 1c to illustrate the approach.
- We added a classifier approach to Fig. 1 to show that identity coding is preserved in the neuronal activity. We also tested with the same approach that such identity coding is preserved in the

AON of OXT mutants, even though they do not discriminate familiar from novel C57BL/6 (shown in Supplementary Fig. 15h).

- In the main Fig. 1 we now focus on the observation that the post-hoc comparisons between the unfamiliar mice from the same genetic background are not significant. We now report this as boxplots showing only the post-hoc values associated with the comparison between the two unfamiliar mice from the same genetic background, and moved the p-value heatmaps of the full analysis now to Supplementary Fig. 2.
- We changed the description in the results (starting l. 135) accordingly: “**Firstly, we probed if the identities of unfamiliar individuals can be discriminated from the activation pattern in the neuronal population response. Indeed, odor signatures emitted by individuals were encoded by diverging activation patterns in cortical neurons (Fig. 1d, Supplementary Figs. 2c, 3) as also previously observed for identity encoding of non-social odors in the piriform cortex (Stettler and Axel, 2009; Bolding and Franks, 2017). Consistently, a linear classifier reliably predicted the identity of the different odors from the neuronal activation pattern (Fig. 1e; Fisher’s exact test against classifier trained with shuffled labels; AON: $p < 0.0001$, pPC: $p < 0.0001$, LEC: $p < 0.0001$). Secondly, the encoding of features like salience or familiarity differentiating two groups of otherwise very similar odor emitters should reflect in the amplitude of the population response. The amplitude of the population response can be quantified using the Euclidean distance from the baseline population vector (Fig. 1c). Consequently, unfamiliar conspecifics that share also other features like strain and sex should not differ in the amplitude of the population response. We probed this on the population vectors of the single-unit spike counts concatenated across emitters. Indeed, two unknown C57BL/6 odors produced responses with a similar Euclidean distance from baseline (Fig. 1f-g). This finding applied to all three olfactory cortices (Fig. 1f-g, Supplementary Fig. 2e-f) as well as to two sub-cortical regions, the MOB and the VTA (see Supplementary Fig. 4). However, the population response amplitude differed between some of the odors that originate from food, flower or mice and are expected to differ in their features (Fig. 1f, Supplementary Fig. 2d-h).**”

The revised sections of Fig. 1:

Fig. 1. Cortical populations encode identity of conspecifics. Representation of social odors.

c The population vectors encode two components. Firstly, they encode the individual identity of an odor in their orientation, which stems from differential cortical activation patterns. Secondly, we hypothesize that they encode features like familiarity in their overall response amplitude, which can be quantified using the Euclidean distance from baseline. Population vectors were computed for each brain region by pooling units across sessions and matching trials according to trial type. Temporally embedded trajectories and Euclidean distances from baseline were computed from the population vectors. **e** The first three principal components of the time-embedded average population responses to the different odors. The trajectories reveal similar responses to social odors and diverging responses between social and non-social odors (asterisks mark odor onset, circles mark odor offset, arrows indicate the temporal evolution) ($n = 20$ trials per odor). **d** Population responses in a representative experiment with 75 simultaneously recorded neurons from AON responding to the 5 different odorants. Single emitter individuals can be discriminated based on diverging responses in single units. **e** The confusion matrices of linear decoders, which were trained to predict the odor identity of a single trial from the neuronal population activity, shows high accuracy in AON, pPC and LEC. Prediction accuracy was determined on trials, that were not included in the training dataset. **f** The temporal evolution of the Euclidean distance from baseline of the population vector in the AON (mean \pm SEM, $n = 20$ trials per odor). Gray bar represents odor duration. **g** The mean Euclidean distance from baseline was compared for the different odors (0 to +1 s relative to odor onset), indicating differentiation in the population response magnitude (repeated-measures one-way ANOVA with post-hoc Tukey's test for multiple comparisons). None of the recorded cortices showed significant differences between the two unfamiliar mice from the same genetic background (see Supplementary Fig. 2d,g,h for all pairwise comparison results). P-values of the post-hoc tests were visualized as a heatmap. **h** The sniff frequency response also did not differ between the two C57BL/6 mice. P-values of pairwise comparisons of sniff frequency responses between odors (repeated-measures one-way ANOVA with post-hoc Tukey's test for multiple comparisons; $n = 13$ animals with 1 session each; see Supplementary Fig. 5c for all pairwise comparison results).

2. Neuronal encoding of familiarity

It is a bit surprising that the authors make a cross-comparison of the representation of two different mice, one with which the experimental animal has become familiar, and one with which it has not become familiar. They then again compare vector length by Euclidian distance to baseline. If indeed the training of a familiar mouse leads to a stronger representation (potentiation of responses) by the originally encoding neurons, it would be more straight forward to longitudinally test in the

experimental mouse what changes occur in the representation of the demonstrator mouse before and after familiarization in a longitudinal comparison.

Indeed, it seems rather surprising that the length of the vector (the euclidian distance between baseline and stimulus exposure) for the familiar stimulus would be longer in length than the unfamiliar stimulus as both are likely to have completely differently oriented representations in this multidimensional vector space. On the other hand, if indeed the social recognition memory is represented by (as the authors hypothesize in line 83) "stronger neuronal responses", the representation in the multidimensional space should stay in similar direction, but with a longer Euclidian length of the population vector from the baseline. Thus, the crucial experiment to address their hypothesis would be to make longitudinal comparisons between neuronal representations before and after familiarization with the same mouse.

When designing the experiment, we had discussed different possibilities including a longitudinal design. However, following an in-depth discussion, we would like to summarize here the main reason why it unfortunately cannot be done. In a longitudinal experiment, we would first have to perform the head-fixed recording before familiarization, then detach the animal from the recording setup, perform the freely moving interaction and return to the head-fixed condition to perform another recording session. While units are stable within one session, breaks (like the interaction) and reconnection can result in sampling or spike clustering of different unit sets. We tried this approach in other experiments and found exactly such problems. This eventually prevents comparing the activity of the same set of units pre and post familiarization, and thus conclusions on longitudinal changes due to familiarity. Therefore, the advantage of running the "pre" experiment together with the "post" experiment (that is our real test) is lost.

The idea of comparing the average response amplitudes to the odorants with and without prior familiarization across mice solves instead this problem. As outlined by the reviewer, and now better exemplified in the new Fig. 1c, 1) the familiar and unfamiliar stimulus will have differently oriented representations in the multidimensional vector space, and 2) familiarity will increase the Euclidean distance of the population vector from the baseline. Accordingly, our analysis showed that 1) the identity of the conspecifics is encoded in the activation pattern of single unit responses that can be reliably predicted by a linear decoder, Fig. 1e (from vectors that have on average the same length, Fig. 1g), and that 2) familiarity induces a stronger neuronal response (as evidenced by a higher Euclidean distance from baseline after familiarization, Fig. 1f-g and Fig. 2). Even if tested on different neurons, our analysis establishes that the amplitude of the population response only differs between C57BL/6 mice if one of them is familiar.

We added a clarifying comment to l. 558: "A longitudinal experimental design for this question is not possible with tetrode recordings, because we cannot be certain that the same units are included in the population, when switching back-and-forth between head-fixed and freely-moving conditions."

Other points:

1. To obtain a better appreciation of "ground-truth" differences in representations of single unit

responses to the odor in the different nuclei the supplementary figures 3,4,7, 9b, 11b,13e&h are most useful. In particular, to gain an understanding concerning the differences in encodings of different C57BL6 mice (both unfamiliar vs familiar), it is important to sort the activity of these units according to one of the C57BL6 mice. In Supplementary Figure 3, for example, unit activity seems to have been sorted according to different single unit responses to the odor ylang ylang. To gain an understanding of the differences between C57BL6#1 and #2, it would be important to add a similar figure, but with all units sorted according to the responses of the units to the odor of C57BL6#1. This would allow a better appreciation of the level to which the responses to C57BL6#2 would be different (or similar) and give us a more intuitive understanding of why there is so much overlap in the PCA representation by these 313 units of C57BL6#1 and #2 in figure 1e. Idem for supplementary figures 4a&b.

We thank the reviewer this suggestion and updated the sorting of unit responses in Supplementary Figs. 3 and 4 to a sorting by the response to C57BL6 #1. We updated the figure legends accordingly.

A similar questions arises for the representation presented in Supplementary figure 7. Here, the authors would like to claim that the representation of the familiar mouse (versus the novel mouse) occurs through stronger single units responses (and thus a longer Euclidian distance of the population vector to the baseline). At first glance this may indeed seem to be the case (there are more, and more intense, and longer red traces of units in the familiar versus the novel mouse). However, this impression may also be caused by the sorting of the units according to the familiar mouse. The authors seem to be aware of this, and have thus also included a "resorted" column on the far right, that seems to show the sorting of the units according to the "novel mouse". However, one crucial column is lacking in this representation, namely the column of the "familiar mouse" when its units have also been sorted according to the "novel mouse". To appreciate to what extent this visual impression of a stronger representation of the familiar mouse persists, it is crucial to also show this fourth column of the familiar mouse when the units have been resorted according to the novel mouse. (idem for supplementary figures 11b and 13 e and h).

As suggested, we added the additional column of unit responses to the familiar odor, sorted by the responses to the novel odor, now in Supplementary Figs. 11, 13, 15 (please note that with the added data the numbers of Supplementary Figs. changed).

Fig. 2c. and Supplement. The absolute differences in spiking activity of units during exposure to familiar versus novel mice seems very small and not very convincing. The significant differences that the statistical tests come out with may possibly be caused by which test is chosen and how it is applied. For that matter, one wonders to what extent these differences in unit activity per individual animal show a correlation with individual differences in behavioral measurements that have been recorded.

Indeed, supplementary figure 6a shows quite some variability in sniffing frequencies across trials for the (8?) mice that have been tested. These differences in behavior should correlate with differences in

spiking activities of neurons in the AON, pPC and LEC that the authors measure across these different 8 mice.

Did these two parameters (sniffing frequency per animal versus differences in single unit activity per animal) show a correlation? Can the authors provide such a plot (and a table) showing the differences in behavior per animal and the number of units per animal that responded significantly differently between exposure to novel versus familiar rats?

Can the authors correlate the differences in behavioral responses per mouse (e.g. sniffing frequencies) with the differences in single unit activities per mouse (Euclidian lengths of the population vectors). Such a correlation could strengthen the relationship between encoding of familiarity responses by single unit activity. (or, for example, normalize the differences in unit responses on the differences in sniffing behavior)

We thank the reviewer for raising the question on correlations between sniff and neuronal responses in social recognition. The sniff was simultaneously recorded for a subset of mice implanted with recording tetrodes in the AON ($n = 8$ mice in Fig. 2b). The other data points are from sniff recordings without electrophysiology. Mice implanted with recording electrodes in the pPC/LEC had no sniff recording.

Using this AON data sample, we performed two analyses:

1. We tested the prediction that the stronger the memory, the more it reflects in both session-wise average sniff and neuronal responses.
2. To control that the observed neuronal effect of familiarity is not merely a consequence of a stronger sniff modulation, we further tested whether a trial-by-trial variation in sniff and rate response exists.

Observations:

Ad 1. The correlation of difference in the per-mouse-average Euclidean distance from baseline vs. sniff difference shows a positive correlation coefficient. This supports that per-subject-average sniff and neuronal responses are associated, consistent with the idea that familiarity is observed in either readout. We added this to Supplementary Fig. 9d-e and adapted the figure legend:

Supplementary Fig. 9, related to Fig. 2. Correlations of behavioral readouts and neuronal activity.

d-e The correlation between memory strength (difference in (d) Euclidean distance from baseline and (e) average firing rate between familiar and novel) and sniff difference shows a positive correlation coefficient. This supports that per subject average sniff and neuronal responses correlated, consistent with the idea that familiarity is observed in either readout. (n = 8 sessions, same as b-c).

We added to the results l. 206: “We tested the prediction that the stronger the memory, the more it reflects in both session-wise average sniff and neuronal responses. Indeed, the difference in mean firing rate response or Euclidean distance from baseline correlated positively with simultaneously recorded sniff response differences between familiar and novel emitters (Supplementary Fig. 9d-e).”

And to the methods section, l. 742: “To investigate if the memory strength was associated with the averaged sniff response during the recognition, we computed Spearman’s correlation coefficient for the single-session average difference in Euclidean distance from baseline between familiar and novel responses or the mean difference in firing rate response (0 to +1 s relative to odor onset) versus the difference in sniff frequency between familiar and novel responses. We included all AON sessions with simultaneous sniff recording.”

Ad 2. At the single-trial level, the sniff rate and population response amplitude were not correlated, supporting the notion that the increased sniff rate does not explain the strengthened population response by itself (Supplementary Fig. 9b-c). For every session, we plotted the sniff frequency during a single trial versus the population response amplitude (Euclidean distance from baseline) of the same trial and calculated Spearman’s correlation coefficient. The number (in this case zero) of significant correlation coefficients among the sessions is indicated in the figure.

This is now described in the results l. 205: “The stronger neuronal response in the AON to the familiar odor was however not explained by the sniff rate modulation in a trial-by-trial correlation (Supplementary Fig. 9b-c).”

and method section l. 733: “**Correlation analysis: neural response vs. sniff response.** To investigate whether the population response amplitude was mediated by the sniff frequency, we performed a trial-by-trial correlation of sniff frequencies versus the Euclidean distance from baseline. We included all AON sessions with simultaneous sniff recordings, as the per-session unit count was highest in AON among regions and sufficient to build population vectors for single sessions. For every session, we then computed the Euclidean distances from baseline from the single-session population response vectors and correlated the mean distance during odor presentation (0 to +1 s relative to odor onset) to the average sniff frequency of the trial (0 to +1 s relative to odor onset) using Spearman’s correlation coefficient.”

In addition to these two points, we further provide evidence supporting that the social interaction during the familiarization causes the reinforced cortical representations by including in the manuscript additional data and analyses (Fig. 2d, Supplementary Fig. 9a). For recordings from the AON, we manually annotated the freely-moving exploration and correlated the number of interaction bouts or the total interaction time to the difference in population response amplitude between familiar and novel odors during the recognition phase. We found that the difference in population response amplitude (familiar – novel) positively correlated with the number of exploration events (Fig. 2d). Interestingly, the sheer time spent in close contact did not correlate with the population response amplitude (Supplementary Fig. 9a).

We added to the Results section, l. 202: “The expressed neuronal memory (familiar – novel) positively correlated with the number of sampling events during the exploration phase (Fig. 2d), but not the sheer total contact duration (Supplementary Fig. 9a). This may hint on that social memory formation is promoted by repeated sampling.”

and in the method section l. 548: “The interaction was recorded on video using a Sony FDR-X1000V camera with a frame rate of 30 fps. Exploration behavior was later annotated manually using the BORIS software (Friard and Gamba, 2016). Frames where the experimental subject’s nose touched the conspecific were labelled as interaction. Two subsequent interaction bouts were counted as individual events if they were separated by more than one second.”

and l. 748: “**Correlation analysis: neural response vs. exploration behavior.** To investigate if the memory strength was associated with the intensity of exploratory behavior during the familiarization period, we performed a correlation of the number of interaction bouts or the investigation time versus the difference in Euclidean distance from baseline between the familiar and novel odor. We included all AON sessions as the per unit session count was sufficient here to build single-session population vectors. For every session, we computed the mean difference in Euclidean distance from baseline between familiar and novel odor responses and correlated that to the number of interaction bouts or to the total interaction time using Spearman’s correlation coefficient.”

The new Figure 2d panel:

Fig. 2. Neural coding of familiarity.

d The correlation of the memory strength (difference in Euclidean distance from baseline between familiar and novel) and the number of interaction bouts during the freely-moving familiarization period shows a positive association ($n = 8$ animals, 3 sessions each; same data as c).

Finally, related to the reviewer’s question on the number of units that responded significantly differently, we provide now a more detailed analysis based on the previous finding that the larger fraction of units significantly responding with excitation in the AON is non-selective (i.e. responds to both familiar and novel odors). We show that the stronger response to the familiar odor in these non-selective odor excited units contributes to the net effect (new Supplementary Fig. 8b-e).

We removed the statement that more units respond to the familiar odor in AON, as it is redundant. Weakly responsive units may be just below our statistical detection threshold. Thus, stronger responses to the familiar odor in significantly responding units and a slight increase in the number of significant

units to the familiar odor shows per se the same phenomenon. It is however statistically unambiguous to focus the comparison between familiar and novel odors only on significant responses. Consistently, we show that when including all units in the analysis (also non-responding units according to the significance criteria), the firing rate response to familiar is stronger (Supplementary Fig. 8a), but also when only selecting significantly odor-excited AON units (Supplementary Fig. 8b-e).

We adapted the results section accordingly, l. 212: “AON units excited by both conspecifics recruited by the familiar mouse had stronger mean responses than selective units and also showed stronger firing rate responses to the familiar odor in AON and pPC as compared to the smell of the novel mouse (Supplementary Fig. 8e).”

3. Idem for the modulation of the oxytocin responses, be it by potentiation of oxytocin or inhibition with antagonist: The authors seem to show only single unit responses but none of the behavioral responses of the individual animals.

The sniff responses of the individual animals from the OXT intervention were already present but maybe a bit hidden in Supplementary Figs. now 13i (previously S11i) (sniff frequency in response to familiar and novel odors for the condition with boosted OXT during initial interaction) and Supplementary Fig. now 14c (previously 12c) (difference of sniff frequencies between familiar and novel odor responses for the OXTR^{AON} and matched control group).

We have adapted the results section to better highlight this, l. 316: “Also, the sniff frequency was higher in response to the familiar odor than for the novel odor (Supplementary Fig. 13i).”

Minor points

The structure of the introduction seems to lack some logic. In particular, the authors conclude in line 75 that these findings generate the prediction that the plasticity in the neuronal representation of conspecifics should be modulated bi-directionally". Do they mean both potentiation and depotentiation? If so, could they clearly state this and also on which arguments they conclude that this should be in both directions? (from the preceding it only seems that OT increases behavioral SRM.

We thank the reviewer for the comment. We now placed the sentence so that it follows the reference to the prior studies showing bidirectional modulation of the behavioral expression of SRM through OXT. Oettl et al. showed that optogenetic release of OXT prolongs the duration of SRM, while deletion of OXT receptors from the AON prevents behavioral expression of it (Oettl et al., 2016). To avoid ambiguity we now removed the term “bi-directionally” and directly described our prediction, l. 69: “Boosting OXT release when rodents first meet prolongs the duration of the behavioral SRM, while depletion of OXT receptors in the anterior olfactory nucleus (AON) prevents its behavioral expression (Oettl et al., 2016). These findings suggest that OXT can boost the induction of familiarity memory in the neuronal representation while a lack of OXT action prevents its formation.”

2. Line 78: "There are thus two competing hypotheses how SRM is encoded". From the previous text it is not immediately clear why there would be TWO competing hypotheses. Can the authors explain this argumentation more clearly? Or maybe slightly rephrase, since the premiss does not automatically lead to this conclusion.

We thank the reviewer for highlighting this and modified the introduction accordingly, starting l. 80: "There are ~~thus two~~ competing hypotheses how SRM is encoded in olfactory regions. Repeated exposure to odors can result in adaptation both in behavior and in the neuronal responses of the main olfactory bulb (MOB) (Chaudhury et al., 2010; Kato et al., 2012; Yamada et al., 2017) and cortex (Wilson, 1998). This, together with the fewer spontaneous approaches towards familiar animals, has led to models of social recognition memory as a habituation process (Wilson, 1998; Wilson and Linster, 2008; Linster and Kelsch, 2019).~~The currently prevalent one is that recognition is-~~ It may thus be a habituation memory in terms of an odor familiarization process that results in lower salience of familiar subjects. On the other hand, social interactions are innately rewarding (Dölen et al., 2013b). A competing hypothesis thus states that social interactions reinforce the representations of these familiar animals through experience-dependent plasticity.~~The competing hypothesis states that familiar animals may gain more distinct representations that are reinforced upon social interactions through experience dependent plasticity.~~ In this case, a familiar conspecific should trigger a stronger ~~and more differentiated~~ neuronal response ~~than a novel conspecific~~. We tested the competing hypotheses."

3. Line 82: "In this case, sampling a familiar conspecific should trigger a stronger and more differentiated neuronal responses." This sentence is unclear: "stronger and more differentiated" "Stronger" -where?, "differentiated" - From what? I would say this should depend in which area of the brain one measures, one could imagine: E.g. an upstream area where the responses are more strongly differentiated between familiar and non-familiar conspecific, and a downstream area where there is less response to familiar conspecifics and thus less drive for action.....

We thank the reviewer for pointing this out and clarified our hypothesis statement.

We changed the text to clarify that our main hypothesis focuses on reinforced representations in the olfactory system, l. 80: "There are ~~thus two~~ competing hypotheses how SRM is encoded in olfactory regions."

And in l. 90:

"In this case, a familiar conspecific should trigger a stronger ~~and more differentiated~~ neuronal response ~~than a novel conspecific~~."

4. Line 87: "more distinct population responses" ∅ distinct from what?

We specified this and changed the text in l. 86: "On the other hand, social interactions are innately rewarding (Dölen et al., 2013b). A competing hypothesis thus states that social interactions reinforce the representations of these familiar animals through experience-dependent plasticity. In this case, a familiar conspecific should trigger a stronger ~~and more differentiated~~ neuronal response ~~than a novel conspecific.~~"

5. "The cortical SRM is communicated top-down to earlier sensory circuits" What do the authors mean with "earlier sensory circuits"? What do they mean with "cortical SRM"? A (presumed) representation of the SRM in the AON?, pPC ? MOB?

We thank the reviewer for highlighting this and adapted the sentence in l. 98: "OXT enables the formation of such reinforced representations and ~~the cortical SRM this information is then transmitted communicated~~ top-down ~~from the AON to MOB earlier sensory circuits.~~"

Results:

Line 102: "We presented the odors of two C57BL/6 and one CD1 male (mice, we assume?). To what kind of receiver mouse? (which line, which sex?) And what ages? It would be useful for the reader to receive this information at this point without having to search in the methods section.

We specified this and changed the result section in l. 109: "The olfactometer is set to present repeatedly social and non-social odors for 1 s in a pseudorandomized order every 10-12 s to the ~~male, adult~~ receiver mouse (~~C57BL/6J background, see methods for details on recording cohorts~~) in head-fixed configuration (Fig. 1a). We presented the odors of two ~~male~~ C57BL/6 mice (~~#1: age: P(ostnatal day) 35 to P50; #2: age: P84 to P105~~) and one ~~male~~ CD1 mouse (~~age: P84 to P105~~) and also of peanut butter and a flower for comparison." (C57BL6 #1 now in purple and #2 in blue) and l. 175: "~~the Euclidean distance from baseline of the examined regions and the physiological readouts pooled across individuals do not differentiate under our conditions among unfamiliar male mice of the same C57BL/6J genetic inbred strain independent of age variations ranging from adolescent to adult.~~" We added two Supplementary Tables (1 and 2) which detail the size of the pool of emitter animals used for different recording cohorts (e.g. pPC/LEC). Note that different recording condition groups were run with partially overlapping emitter animals. This pool of animals was used to draw different combinations for every session, avoiding the repetition of unique combinations.

Line 104: "The contributions of different olfactory cortex regions in the coding of SRM are not known." (not clear what is the function of this sentence at this point in the text. Seems to more belong to the introduction.

We thank the reviewer for this comment and removed the sentence from the Results section.

Line 111: "SRM entails two aspects" The two aspects that follow have nothing to do with the "memory of SR", rather with the "social recognition itself "

We thank the reviewer for pointing this out and replaced “SRM” as suggested. We adapted the respective paragraph to clarify the approach and hypothesis tested in this manuscript, l. 122: “~~SRM The process of recognizing a conspecific~~ entails two aspects. Firstly, individual animals need to be ~~identified discriminated~~ by their unique odor signatures. ~~Indeed, odor signatures emitted by individuals were encoded by diverging responses in cortical neurons (Fig. 1c, Supplementary Figs. 2c, 3), similar to the identity encoding of non-social odors in the piriform cortex.~~ Secondly, familiarity with the animal's odor is recalled.”

Line 121 "The population responses to the odors from unknown C57BL/6 displayed a shared trajectory across the cortices (Fig. 1e, Supplementary Fig. 2e)". In fact, the trajectories shown in suppl Fig. 2e (for the pPC and LEC) are very different.

As discussed above, we also replaced these trajectories by separating the analysis into linear decoding analysis and description of the Euclidean distance from baseline. We hence removed the sentence from the Results section.

Line 123: In the AON, the difference between the "CD1" and "ylang ylang" in the first half second is (according to Supplem. Fig. 1d) significant but not anymore in the second half second. However, the Euclidian distance between these two odors is very small in the first half second and larger in the second half second (Fig. 1f). Could the authors verify this significance?

We double checked the significance and confirmed it. The different levels of significance come from two ways of computing the data in the respective panels. Line plot of the temporal evolution of the Euclidean distance from baseline reports the mean \pm SEM across trials with temporal smoothing, while the p-value heatmaps report corrected post-hoc tests from a repeated-measures one-way ANOVA on the unsmoothed data.

To condense Fig. 1 and focus on establishing the approach, we have moved the full p-value heatmaps to Supplementary Fig. 2 and now show the post-hoc comparison between the two unfamiliar C57BL/6 odor responses using box plots with lines connecting the paired values.

Please note that the differentiation of the first half of the odor presentation to the second half was neither described in the result text nor used for conclusions throughout the manuscript. We had used it originally as a sanity check and had observed that testing first and second half of the odor response are usually significant if the full one second window is significant in the analyses. So they provide little additional information. We have therefore simplified the manuscript and only show the test for the full second window throughout the Supplementary Figs..

Supplem. Fig. 3: The most relevant information (namely on the hard core data of single unit responses) can be found in Supplem. Fig. 3. From that figure it seems that the authors had very few units in the pPC and LEC responding to the different odors. In fact, how did the authors calculate the z-score (compared to baseline?).

We thank the reviewer for raising this point. We clarified in the Methods section the paragraph describing how the z-scored response for every unit was computed, l. 672: “First, responses were

averaged across trials of one odor-type, FR , and then the z-score was computed for every unit at time-bin t by: $z(t) = \frac{FR(t) - FR_{baseline}}{\sigma_{baseline}}$, where $FR_{baseline}$ and $\sigma_{baseline}$ is the mean firing rate and standard deviation during the baseline window." The baseline window for odor responses was set from -4 to 0s relative to odor onset.

Also, we used the same z-score range for the heatmaps of all regions. This comes at the price that one may underestimate the responses in pPC. We therefore provide a map with a narrower z-score range for illustration:

Line 160: "These units recruited by the familiar mouse also had stronger firing rate responses in AON and pPC as compared to the smell of the novel mouse (Supplementary Fig. 6f)."

This statement is unclear and confusing. If a unit is recruited by the familiar mouse (and thus was not considered recruited to the unfamiliar mouse), then it should of course be showing stronger firing rates as compared to the smell of the novel mouse.

What is important here is to define the criteria by which a unit is considered to be recruited or not (responsive or not) to the familiar versus novel mouse.

The question here is: HOW did the authors average their responses in supplementary figure 6f? Did they take averages only of those units that were recruited and compared these with their activities to a mouse that was novel? (e.g. the 9% that is shown in Suppl. Fig. 6e). What about the units that

responded to BOTH familiar and novel mice (grey pie chart). Were there any differences in responses of these units to familiar and non-familiar mice ?

We thank the reviewer for suggesting a more detailed analysis here. To determine whether a unit responds to an emitter, we performed a nonparametric statistical test to compare the firing rates during odor presentation of the different trials of one odor against the firing rates during the baseline window as indicated in the Methods section.

We modified the legend and panel title of now Supplementary Fig. 8b-d (and consistently also of Supplementary Fig. 13e-h). Units that respond to any of the emitters are called “all odor-responsive units”; among these, units that respond significantly to only one of the emitters are called “selective units”, and the ones that respond significantly to both are called “non-selective units”.

We added more detailed analyses and show also selective and non-selective units from AON in Supplementary Figs. 8d-e. We also added such subsample analyses for AON with optogenetic OXT release (Supplementary Fig. 13). For the other brain regions, we did not split the responses as some subgroups would have become small (n = 3-12).

The results support the conclusion that non-selective units are the largest subpopulation among all odor-responsive units in AON and that firing responses to the familiar odor are stronger, for all odor-responsive and non-selective units.

We now have changed the sentence in the results section, l. 212: “AON units excited by both conspecifics recruited by the familiar mouse had stronger mean responses than selective units and also showed stronger firing rate responses to the familiar odor in AON and pPC as compared to the smell of the novel mouse (Supplementary Fig. 8e).”

And l. 314: “Again, particularly the large fraction of units significantly excited by both conspecifics showed stronger firing rate responses to the familiar odor (Supplementary Fig. 13e-h).”

Supplementary Fig. 8, related to Fig. 2. Neuronal coding of familiarity **b** Neuronal response tuning of recorded units. For every odor type, the trial-averaged firing rate response was tested against baseline (Wilcoxon signed-rank test with Bonferroni correction for multiple comparisons). Significant increases in firing were termed odor-excited responses and significant decreases in spike rate were termed odor-inhibited responses. Mixed responses refer to units that show both significant odor-excited and odor-inhibited bins in the tested odor presentation period. Pie charts show fractions of units responding to one or more of the odors. The fractions of selective odor-excited responses to the familiar vs. selective odor-excited responses to the novel animal were compared using Fisher's exact test for the AON. **c** The temporal evolution of mean z-scored rate responses of **all** (top) odor-excited and (bottom) odor-inhibited units (mean \pm SEM). Units were considered odor-excited if activated by at least one of the two odors and vice versa for odor-inhibited cells (Wilcoxon signed rank-test comparing the mean z-scored response per unit. The corresponding p-values for the 1 s odor presentation window is shown above the respective line plots). **d** Similar to **c**, but for "selective" units that were significantly activated by either the familiar or the novel odor (Wilcoxon rank sum test was used to compare the mean z-score response per unit. The corresponding p-values for the 1 s odor presentation window is shown above the respective line plots). **e** Similar to **c**, but for "non-selective" units that were significantly activated by both the familiar and novel odor. For the other brain regions, we did not split the responses as some subgroups become small ($n = 3-12$).

References cited in the point-by-point reply.

- Bolding KA, Franks KM (2017) Complementary codes for odor identity and intensity in olfactory cortex. *eLife* 6:e22630.
- Boyd AM, Sturgill JF, Poo C, Isaacson JS (2012) Cortical feedback control of olfactory bulb circuits. *Neuron* 76:1161–1174.
- Chaudhury D, Manella L, Arellanos A, Escanilla O, Cleland TA, Linster C (2010) Olfactory bulb habituation to odor stimuli. *Behav Neurosci* 124:490–499.
- Choe HK, Reed MD, Benavidez N, Montgomery D, Soares N, Yim YS, Choi GB (2015) Oxytocin Mediates Entrainment of Sensory Stimuli to Social Cues of Opposing Valence. *Neuron* 87:152–163.
- Dölen G, Darvishzadeh A, Huang KW, Malenka RC (2013a) Social reward requires coordinated activity of nucleus accumbens oxytocin and serotonin. *Nature* 501:179–184.
- Dölen G, Darvishzadeh A, Huang KW, Malenka RC (2013b) Social reward requires coordinated activity of nucleus accumbens oxytocin and serotonin. *Nature* 501:179–184.
- Friard O, Gamba M (2016) BORIS: a free, versatile open-source event-logging software for video/audio coding and live observations. *Methods in Ecology and Evolution* 7:1325–1330.
- Haberly LB, Price JL (1978) Association and commissural fiber systems of the olfactory cortex of the rat. I. Systems originating in the piriform cortex and adjacent areas. *Journal of Comparative Neurology* 178:711–740.
- Jaegle A, Mehrpour V, Mohsenzadeh Y, Meyer T, Oliva A, Rust N (2019) Population response magnitude variation in inferotemporal cortex predicts image memorability Behrens TE, Pasternak T, Buffalo EA, eds. *eLife* 8:e47596.
- Kato HK, Chu MW, Isaacson JS, Komiyama T (2012) Dynamic Sensory Representations in the Olfactory Bulb: Modulation by Wakefulness and Experience. *Neuron* 76:962–975.
- Kay LM, Beshel J, Brea J, Martin C, Rojas-Libano D, Kopell N (2009) Olfactory oscillations: the what, how and what for. *Trends in Neurosciences* 32:207–214.
- Knobloch HS, Charlet A, Hoffmann LC, Eliava M, Khrulev S, Cetin AH, Osten P, Schwarz MK, Seeburg PH, Stoop R, Grinevich V (2012) Evoked Axonal Oxytocin Release in the Central Amygdala Attenuates Fear Response. *Neuron* 73:553–566.
- Linster C, Kelsch W (2019) A Computational Model of Oxytocin Modulation of Olfactory Recognition Memory. *eNeuro* 6 Available at: <https://www.ncbi.nlm.nih.gov/pmc/articles/PMC6727149/> [Accessed August 5, 2020].
- Markopoulos F, Rokni D, Gire DH, Murthy VN (2012) Functional Properties of Cortical Feedback Projections to the Olfactory Bulb. *Neuron* 76:1175–1188.
- Oettl L-L, Ravi N, Schneider M, Scheller MF, Schneider P, Mitre M, da Silva Gouveia M, Froemke RC, Chao MV, Young WS, Meyer-Lindenberg A, Grinevich V, Shusterman R, Kelsch W (2016) Oxytocin Enhances Social Recognition by Modulating Cortical Control of Early Olfactory Processing. *Neuron* 90:609–621.

- Oetli L-L, Scheller M, Filosa C, Wieland S, Haag F, Loeb C, Durstewitz D, Shusterman R, Russo E, Kelsch W (2020) Phasic dopamine reinforces distinct striatal stimulus encoding in the olfactory tubercle driving dopaminergic reward prediction. *Nat Commun* 11:3460.
- Otazu GH, Chae H, Davis MB, Albeanu DF (2015) Cortical Feedback Decorrelates Olfactory Bulb Output in Awake Mice. *Neuron* 86:1461–1477.
- Stettler DD, Axel R (2009) Representations of Odor in the Piriform Cortex. *Neuron* 63:854–864.
- Wahis J et al. (2021) Astrocytes mediate the effect of oxytocin in the central amygdala on neuronal activity and affective states in rodents. *Nat Neurosci* 24:529–541.
- Wilson DA (1998) Habituation of Odor Responses in the Rat Anterior Piriform Cortex. *Journal of Neurophysiology* 79:1425–1440.
- Wilson DA, Linster C (2008) Neurobiology of a Simple Memory. *Journal of Neurophysiology* 100:2–7.
- Winkelmeier L, Filosa C, Hartig R, Scheller M, Sack M, Reinwald JR, Becker R, Wolf D, Gerchen MF, Sartorius A, Meyer-Lindenberg A, Weber-Fahr W, Clemm von Hohenberg C, Russo E, Kelsch W (2022) Striatal hub of dynamic and stabilized prediction coding in forebrain networks for olfactory reinforcement learning. *Nat Commun* 13:3305.
- Yamada Y, Bhaukaurally K, Madarász TJ, Pouget A, Rodriguez I, Carleton A (2017) Context- and Output Layer-Dependent Long-Term Ensemble Plasticity in a Sensory Circuit. *Neuron* 93:1198–1212.e5.

REVIEWER COMMENTS

Reviewer #1 (Remarks to the Author):

The authors have addressed our prior concerns. After revisions this is an even stronger and more interesting story. well done!

Reviewer #2 (Remarks to the Author):

The authors have satisfactorily addressed most concerns raised in the initial review and substantially improved an already strong manuscript.

Reviewer #3 (Remarks to the Author):

The authors have responded fully to my comments and questions in the earlier submission, and I have no further comments or concerns.

Reviewer #4 (Remarks to the Author):

In this second version, the authors have taken out the PCA analysis to purely concentrate on the vectorial representation of the neuronal responses to the odor. With this they then tried to test the hypothesis that this vector increases in Euclidian distance from baseline after familiarization with a previously novel olfactory stimulus (another mouse).

To exemplify their approach the authors introduced a new citation that was not mentioned in the first version of their manuscript (Jaegle et al. 2019). In Jaegle et al, two macaques are presented with images twice, and the macaques have to indicate whether the image was new or not with an eye movement. Recordings are made in the inferotemporal cortex and of hundreds of neurons that are collected by concatenation over several weeks (5 for macaque 1 and 4 for macaque 2). Through

these findings Jaegle et al show that in macaques, the population response magnitude variation predicts image memorability.

Wolf et al. then propose, that also for the odor stimulus of another mouse they should find the same changes after familiarization and base their further analysis on this approach. However the comparison with Jaegle et al., 2019 seems to go awry in several aspects:

1. . It seems the authors claim their work should find, by analogy, similar outcome as the Jaegle study. However Jaegle et al. recordings are made in different species (macaques vs mice), different brain regions (IT in macaques vs. AON, pPC and LEC in mice) and different types of stimuli (visual versus olfactory). These therefore do not allow for a direct extrapolation to the current work and the authors really still need to conduct the same type of test in their experimental paradigm as performed by Jaegle et al.
2. The approach taken by the authors in their further design of their study further differs from Jaegle et al in that the work in macaques is still longitudinal: indeed, both novel and familiar stimuli are presented and recorded in the same animal (by sequentially presenting the same visual stimulus twice)

While I appreciate the admittance by the authors that they had indeed considered a longitudinal approach as a better one for substantiating their claims, the arguments they put forward for not having followed this seem quite weak. They claim that the double exposure to the stereotaxic frame would could a loss of original neuronal unit recordings and thus provide no advantage over a transversal approach. To substantiate their claim they refer to some previous experience with longitudinal recordings, but they do not show any hard core evidence to substantiate this claim.

1. It should not be too difficult to record twice from the same mouse on the same day. This, in fact, casts quite some doubt on the technical skills of the experimental part. In electrophysiology of this kind in mice, one should be able to obtain a good quality second series recordings certainly the same day, if not over several days. It suffices to protect the head implant against the approach of the other animal. The exposure to another rodent for familiarization should not lead to too much changes in the neurons that are being recorded.
2. Furthermore, even if a loss of units occurs, , it is rather simple to assess how well the second series of recordings matches the first one. Especially with tetrodes, one can immediately assess through the shapes of the waveforms across the four electrodes how many of the original neurons are still present at the second recording.
3. And finally, even if the second recording would show that a completely different set of neurons is now being recorded, then this would provide an excellent opportunity to precisely substantiate the hypothesis of the authors, namely that this second set of neurons (which would show up as a change in angle of the representation vector) should have now an increased amplitude as it represents a second exposure to the same stimulus.

Thus, the arguments for not using a longitudinal analysis does not seem to hold up to further examination.

On the contrary, the claim that the recordings in the mice are so unstable as to not allow for a second measurement after familiarization with the novel mouse odor stimulus, casts doubt on the quality of the current findings in this manuscript.

If these recordings are so unstable, it seems rather unlikely that repeated measurements over time (in the same animal) and between different observer animals would record from the same neuronal populations. This needs to be much better demonstrated by an in depth analysis of the stability of the waveform shapes of the tetrodes over time.

The claim that "a linear classifier reliably predicted the identity of the different odors from the neuronal activation pattern" needs to be explained much better. The accuracy of this test is not convincingly demonstrated by the current illustrations.

While I appreciate the ground truth representations of the recordings of all units in supplementary figures 7, 11 13 & 15 that show side by side comparisons between novel and familiar stimuli, these findings do not show any striking differences in intensity between familiar and novel stimuli.

In this regard I am also not sure how the two observations about sniffing frequency and Euclidian distance can fit together. On one hand the authors present no correlation for individual observations (Euclidian distance from baseline versus sniffing frequency, Supplem. Fig. 9b&c). On the other hand the authors provide a correlation between "changes in sniffing frequency" and "changes in Euclidian baseline distance" between familiar versus non familiar mice (Supplem. Fig. 9d). If a memory for an odor is being encoded at the level of the Euclidian distance of the vector, this would mean that the odor concentration should provoke a higher response "per sniff", regardless of the sniff frequency. However, in supplementary figures 9b versus 9c there does not seem to be an overall shift of Euclidian distance to a higher level for familiar as compared to novel responses.

Taken together, I am not sure that the manuscript has much increased in strength by the current changes. First of all the PCA analysis is taken out (for not so clear reasons) which rather weakens the presentation of the data. Second, the argumentation for not using longitudinal analysis because of unstable recordings rather reveals some important weaknesses on the electrophysiological recordings that cast doubt on the reliability of the data. Finally, the ground truth data are not very convincing in showing clear differences in intensity of responses in the AON between novel and familiar stimuli.

Reply to Reviewers Wolf et al. (2nd round of revision)

Reviewers #1, #2 and #3 had no further questions and recommended the manuscript for publication. We would like to thank them for their constructive suggestions, which have further improved the manuscript.

Reviewer #4 (Remarks to the Author):

We have read and discussed in depth the points raised by Reviewer #4. We would like to respectfully explain the reasons why we disagree with the Reviewer's comments and attempt to clarify any misunderstandings in the point-by-point response.

The complete comment of Reviewer #4 is marked in **bold** and our clarification in regular font:

In this second version, the authors have taken out the PCA analysis to purely concentrate on the vectorial representation of the neuronal responses to the odor. With this they then tried to test the hypothesis that this vector increases in Euclidian distance from baseline after familiarization with a previously novel olfactory stimulus (another mouse).

Following the suggestion in the first major point of Reviewer #3, the PCA trajectories (a visualization tool) were substituted by confusion matrices (output of a quantitative classifier analysis).

To exemplify their approach the authors introduced a new citation that was not mentioned in the first version of their manuscript (Jaegle et al. 2019). In Jaegle et al, two macaques are presented with images twice, and the macaques have to indicate whether the image was new or not with an eye movement. Recordings are made in the inferotemporal cortex and of hundreds of neurons that are collected by concatenation over several weeks (5 for macaque 1 and 4 for macaque 2). Through these findings Jaegle et al show that in macaques, the population response magnitude variation predicts image memorability.

Wolf et al. then propose, that also for the odor stimulus of another mouse they should find the same changes after familiarization and base their further analysis on this approach. However the comparison with Jaegle et al., 2019 seems to go awry in several aspects:

1. . It seems the authors claim their work should find, by analogy, similar outcome as the Jaegle study. However Jaegle et al. recordings are made in different species (macaques vs mice), different brain regions (IT in macaques vs. AON, pPC and LEC in mice) and different types of stimuli (visual versus olfactory). These therefore do not allow for a direct extrapolation to the current work and the authors really still need to conduct the same type of test in their experimental paradigm as performed by Jaegle et al.

Jaegle et al. is a paper mentioned in a single sentence of the discussion L. 341-2: "Scaling of responses has also been observed in primates for salience coding of visual objects [49]." The sentence was added in the revision because Jaegle et al. also investigated changes in the strength of population responses to stimuli and we wanted to give credit and embed our work within the literature. It was not our intent to replicate Jaegle et al. or directly compare to our work. Our approach has been detailed and put in context with the field's methods in the manuscript. Its validity stands on its own and does not rely on direct comparison with the aforementioned study.

Moreover, we feel that comparison to results in other species and brain regions is appropriate and necessary for a thorough discussion of our data. Generally speaking, applying analyses and

observing principles across species and experiments can strengthen the scientific findings and is the basis for the search of general principles.

2. The approach taken by the authors in their further design of their study further differs from Jaegle et al in that the work in macaques is still longitudinal: indeed, both novel and familiar stimuli are presented and recorded in the same animal (by sequentially presenting the same visual stimulus twice)

We examined the neuronal substrate of social recognition in a social exploration-recognition sequence that is, with small variations, the standard used in more than a thousand rodent behavioral studies. To investigate the neuronal bases of this broadly studied recognition memory, its behavioral sequence, composed of an initial interaction step (when familiarity with a new animal is acquired) followed by the recognition step, has to be maintained. In these studies, recognition is measured as the difference in responses to familiar and novel animals. We followed this sequence and tested the recognition memory after an initial interaction step.

In contrast, Jaegle et al. examined visual short-term working memory with a juice reward, and thus a different type of learning than social recognition paradigms in rodents. We will explain the reasons for the design of our experiment and the problems that would arise with the proposed longitudinal design in more detail below (in section 3. of the Reviewer's remarks).

While I appreciate the admittance by the authors that they had indeed considered a longitudinal approach as a better one for substantiating their claims, the arguments they put forward for not having followed this seem quite weak.

Please note that we never claimed “**a longitudinal approach as a better one for substantiating their claims**” (Reviewer statement). In our reply in the first review round, we discussed one reason why longitudinal multi-session concatenation can have its problems. At this point, we would like to emphasize once again that our motivation for the choice of the experimental design was not based on technical concerns regarding longitudinal recordings, but on the biological question (see reply to point 3. of the Reviewer further below). These considerations are described in the introduction, result section and discussion.

They claim that the double exposure to the stereotaxic frame would could a loss of original neuronal unit recordings and thus provide no advantage over a transversal approach. To substantiate their claim they refer to some previous experience with longitudinal recordings, but they do not show any hard core evidence to substantiate this claim.

1. It should not be too difficult to record twice from the same mouse on the same day. This, in fact, casts quite some doubt on the technical skills of the experimental part. In electrophysiology of this kind in mice, one should be able to obtain a good quality second series recordings certainly the same day, if not over several days. It suffices to protect the head implant against the approach of the other animal. The exposure to another rodent for familiarization should not lead to too much changes in the neurons that are being recorded.

Given our experience with these types of experiments, it is not a problem to record multiple times a day. We recorded from the mice in the present study for many weeks. The rationale for the current design lies in the biology of social recognition. In the behavioral literature, the social recognition experiment is performed usually not more than once per week in rodents. We aligned our experimental design to this standard. Moreover, changing our paradigm to multiple separate recording sessions on the same day would leave a margin of uncertainty in matching the identity of units recorded in separate sessions and, more importantly, would be detrimental to the validity

of investigating neural processes of memory formation relevant to the social recognition paradigm, which has a long tradition in behavioral research (as detailed further below related to point 3. of the Reviewer).

2. Furthermore, even if a loss of units occurs, , it is rather simple to assess how well the second series of recordings matches the first one. Especially with tetrodes, one can immediately assess through the shapes of the waveforms across the four electrodes how many of the original neurons are still present at the second recording.

The advantage of performing a head-fix recording before and after the interaction (= freely moving exploration) only becomes relevant when aiming to elucidate changes in single-unit coding properties across sessions, such as tracking the identity code of odors in the population. Although criteria for identifying cells in consecutive sessions can be defined (e.g. Tolias et al., (2007) J Neurophysiol 98: 3780-3790), it is not possible to prove the accuracy of such unit identity assignment with certainty. This conservative standpoint is a widely held view in the field.

3. And finally, even if the second recording would show that a completely different set of neurons is now being recorded, then this would provide an excellent opportunity to precisely substantiate the hypothesis of the authors, namely that this second set of neurons (which would show up as a change in angle of the representation vector) should have now an increased amplitude as it represents a second exposure to the same stimulus.

We find this suggestion surprising. Either a longitudinal approach is needed, in which case we would need to perform the analyses on the same set of units, or is not, in which case the experimental design chosen in our manuscript is to be favored as it controls for confounds (see next point below). Furthermore, in the speculative case of a second recording session with a completely different set of neurons, it would be interpreted as an issue with the unit-matching algorithm or electrode position drifts, but not as a sign of reinforced responses per se.

Thus, the arguments for not using a longitudinal analysis does not seem to hold up to further examination.

Here, we would like to address arguments in connection with a longitudinal design:

In the first review round, Reviewer #4 (major point 2.) suggested: "...the crucial experiment to address their hypothesis would be to make longitudinal comparisons between neuronal representations before and after familiarization with the same mouse."

If we understand the Reviewer correctly, we should have run the following experiment:

Longitudinal Paradigm A:

- Step 1 in head-fix: emitter #1
- Step 2 as freely moving interaction: emitter #1
- Step 3 in head-fix: emitter #1 & #2

Or, alternatively, as Reviewer #4 does not specify in detail the paradigm he wants:

Longitudinal Paradigm B:

- Step 1 in head-fix: emitter #1 & #X
- Step 2 as freely moving interaction: emitter #1
- Step 3 in head-fix: emitter #1 & #2

Several shortcomings arise from such “longitudinal” paradigms, in which animals undergo an emitter presentation before the memory-inducing interaction:

- The first problem is that the uncertainty of obtaining the exact same units in step 1 and 3 remains, thus it is difficult to claim a truly longitudinal examination.
- In paradigm A, the effects of repeated testing are not controlled for (like changes in attention that affect population response strength from step 1 to 3). This could be controlled for by paradigm B. However, if the emitter #X is #2, then #2 is not novel anymore during the recognition step, but it would become a “familiar odor from someone I never interacted with”. And, if #X is an animal not used in step 3, the comparison would be essentially transversal. As a note, between the two paradigms described above, the paradigm B becomes more similar to ours, however with the following critical problem:
- Importantly, it is not clear in how far induction of memory achieved by the freely-moving exploration of animal #1 is hampered by pre-exposure to its odor in step 1 (paradigms A and B). Thus, it would not be anymore the social recognition paradigm used in a thousand studies in the last decades, therefore not help to explain its neuronal underpinnings. In the longitudinal paradigms, other types of plasticity may occur, that are not in the focus of standard recognition memory. This would have to be studied as an independent memory mechanism of questionable ethological relevance.

These problems are taken into account in our original paradigm in which we used a classic exploration-recognition sequence without emitter odor exposure before the exploration, and on a separate day examine the response to purely novel odor emitters as control.

In Fig. 1, we show that the population response amplitude is not different between novel emitters (Fig. 1), while the population response amplitude differs between #1 and #2 following the exploration step (Fig. 2-5). Note that the same groups of animals were used for Fig. 1 and the recognition test. This has multiple advantages:

- The observed difference in population response is clearly attributable to the preceding social interaction since familiarity with #1 is the only feature differentiating the odors of #1 and #2. We do not run into the problem that the neuronal phenomena may describe a different memory induction mechanism during freely moving exploration due to odor “pre”-exposure.
- The animals undergo only one session on that day, thus attentional effects are less of a concern and matched in the control.
- As an important note, we additionally show that the difference in population amplitude during recognition is lost in the OXTR-KO, and boosted when optogenetically enhancing oxytocin release during the interaction step.

On the contrary, the claim that the recordings in the mice are so unstable as to not allow for a second measurement after familiarization with the novel mouse odor stimulus, casts doubt on the quality of the current findings in this manuscript.

If these recordings are so unstable, it seems rather unlikely that repeated measurements over time (in the same animal) and between different observer animals would record from the same neuronal populations. This needs to be much better demonstrated by an in depth analysis of the stability of the waveform shapes of the tetrodes over time.

As detailed in the method section (L. 600-15), standard quality measures for the isolation of single units and the waveform stability were performed and only stable units were included in the data set. Also, it should be noted that in the present study, we recorded from the mice for many weeks.

The claim that “a linear classifier reliably predicted the identity of the different odors from the

neuronal activation pattern" needs to be explained much better. The accuracy of this test is not convincingly demonstrated by the current illustrations.

The significance of the classifier analysis is explicitly reported in the result section "(Fig. 1e; Fisher's exact test against classifier trained with shuffled labels; AON: $p < 0.0001$, pPC: $p < 0.0001$, LEC: $p < 0.0001$)", the classifier confusion matrices are shown in Fig. 1e, and the classifier and statistical testing are described in detail in the method section (L. 649-58).

The "current illustrations", which are actually quantitative heatmaps (Fig. 1e), show few classification mistakes, in line with the result of the statistical tests.

While I appreciate the ground truth representations of the recordings of all units in supplementary figures 7, 11 13 &15 that show side by side comparisons between novel and familiar stimuli, these findings do not show any striking differences in intensity between familiar and novel stimuli.

The statement "**these findings do not show any striking differences in intensity between familiar and novel stimuli**" (Reviewer) is a subjective evaluation, as the differences between familiar and novel are statistically significant. The heatmaps were plotted to give the reader an impression of the general response patterns in the population to different stimuli, but are not suited to draw conclusions on population response intensities, which require statistical testing instead.

Moreover, indicative of the robustness of the detected difference in response to familiar and novel stimuli, the study contains replications of the finding in multiple cohorts, brain regions, and readouts (single unit population activity, LFP, calcium signals). Further, we show that the amplitude difference is lost in the OXTR-KO. And additionally, a linear mixed effect model (that considers the hierarchical structure of the data) confirms that response amplitude differences for familiar versus novel are differently expressed in the conditions OXTR-KO < control condition < boosted OXT in Fig. 5g.

In this regard I am also not sure how the two observations about sniffing frequency and Euclidian distance can fit together. On one hand the authors present no correlation for individual observations (Euclidian distance from baseline versus sniffing frequency, Supplem. Fig. 9b&c). On the other hand the authors provide a correlation between "changes in sniffing frequency" and "changes in Euclidian baseline distance" between familiar versus non familiar mice (Supplem. Fig. 9d). If a memory for an odor is being encoded at the level of the Euclidian distance of the vector, this would mean that the odor concentration should provoke a higher response "per sniff", regardless of the sniff frequency. However, in supplementary figures 9b versus 9c there does not seem to be an overall shift of Euclidian distance to a higher level for familiar as compared to novel responses.

These statements and conclusions seem to be based on a misunderstanding. The trial-by-trial correlation of sniff rate and population response to one of the odors (Suppl. Figure 9b,c) served to clarify whether neuronal amplitude differences can be explained by variations in the sniff rate (as a possible confound). This was not the case and is important for the conclusion that the stronger neuronal response to the familiar odors is not due to a central sniff modulation. Importantly, we did not test the trial-by-trial variation for the difference in population amplitude between familiar and novel odors as the Reviewer is implying. This would not have been informative.

These points are described in the result section starting L. 174: "The stronger neuronal response in the AON to the familiar odor was however not explained by the sniff rate modulation in a trial-by-trial correlation (Supplementary Fig. 9b-c). We tested the prediction that the stronger

the memory, the more it reflects in both session-wise average sniff and neuronal responses. Indeed, the difference in mean firing rate response or Euclidean distance from baseline correlated positively with simultaneously recorded sniff response differences between familiar and novel emitters (Supplementary Fig. 9d-e).”

Taken together, I am not sure that the manuscript has much increased in strength by the current changes. First of all the PCA analysis is taken out (for not so clear reasons) which rather weakens the presentation of the data. Second, the argumentation for not using longitudinal analysis because of unstable recordings rather reveals some important weaknesses on the electrophysiological recordings that cast doubt on the reliability of the data. Finally, the ground truth data are not very convincing in showing clear differences in intensity of responses in the AON between novel and familiar stimuli.

Please see our responses to the individual points raised.

REVIEWER COMMENTS

Reviewer #4 (Remarks to the Author):

We thank the authors for their responses to our questions.

The manuscript of Wolf et al. stands out for addressing trying to understand how social recognition memory is encoded at the level of the olfactory bulb and cortex. The authors advance two possible hypotheses taken from references 18-21 and 23: They propose that it is either encoded by habituation (lower salience of familiar subjects) or through experience-dependent plasticity (stronger neuronal responses than a novel conspecific). It is, by the way, not sure whether lower salience might also be caused by experience dependent plasticity.

The articles that the authors quote are based on either calcium imaging or single unit recordings with single electrode in head-fixed animals. The novelty of the approach that the authors use is that they approach this question with custom-built chronic tetrode arrays, which allows for simultaneous recordings over time of multi single units. This poses the challenge of making sure that over time the recordings of these single units are stable. On the other hand this approach allows to assess, compared to the previous work they quote (18-21), whether the spiking activity of multi-single units encodes for familiarity. With their method, the authors can capture the simultaneous activity of 75 units (fig. 1, AON) and similar numbers or more in other regions. The challenge with this large number of neurons is now to distinguish whether familiarity is encoded by increases or decreases in responses of these neurons as formulated in the initial two alternative hypotheses above. To determine how this population of neuronal responses encodes for different odors (novel or familiar), the authors construct a population vector of the responses of all neurons in a multidimensional space. Each odor is thus encoded by the direction (orientation) of the vector in the multidimensional space and the length of the vector (amplitude of the responses). The authors then go to show that for different odors that the orientation of the vector shows a difference. The length of the vector (Euclidian distance from baseline) seems not significantly different between different odors across the different regions recorded (AON, pPC, LEC). This is rather remarkable, because the orientation are different (Fig. 1c) meaning that distinct neurons encode distinct odors. If that is the case, then the absolute amplitude from baseline would likely depend on the subpopulation of neurons that are captured by the tetrode array that the authors capture. This means that there is no topological distribution of subunits encoding for different odors and the authors should recognize this and compare it with previous work done in these areas.

Subsequently, the authors hypothesize that "these neurons encode features like familiarity in their overall response amplitude" (legend of figure 1). And they analyse this by Euclidian distance from baseline.

It is here that their hypothesis is too narrowly positioned and ignores the possibility that not just the overall response amplitude changes following familiarity, but also the direction of the vector. To give a counter-example: It is possible that from the total neuronal population that they recorded, some neurons may increase in amplitude and others decrease in amplitude following familiarization with the odor. In total, this may still lead to an overall increase in response amplitude to the odor, but this does not take into account that the encoding of the familiarity still also requires a subpopulation of neurons to decrease in amplitude responses for the encoding of familiarity. Without measuring changes in the orientation of the vector, one can therefore NOT conclude that familiarity is encoded by enhanced responses of the neuronal population that the authors measured their responses from. In order to address this, it is necessary to record and compare neuronal responses before and after familiarization with the odor.

In their responses, the authors suggest an experiment that indeed can address this question.

It is the longitudinal paradigm B, in which they expose the recorded mouse in step 1 to emitter 1 and emitter 2, in step 2 expose the recorded mouse in freely interaction with emitter 1 (and not emitter 2) and subsequently again in step 3 expose the recorded mouse in head fix to emitter 1 and 2. The encoding of social familiarization can then be extracted by the orientation and length of the population vector of encoded neurons.

As the authors acknowledge, this would require verification that the same neuronal population is recorded from in the headfix situations of step 1 and step 3. To show that this is indeed the case, the authors should define very clear criteria for identifying stability of single unit recordings,

The authors indicate in their responses that "Given their experience with these types of experiments, it is not a problem to record multiple times a day. We recorded from the mice in the present study for many weeks. ", so it seems this should not be a problem to demonstrate the above point.

Minor points:

It would be helpful if the authors could be more precise in their answers with precise references for the following points:. Notably:

1. They claim that it is not possible to prove accuracy of such unit identity assignment with certainty. This conservative standpoint is a widely held view in the field".

2. "Importantly, it is not clear in how far induction of memory achieved by the freely-moving exploration of animal #1 is hampered by pre-exposure to its odor in step 1 (paradigms A and B)."

This social recognition paradigm is used in a thousand studies in the last decade". It is unclear to the reviewer to which studies the authors refer. The precise argument against Paradigm B remains unclear.

3. The authors mention that they used "standard quality measures for the isolation of single units and the waveform stability were performed and only stable units were included in the data set. Also, it should be noted that in the present study, we recorded from the mice for many weeks". Could the authors provide more precise information for their quality measures and also provide additional data showing their recordings "for many weeks"?

The following references may serve as guidance:

-Okun et al. (2016): <https://journals.plos.org/plosone/article?id=10.1371/journal.pone.0151180>.

-Rossant et al. (2016): <https://www.ncbi.nlm.nih.gov/pmc/articles/PMC4817237/>.

-Chung et al. (2017): <https://www.ncbi.nlm.nih.gov/pmc/articles/PMC5743236/> (not head-fixed, but the algorithm is relevant)

Reply to Reviewers Wolf et al. (3rd round of revision)

Reviewers #1, #2 and #3 had no further questions and recommended the manuscript for publication. We would like to thank them for their constructive suggestions, which have further improved the manuscript.

Reviewer #4 (Remarks to the Author):

We thank the authors for their responses to our questions.

The manuscript of Wolf et al. stands out for addressing trying to understand how social recognition memory is encoded at the level of the olfactory bulb and cortex. The authors advance two possible hypotheses taken from references 18-21 and 23: They propose that it is either encoded by habituation (lower salience of familiar subjects) or through experience-dependent plasticity (stronger neuronal responses than a novel conspecific). It is, by the way, not sure whether lower salience might also be caused by experience dependent plasticity. The articles that the authors quote are based on either calcium imaging or single unit recordings with single electrode in head-fixed animals. The novelty of the approach that the authors use is that they approach this question with custom-built chronic tetrode arrays, which allows for simultaneous recordings over time of multi single units. This poses the challenge of making sure that over time the recordings of these single units are stable. On the other hand this approach allows to assess, compared to the previous work they quote (18-21), whether the spiking activity of multi-single units encodes for familiarity. With their method, the authors can capture the simultaneous activity of 75 units (fig. 1, AON) and similar numbers or more in other regions. The challenge with this large number of neurons is now to distinguish whether familiarity is encoded by increases or decreases in responses of these neurons as formulated in the initial two alternative hypotheses above. To determine how this population of neuronal responses encodes for different odors (novel or familiar), the authors construct a population vector of the responses of all neurons in a multidimensional space. Each odor is thus encoded by the direction (orientation) of the vector in the multidimensional space and the length of the vector (amplitude of the responses). The authors then go to show that for different odors that the orientation of the vector shows a difference. The length of the vector (Euclidian distance from baseline) seems not significantly different between different odors across the different regions recorded (AON, pPC, LEC). This is rather remarkable, because the orientation are different (Fig. 1c) meaning that distinct neurons encode distinct odors. If that is the case, then the absolute amplitude from baseline would likely depend on the subpopulation of neurons that are captured by the tetrode array that the authors capture. This means that there is no topological distribution of subunits encoding for different odors and the authors should recognize this and compare it with previous work done in these areas. Subsequently, the authors hypothesize that "these neurons encode features like familiarity in their overall response amplitude" (legend of figure 1). And they analyse this by Euclidian distance from baseline. It is here that their hypothesis is too narrowly positioned and ignores the possibility that not just the overall response amplitude changes following familiarity, but also the direction of the vector. To give a counter-example: It is possible that from the total neuronal population that they recorded, some neurons may increase in amplitude and others decrease in amplitude following familiarization with the odor. In total, this may still lead to an overall increase in response amplitude to the odor, but this does not take into account that the encoding of the familiarity still also requires a subpopulation of neurons to decrease in amplitude responses for the encoding of familiarity. Without measuring changes in the orientation of the vector, one can therefore NOT conclude that familiarity is encoded by enhanced responses of the neuronal population that the authors measured their responses from.

Reply: In response to the comments of Reviewer #4, we respectfully like to clarify the points raised by the reviewer. We do not imply that only amplitude and not direction of the population vector change in response to learning, in contrast, we show that the distance between the familiar and

novel odor increases after successful learning only, and conclude that in l. 310-1: “In summary, SRM is encoded in the olfactory cortices and associated regions by reinforced and more distinct representations.” The question raised by the reviewer has already been answered with our data: before or in the absence of learning (Fig. 1e, Fig. 5i, Suppl. Fig. 15h-i), neural representations of distinct conspecific odors are less distinct from each other than after successful learning (Fig. 5h, Suppl. Fig. 13j). Hence, our data clearly show increased average response amplitudes (distance from baseline) and changes in the angle spanned between two odors (direction of vector). Changes in the orientation of the vectors can be implied by comparing odors to each other before learning, after successful learning, and after unsuccessful learning in the OXTR knockout mice.

In order to address this, it is necessary to record and compare neuronal responses before and after familiarization with the odor.

In their responses, the authors suggest an experiment that indeed can address this question. It is the longitudinal paradigm B, in which they expose the recorded mouse in step 1 to emitter 1 and emitter 2, in step 2 expose the recorded mouse in freely interaction with emitter 1 (and not emitter 2) and subsequently again in step 3 expose the recorded mouse in head fix to emitter 1 and 2. The encoding of social familiarization can then be extracted by the orientation and length of the population vector of encoded neurons.

As the authors acknowledge, this would require verification that the same neuronal population is recorded from in the headfix situations of step 1 and step 3. To show that this is indeed the case, the authors should define very clear criteria for identifying stability of single unit recordings,

The authors indicate in their responses that "Given their experience with these types of experiments, it is not a problem to record multiple times a day. We recorded from the mice in the present study for many weeks. ", so it seems this should not be a problem to demonstrate the above point.

Reply: The reviewer has asked again that a design be used in which neural responses to the familiar and novel odor be recorded before and after the learning process. As we discussed in our previous responses, this is not a logical way to proceed. In order to do these recordings in the same animal, we would have to proceed with multiple presentations of the familiar and novel odors before learning; as a consequence, the novel odor is rendered familiar and the paradigm will not work. Hence this proposed way of proceeding would not be testing the neural mechanisms underlying the widely used behavioral paradigm. We do not “suggest” this experiment as implied by the reviewer but rather discuss that it would not be an appropriate way to proceed (see reply to the minor point 2 below here with a copy of the detailed response given the previous round).

For clarity, we removed the sentence on cross-session unit matching as it is little relevant here given the biological phenomenon under study and added in l. 495-8: “**In this study, we either compared the response to two unfamiliar body odors as a control condition or, in a separate experiment, the response to a familiar and an unfamiliar mouse. This design was chosen to maintain the order of the social recognition test.**”

(l.514-7): “**A longitudinal experimental design for this question is not possible with tetrode recordings, because we cannot be certain that the same units are included in the population, when switching back-and-forth between head-fixed and freely-moving conditions.**”

Minor points:

It would be helpful if the authors could be more precise in their answers with precise references for the following points:. Notably:

1. They claim that it is not possible to prove accuracy of such unit identity assignment with certainty. This conservative standpoint is a widely held view in the field".

Reply: As discussed in the previous round of the review process and the main points above, the key argument for not performing a longitudinal multi-session experiment is the biology of the social recognition memory. Further, we want to reemphasize here that our above cited statement simply indicated that, even if (part of the) units can be identified with high likelihood in multiple sessions, a certain level of uncertainty remains in a concatenation approach. This consideration would be relevant in a hypothetical case where concatenation of multiple sessions is used in questions where the entire population of sampled units must match across all sessions.

2. "Importantly, it is not clear in how far induction of memory achieved by the freely-moving exploration of animal #1 is hampered by pre-exposure to its odor in step 1 (paradigms A and B)." This social recognition paradigm is used in a thousand studies in the last decade". It is unclear to the reviewer to which studies the authors refer. The precise argument against Paradigm B remains unclear.

Reply: Examples of relevant studies using the social recognition paradigm include Refs. 2, 4, 5, 6, 9, 10, 11, 12, 13, 16, 17, 22, 28, 30, and 42 in the manuscript.

The precise argument against the hypothetical Paradigm B is embedded in a chain of arguments in the previous reply. In the following, we copy this chain of arguments (text from the previous reply) and highlight the detailed argument against the hypothetical paradigm B (underlined text):

"In the first review round, Reviewer #4 (major point 2.) suggested: "...the crucial experiment to address their hypothesis would be to make longitudinal comparisons between neuronal representations before and after familiarization with the same mouse."

If we understand the Reviewer correctly, we should have run the following experiment:

Longitudinal Paradigm A:

- Step 1 in head-fix: emitter #1
- Step 2 as freely moving interaction: emitter #1
- Step 3 in head-fix: emitter #1 & #2

Or, alternatively, as Reviewer #4 does not specify in detail the paradigm he wants:

Longitudinal Paradigm B:

- Step 1 in head-fix: emitter #1 & #X
- Step 2 as freely moving interaction: emitter #1
- Step 3 in head-fix: emitter #1 & #2

Several shortcomings arise from such "longitudinal" paradigms, in which animals undergo an emitter presentation before the memory-inducing interaction:

- The first problem is that the uncertainty of obtaining the exact same units in step 1 and 3 remains, thus it is difficult to claim a truly longitudinal examination.
- In paradigm A, the effects of repeated testing are not controlled for (like changes in attention that affect population response strength from step 1 to 3). This could be controlled for by paradigm B. However, if the emitter #X is #2, then #2 is not novel anymore during the recognition step, but it would become a "familiar odor from someone I never interacted with". And, if #X is an animal not used in step 3, the comparison would be essentially transversal. As a note, between the two paradigms described above, the paradigm B becomes more similar to ours, however with the following critical problem:

- Importantly, it is not clear in how far induction of memory achieved by the freely-moving exploration of animal #1 is hampered by pre-exposure to its odor in step 1 (paradigms A and B). Thus, it would not be anymore the social recognition paradigm used in a thousand studies in the last decades, therefore not help to explain its neuronal underpinnings. In the longitudinal paradigms, other types of plasticity may occur, that are not in the focus of standard recognition memory. This would have to be studied as an independent memory mechanism of questionable ethological relevance.

These problems are taken into account in our original paradigm in which we used a classic exploration-recognition sequence without emitter odor exposure before the exploration, and on a separate day examine the response to purely novel odor emitters as control.

In Fig. 1, we show that the population response amplitude is not different between novel emitters (Fig. 1), while the population response amplitude differs between #1 and #2 following the exploration step (Fig. 2-5). Note that the same groups of animals were used for Fig. 1 and the recognition test. This has multiple advantages:

- The observed difference in population response is clearly attributable to the preceding social interaction since familiarity with #1 is the only feature differentiating the odors of #1 and #2. We do not run into the problem that the neuronal phenomena may describe a different memory induction mechanism during freely moving exploration due to odor “pre”-exposure.
- The animals undergo only one session on that day, thus attentional effects are less of a concern and matched in the control.
- As an important note, we additionally show that the difference in population amplitude during recognition is lost in the OXTR-KO, and boosted when optogenetically enhancing oxytocin release during the interaction step.“

3. The authors mention that they used "standard quality measures for the isolation of single units and the waveform stability were performed and only stable units were included in the data set. Also, it should be noted that in the present study, we recorded from the mice for many weeks". Could the authors provide more precise information for their quality measures and also provide additional data showing their recordings "for many weeks"?

The following references may serve as guidance:

-Okun et al. (2016): <https://journals.plos.org/plosone/article?id=10.1371/journal.pone.0151180>.

-Rossant et al. (2016): <https://www.ncbi.nlm.nih.gov/pmc/articles/PMC4817237/>.

-Chung et al. (2017): <https://www.ncbi.nlm.nih.gov/pmc/articles/PMC5743236/> (not head-fixed, but the algorithm is relevant)

Reply: We thank the Reviewer for the references that, together with recent tools like UnitMatch from the Carandini and Harris groups (<https://doi.org/10.1101/2023.10.12.562040>), are developed for multisession concatenation. Yet, for the biologic reasons described above, a multi-session approach is not suited for studying memory mechanisms underlying the behavioral social recognition paradigm.